# CHANGE POINT LOCALIZATION AND INFERENCE IN DYNAMIC MULTILAYER NETWORKS

**Fan Wang**
School of Mathematics and Statistics
University of Melbourne
`fan.wang.2@unimelb.edu.au`

**Kyle Ritscher**
Department of Statistics
University of California, Los Angeles
`kritscher@g.ucla.edu`

**Yik Lun Kei**
Department of Statistics
University of California, Santa Cruz
`ykei@ucsc.edu`

**Xin Ma**
Department of Biostatistics
Columbia University
`xm2141@cumc.columbia.edu`

**Oscar H. Madrid Padilla**
Department of Statistics
University of California, Los Angeles
`oscar.madrid@stat.ucla.edu`

## ABSTRACT

We study offline change point localization and inference in dynamic multilayer random dot product graphs (D-MRDPGs), where at each time point, a multilayer network is observed with shared node latent positions and time-varying, layer-specific connectivity patterns. We propose a novel two-stage algorithm that combines seeded binary segmentation with low-rank tensor estimation, and establish its consistency in estimating both the number and locations of change points. Furthermore, we derive the limiting distributions of the refined estimators under both vanishing and non-vanishing jump regimes. To the best of our knowledge, this is the first result of its kind in the context of dynamic network data. We also develop a fully data-driven procedure for constructing confidence intervals. Extensive numerical experiments demonstrate the superior performance and practical utility of our methods compared to existing alternatives.

## 1 INTRODUCTION

Statistical network analysis models entities as nodes and their interactions as edges. While single-layer networks capture pairwise interactions efficiently, many real-world systems involve multiple types of interaction among the same set of nodes. Multilayer networks address this complexity by organizing these varied interactions into distinct layers over a common node set, enabling both the capture of heterogeneity and the identification of shared latent structures. In practice, network structures often evolve over time. For instance, transportation networks may exhibit gradual diurnal variations or sudden structural changes due to accidents or road closures. Detecting such sudden shifts and providing adaptive strategies, such as dynamic traffic signal control or rerouting recommendations, is crucial for efficient transportation management. These abrupt structural shifts are referred to as change points. This naturally falls in the territory of change point analysis.

Change point analysis is a well-established area in statistics concerned with detecting abrupt structural changes in ordered data. It can be broadly classified into online and offline settings, depending on whether data are analyzed sequentially as they are collected or retrospectively after the full dataset has been observed. In the context of dynamic networks, online change point detection has been studied in models such as inhomogeneous Bernoulli networks (e.g., Yu et al., 2021) and random weighted edge networks (e.g., Chen et al., 2024). Offline detection has been explored in various network models, including inhomogeneous Bernoulli networks (e.g., Wang et al., 2021), stochastic block models (e.g., Xu and Lee, 2022; Bhattacharjee et al., 2020) and random dot product graphs

(e.g., Padilla et al., 2022). More recently, Wang et al. (2025) investigated online change point detection in dynamic multilayer random dot product graphs (D-MRDPGs).

In this paper, we study offline change point localization and inference for D-MRDPGs. Specifically, at each time point, we observe a realization of an $L$-layered multilayer network, where nodes are associated with fixed but latent positions, and layer-specific weight matrices capture heterogeneous interactions across layers. These weight matrices are allowed to vary over time. Our goal is to develop efficient procedures for localizing and inferring change points under this dynamic multilayer structure in the offline setting.

## 1.1 List of contributions

The main contributions of this paper are summarized as follows.

First, to the best of our knowledge, this is the first study on offline change point detection in dynamic multilayer networks. We propose a novel two-stage procedure: ($i$) seeded binary segmentation with refined CUSUM statistics to generate a coarse set of candidates, and ($ii$) refinement via low-rank tensor estimation. We establish consistency for both the estimated number of change points and their locations.

Second, we derive the limit distributions of the refined estimators, revealing two distinct regimes depending on whether the jump size is fixed or vanishes as the time horizon grows. To the best of our knowledge, these are the first such results in the network literature. We further provide a data-driven procedure for constructing confidence intervals.

Finally, extensive numerical experiments demonstrate that our methods substantially outperform existing state-of-the-art algorithms.

## 1.2 Notation and organization

For $p \in \mathbb{N}^+$, let $[p] = \{1, \dots, p\}$. For sequences $\{a_n\}_{n \in \mathbb{N}^+}, \{b_n\}_{n \in \mathbb{N}^+} \subset \mathbb{R}^+$, write $a_n = O(b_n)$ if $a_n \leq C b_n$ for some constant $C > 0$ and all sufficiently large $n$, and $a_n = \Theta(b_n)$ if both $a_n = O(b_n)$ and $b_n = O(a_n)$. For a sequence of random variables $\{X_n\}_{n \in \mathbb{N}^+}$, $X_n = O_p(a_n)$ if $\lim_{M \to \infty} \limsup_n \mathbb{P}(|X_n| \geq M a_n) = 0$. For sets $\mathcal{C}$ and $\mathcal{C}'$, define the one-sided Hausdorff distance $d(\mathcal{C}'|\mathcal{C}) = \max_{c \in \mathcal{C}} \min_{c' \in \mathcal{C}'} |c' - c|$, with $d(\mathcal{C}'|\mathcal{C}) = \infty$ if either set is empty.

For $A \in \mathbb{R}^{p_1 \times p_2}$, let $A_i$ and $A^j$ denote its $i$th row and $j$th column, and $\sigma_1(A) \geq \cdots \geq \sigma_{p_1 \wedge p_2}(A) \geq 0$ its singular values. For tensors $\mathbf{M}, \mathbf{Q} \in \mathbb{R}^{p_1 \times p_2 \times p_3}$, define $\langle \mathbf{M}, \mathbf{Q} \rangle = \sum_{i=1}^{p_1} \sum_{j=1}^{p_2} \sum_{l=1}^{p_3} \mathbf{M}_{i,j,l} \mathbf{Q}_{i,j,l}$ and $\|\mathbf{M}\|_{\mathrm{F}}^2 = \langle \mathbf{M}, \mathbf{M} \rangle$. The mode-$s$ matricization of $\mathbf{M}$ is denoted by $\mathcal{M}_s(\mathbf{M})$ with $\mathcal{M}_1(\mathbf{M})_{i_1,(i_2-1)p_3+i_3} = \mathbf{M}_{i_1,i_2,i_3}$ and $\mathcal{M}_s(\mathbf{M}) \in \mathbb{R}^{p_s \times \prod_{t \neq s} p_t}$. Tucker ranks $(r_1, r_2, r_3)$ are given by $r_s = \mathrm{rank}(\mathcal{M}_s(\mathbf{M}))$. For $U_s \in \mathbb{R}^{q_s \times p_s}$, the marginal multiplication operator $\times_1$ is defined as $\mathbf{M} \times_1 U_1 = \{\sum_{k=1}^{p_1} \mathbf{M}_{k,j,l}(U_1)_{i,k}\}_{i \in [q_1], j \in [p_2], l \in [p_3]} \in \mathbb{R}^{q_1 \times p_2 \times p_3}$, with $\times_2$ and $\times_3$ defined analogously.

The paper is organized as follows. Section 2 introduces the D-MRDPG model, the two-stage localization procedure and theoretical guarantees. Section 3 derives limiting distributions of the refined estimators and proposes a data-driven method for confidence intervals. Section 4 presents numerical experiments and Section 5 concludes. Proofs and auxiliary results are in the Appendix.

## 2 Change point localization

### 2.1 Problem formulation

We consider the multilayer random dot product graph (MRDPG) model (Jones and Rubin-Delanchy, 2020), an extension of the random dot product graph (Young and Scheinerman, 2007) to multilayer networks. Each layer is characterized by a distinct weight matrix, while all layers share a common set of latent positions. We focus on undirected edges, noting that the directed case is analogous.

**Definition 1** (Multilayer random dot product graphs, MRDPGs)**.** *Given a sequence of deterministic matrices $\{W_{(l)}\}_{l=1}^L \subset \mathbb{R}^{d \times d}$, let $\{X_i\}_{i=1}^n \subset \mathbb{R}^d$ be fixed vectors satisfying $X_i^\top W_{(l)} X_j \in [0,1]$ for*

*all $i, j \in [n], l \in [L]$. An adjacency tensor $\mathbf{A} \in \{0, 1\}^{n \times n \times L}$ follows an MRDPG if*

$$\mathbb{P}\{\mathbf{A}\} = \prod_{l=1}^{L} \prod_{1 \leq i \leq j \leq n} \mathbf{P}_{i,j,l}^{\mathbf{A}_{i,j,l}} (1 - \mathbf{P}_{i,j,l})^{1 - \mathbf{A}_{i,j,l}}$$

$$= \prod_{l=1}^{L} \prod_{1 \leq i \leq j \leq n} \left( X_i^\top W_{(l)} X_j \right)^{\mathbf{A}_{i,j,l}} \left( 1 - X_i^\top W_{(l)} X_j \right)^{1 - \mathbf{A}_{i,j,l}}.$$

*We write $\mathbf{A} \sim \mathrm{MRDPG}(\{X_i\}_{i=1}^n, \{W_{(l)}\}_{l \in [L]})$ and denote the probability tensor by $\mathbf{P} \in \mathbb{R}^{n \times n \times L}$.*

We now extend this static model to a dynamic setting and introduce a change point framework.

**Definition 2** (Dynamic multilayer random dot product graphs, D-MRDPGs)**.** *Let $\{X_i\}_{i=1}^n \subset \mathbb{R}^d$ be latent positions and $\{W_{(l)}(t)\}_{l \in [L], t \in [T]} \subset \mathbb{R}^{d \times d}$ be a weight matrix sequence. A sequence of mutually independent adjacency tensors $\{\mathbf{A}(t)\}_{t \in [T]}$ follows the dynamic MRDPGs if $\mathbf{A}(t) \sim \mathrm{MRDPG}(\{X_i\}_{i=1}^n, \{W_{(l)}(t)\}_{l \in [L]})$ for $t \in [T]$. We write $\{\mathbf{A}(t)\}_{t=1}^T \sim \mathrm{D\text{-}MRDPGs}(\{X_i\}_{i=1}^n, \{\{W_{(l)}(t)\}_{l \in [L]}\}_{t=1}^T)$ and write $\{\mathbf{P}(t)\}_{t=1}^T$ as the corresponding sequence of probability tensors.*

**Model 1.** *Let $\{\mathbf{A}(t)\}_{t \in [T]} \subset \{0, 1\}^{n \times n \times L}$ follow D-MRDPGs as in Definition 2. $(i)$ Assume that there exist change points $0 = \eta_0 < \eta_1 < \cdots < \eta_K < T = \eta_{K+1}$ such that for $t \in [T-1]$, $\{W_{(l)}(t)\}_{l=1}^L \neq \{W_{(l)}(t+1)\}_{l=1}^L$ if and only if $t \in \{\eta_k\}_{k=1}^K$. Let $\Delta = \min_{k \in [K+1]}(\eta_k - \eta_{k-1})$ be the minimal spacing between two consecutive change points and assume $\Delta = \Theta(T)$. $(ii)$ For each $k \in [K]$, define the $k$-th jump size and normalized jump tensor as $\kappa_k = \|\mathbf{P}(\eta_{k+1}) - \mathbf{P}(\eta_k)\|_{\mathrm{F}}$ and $\Psi_k = \kappa_k^{-1}\{\mathbf{P}(\eta_{k+1}) - \mathbf{P}(\eta_k)\}$, and let $\kappa = \min_{k \in [K]} \kappa_k$ denote the smallest jump magnitude.*

Model 1 allows abrupt changes in layer connectivity (via weight matrices), while keeping latent positions unchanged over time. This framework is motivated by applications such as air transportation networks (Section 4.2), where nodes represent airports with relatively stable intrinsic attributes (e.g. geographical location and logistical capacity). In contrast, airline routing preferences, encoded in the weight matrices, may shift due to route optimization strategies or policy interventions. The framework can be further extended to allow latent positions to change at the change points; see Appendix C for details.

In Model 1($i$), we assume that the minimal spacing $\Delta$ between successive change points scales with the time horizon $T$, essentially bounding the number of changes $K$. This assumption can be relaxed (see Section 5 and Appendix G.1). In Model 1($ii$), the change magnitude is quantified via the Frobenius norm of the difference between expected adjacency tensors. This metric is sufficiently general to accommodate both dense changes - small but widespread deviations across many layers - and sparse changes - large deviations concentrated in a few layers. Throughout, we allow all model parameters, including the number of nodes $n$, number of layers $L$, latent dimension $d$, jump size $\kappa$ and minimal spacing $\Delta$ to diverge with $T$.

## 2.2 CHANGE POINT LOCALIZATION ALGORITHM

In this section, we introduce a two-stage procedure for offline change point localization in dynamic multilayer networks, detailed in Algorithm 1. **Stage I** generates a coarse set of change point candidates using seeded binary segmentation and CUSUM statistics. **Stage II** refines them via localized scan statistics constructed using a tensor-based low-rank estimation technique. This approach builds on Wang et al. (2021) for single-layer networks and extends it to the multilayer setting.

For **Stage I**, we begin by defining the seeded intervals (Kovács et al., 2023) and CUSUM statistics (Page, 1954) for dynamic multilayer networks in Definitions 3 and 4.

**Definition 3** (Seeded intervals)**.** *Let $J = \lceil C_J \log_2(T) \rceil$ for some sufficiently large absolute constant $C_J > 0$. For each $j \in [J]$, define the collection of intervals $\mathcal{J}_j$ as $\mathcal{J}_j = \{(\lfloor (i-1)T2^{-j} \rfloor, \lceil (i-1)T2^{-j} + T2^{-j+1} \rceil] : i \in [2^j - 1]\}$. The full collection of seeded intervals is defined as $\mathcal{J} = \bigcup_{j=1}^J \mathcal{J}_j$.*

**Definition 4** (CUSUM statistics). *Given a tensor sequence $\{\mathbf{B}(t)\}_{t\in[T]}$ and any $0 \le s < t < e \le T$, define the CUSUM statistics as*

$$\widetilde{\mathbf{B}}^{s,e}(t) = \sum_{u=s+1}^{e} \omega_{s,e}^{t}(u)\mathbf{B}(u), \quad where \quad \omega_{s,e}^{t}(u) = \begin{cases} \sqrt{\frac{e-t}{(e-s)(t-s)}}, & for\ u \in [t]\backslash[s], \\ -\sqrt{\frac{t-s}{(e-s)(e-t)}}, & for\ u \in [e]\backslash[t]. \end{cases} \quad (1)$$

**Stage I** implements a modified version of seeded binary segmentation (SBS), a computationally efficient algorithm introduced by Kovács et al. (2023). SBS leverages seeded intervals to construct a multiscale collection of candidate regions for detecting multiple change points. Within each interval, the algorithm computes CUSUM statistics and retains time points where the statistic is maximized and exceeds a predefined threshold, as preliminary change point estimators

We next define the refined scan statistics used in **Stage II**, based on tensor heteroskedastic principal component analysis (TH-PCA), a low-rank tensor estimation method proposed by Han et al. (2022) and detailed in Algorithm 2 in Appendix D.

**Definition 5** (Refined scan statistics). *Let $\{\mathbf{A}'(t)\}_{t\in[T]}$ and $\{\mathbf{B}'(t)\}_{t\in[T]}$ be independent sequences generated according to Definition 2. Given $\{(b_k, s_k, e_k)\}_{k=1}^{\widetilde{K}}$, for any $k \in [\widetilde{K}]$ and $t \in (s_k, e_k)$, we define the refined scan statistic as $\widehat{D}_{b_k}^{s_k,e_k}(t) = \left|\left\langle \widehat{\mathbf{P}}^{s_k,e_k}(b_k)/\|\widehat{\mathbf{P}}^{s_k,e_k}(b_k)\|_{\mathrm{F}}, \widetilde{\mathbf{A}'}^{s_k,e_k}(t)\right\rangle\right|$, where $\widehat{\mathbf{P}}^{s_k,e_k}(b_k) = \mathrm{TH\text{-}PCA}\big(\widetilde{\mathbf{B}'}^{s_k,e_k}(b_k), (d, d, m_{b_k}^{s_k,e_k}), \sqrt{(e_k-b_k)(b_k-s_k)/(e_k-s_k)}, \sqrt{(e_k-b_k)(b_k-s_k)/(e_k-s_k)}\big)$ with TH-PCA detailed in Algorithm 2, $\widetilde{\mathbf{B}'}^{\cdot,\cdot}(\cdot)$ defined in Definition 4 and $m_{b_k}^{s,e}$ defined in Assumption 1(ii).*

**Stage II** refines each preliminary change point estimate from **Stage I** by locating the time point that maximizes the refined scan statistics within a local window around the initial estimate. This step employs the TH-PCA procedure with an additional truncation step (see Algorithm 2 in Appendix D) to more accurately estimate the local expected CUSUM adjacency tensors, yielding provably improved localization accuracy.

The assumption of mutual independence among the four sequences in Algorithm 1 is imposed for theoretical convenience. In practice (and in our numerical experiments in Section 4), **Stage I** and **Stage II** are implemented using the same two split tensor sequences via the odd–even splitting approach. The computational cost is $O(Tn^2 L \log^2(T))$ for Stage I and $O(Tn^2 Lr \log(n))$ for Stage II, where $r$ is the maximum input rank in TH-PCA, giving an overall cost of $O(Tn^2 Lr \log^2(T \vee n))$.

## 2.3 THEORETICAL GUARANTEES

This section establishes the theoretical guarantees of the proposed two-stage change point localization procedure (Algorithm 1). We begin by justifying the use of low-rank tensor estimation via TH-PCA (Algorithm 2) in **Stage II** through an analysis of the expected CUSUM-transformed and average adjacency tensors. While the expected averaged adjacency tensors introduced below are not used in this section, they are essential for deriving the limiting distributions in Section 3.

For any $0 \le s < t < e \le T$, define the expected CUSUM-transformed and average adjacency tensors as

$$\widetilde{\mathbf{P}}^{s,e}(t) = \mathbb{E}\big\{\widetilde{\mathbf{B}}^{s,e}(t)\big\} \quad and \quad \mathbf{P}^{s,e} = \mathbb{E}\big\{\mathbf{B}^{s,e}\big\}, \quad where \quad \mathbf{B}^{s,e} = (e-s)^{-1}\sum_{t=s+1}^{e} \mathbf{B}(t), \quad (2)$$

and $\widetilde{\mathbf{B}}^{\cdot,\cdot}(\cdot)$ is defined in Definition 4. Both tensors admit Tucker representations of the form $\widetilde{\mathbf{P}}^{s,e}(t) = \mathbf{S}\times_1 X \times_2 X \times_3 \widetilde{Q}^{s,e}(t)$, and $\mathbf{P}^{s,e} = \mathbf{S}\times_1 X \times_2 X \times_3 Q^{s,e}$, where $X = (X_1, \ldots, X_n)^{\top} \in \mathbb{R}^{n\times d}$ and $\mathbf{S} \in \mathbb{R}^{d\times d\times d^2}$ with $\mathbf{S}_{i,j,l} = \mathbb{1}\{l = (i-1)d+j\}$. The matrices $\widetilde{Q}^{s,e}(t)$ and $Q^{s,e}$ are given by

$$\widetilde{Q}^{s,e}(t) = \sum_{u=s+1}^{e} \omega_{s,e}^{t}(u)Q(u), \quad Q^{s,e} = (e-s)^{-1}\sum_{t=s+1}^{e} Q(t), \quad (3)$$

where $\omega_{s,e}^{t}(u)$ is define in (1) and $Q(u) \in \mathbb{R}^{L\times d^2}$ with rows

$$\big(Q(u)\big)_l = \big((W_{(l)}(u))_1 \cdots (W_{(l)}(u))_d\big), \quad l \in [L]. \quad (4)$$

---

**Algorithm 1** Two-stage change point localization for D-MRDPGs

---

**INPUT:** Mutually independent sequences $\{\mathbf{A}(t)\}_{t\in[T]}, \{\mathbf{A}'(t)\}_{t\in[T]}, \{\mathbf{B}(t)\}_{t\in[T]}, \{\mathbf{B}'(t)\}_{t\in[T]} \subset \{0,1\}^{n\times n\times L}$, threshold $\tau \in \mathbb{R}^+$, collection of seeded intervals $\mathcal{J}$

**Initialise:** $s \leftarrow 0, e \leftarrow T, \widetilde{\mathcal{C}} \leftarrow \emptyset$

    **Stage I:** Seeded Binary Segmentation, $\text{SBS}\big((s,e),\tau,\mathcal{J}\big)$
    **for** $\mathcal{I} = (\alpha', \beta') \in \mathcal{J}$ **do**
        **if** $\mathcal{I} = (\alpha', \beta') \subseteq (s, e]$ **then**
            $(\alpha, \beta) = (\lfloor \alpha' + 64^{-1}(\beta' - \alpha') \rfloor, \lceil \beta' - 64^{-1}(\beta' - \alpha') \rceil])$
            **if** $\beta - \alpha \geq 2$ **then**
                $b_{\mathcal{I}} \leftarrow \arg\max_{\alpha < t < \beta} \big| \langle \widetilde{\mathbf{A}}^{\alpha,\beta}(t), \widetilde{\mathbf{B}}^{\alpha,\beta}(t) \rangle \big|, \ a_{\mathcal{I}} \leftarrow \big| \langle \widetilde{\mathbf{A}}^{\alpha,\beta}(b_{\mathcal{I}}), \widetilde{\mathbf{B}}^{\alpha,\beta}(b_{\mathcal{I}}) \rangle \big|$
            **else** $a_{\mathcal{I}} \leftarrow -1$
            **end if**
        **else** $a_{\mathcal{I}} \leftarrow -1$
        **end if**
    **end for**
    $\mathcal{I}^* \leftarrow \arg\max_{\mathcal{I}\in\mathcal{J}} a_{\mathcal{I}}$
    **if** $a_{\mathcal{I}^*} > \tau$ **then**
        $\widetilde{\mathcal{C}} \leftarrow \widetilde{\mathcal{C}} \cup \{b_{\mathcal{I}^*}\}, \text{SBS}\big((s, b_{\mathcal{I}^*}), \tau, \mathcal{J}\big), \text{SBS}\big((b_{\mathcal{I}^*}, e), \tau, \mathcal{J}\big)$
    **end if**
    **Stage II:** Local Refinement, $\text{LR}(\widetilde{\mathcal{C}})$
    $\{b_k\}_{k=1}^{\widetilde{K}} \leftarrow \widetilde{\mathcal{C}}$ with $0 = b_0 < b_1 < \cdots < b_{\widetilde{K}} < b_{\widetilde{K}+1} = T$
    **for** $k = 1$ to $\widetilde{K}$ **do**
        $(s_k, e_k] \leftarrow \big(\lfloor (b_{k-1} + b_k)/2 \rfloor, \lceil (b_k + b_{k+1})/2 \rceil \big]$
        $\widetilde{\eta}_k \leftarrow \arg\max_{s_k < t < e_k} \widehat{D}_{b_k}^{s_k, e_k}(t)$               ▷ See Definition 5
    **end for**

**OUTPUT:** $\{\widetilde{\eta}_k\}_{k=1}^{\widetilde{K}}$

---

To establish the low-rank structure of $\widetilde{\mathbf{P}}^{s,e}(t)$ and $\mathbf{P}^{s,e}$ (in terms of Tucker ranks, see Section 1.2), and to state theoretical guarantees for Algorithm 1, we state some necessary assumptions below.

**Assumption 1.** *Consider* D-MRDPGs$(\{X_i\}_{i=1}^n, \{\{W_{(l)}(t)\}_{l\in[L]}\}_{t=1}^T)$ *from Definition 2.*

*(i) Let $X = (X_1, \ldots, X_n)^\top \in \mathbb{R}^{n\times d}$. Assume that $\text{rank}(X) = d$, $\sigma_1(X)/\sigma_d(X) \leq C_\sigma$ and $\sigma_d(X) \geq C_{\text{gap}}\sqrt{n}$ with absolute constants $C_\sigma, C_{\text{gap}} > 0$.*

*(ii) For any $0 \leq s < t < e \leq T$, let $\widetilde{Q}^{s,e}(t) \in \mathbb{R}^{L\times d^2}$ be defined in (3). Denote $m_t^{s,e} = \text{rank}(\widetilde{Q}^{s,e}(t))$. Assume that $\sigma_1\big(\widetilde{Q}^{s,e}(t)\big)/\sigma_{m_t^{s,e}}\big(\widetilde{Q}^{s,e}(t)\big) \leq C_\sigma$ and $\sigma_{m_t^{s,e}}\big(\widetilde{Q}^{s,e}(t)\big) \geq C_{\text{gap}}$ with absolute constants $C_{\text{gap}}, C_\sigma > 0$.*

*(iii) For any $0 \leq s < e \leq T$, let $Q^{s,e} \in \mathbb{R}^{L\times d^2}$ be defined in (3). Denote $m^{s,e} = \text{rank}(Q^{s,e})$. Assume that $\sigma_1\big(Q^{s,e}\big)/\sigma_{m^{s,e}}\big(Q^{s,e}\big) \leq C_\sigma$ and $\sigma_{m^{s,e}}\big(Q^{s,e}\big) \geq C_{\text{gap}}$ with absolute constants $C_{\text{gap}}, C_\sigma > 0$.*

Assumption 1$(i)$ imposes a full-rank condition on the latent position matrix $X$, requiring its smallest singular value to be at least of order $\sqrt{n}$, with all singular values of the same order. Since $X$ represents latent positions rather than observed data, the full-rankness of $X$ can be interpreted as a condition on the knowledge of the intrinsic dimension $d$, ensuring that the input dimension to TH-PCA is no smaller than the true latent dimension $d$. Further discussion on rank selection, see Wang et al. (2025) and Section 4.1.

Assumptions 1$(ii)$ and $(iii)$ - with $(iii)$ for Section 3 - impose low-rank conditions on the CUSUM and averaged forms of $\{Q(t)\}_{t=1}^T$, where each $Q(t)$ comprises the weight matrices $\{W_{(l)}(t)\}_{l=1}^L$. In Appendix E, we show that, with high probability, each working interval $(s_k, e_k]$ or $(\tilde{s}_k, \tilde{e}_k]$ contains exactly one change point $\eta_k$, implying $\max\{m_t^{s_k,e_k}, m^{\tilde{s}_k,\tilde{e}_k}\} \leq \text{rank}(Q(\eta_k)) + \text{rank}(Q(\eta_{k+1}))$ for $t \in (s_k, e_k)$. This implicitly constraints the ranks of $\{Q(\eta_k)\}_{k=1}^{K+1}$. While this low-rank structure

may not directly or transparently reflect the explicit model structure, such ambiguity is common in tensor-based models (e.g. Jing et al., 2021).

The signal-to-noise ratio (SNR) is commonly used to characterize the inherent difficulty of change point detection. We now state the SNR condition required for our theoretical guarantees.

**Assumption 2** (Signal-to-noise ratio condition). *Assume that there exists a large enough absolute constant $C_{\mathrm{SNR}} > 0$ such that $\kappa\sqrt{\Delta} \geq C_{\mathrm{SNR}} \log(T)\sqrt{nL^{1/2} + d^2 m_{\max} + nd + Lm_{\max}}$, where $m_{\max} = \max_{k\in[K+1]} \mathrm{rank}\big(Q(\eta_k)\big)$ with $Q(\eta_k)$ defined in (4).*

We compare Assumption 2 to its counterpart in Wang et al. (2021). When the sparsity parameter $\rho = 1$, their SNR condition (Assumption 3) becomes $\kappa\sqrt{\Delta} \geq C_{\mathrm{SNR}} \log^{1+\xi}(T)\sqrt{nd}$ for some $\xi > 0$. Our assumption is consistent with this and extends it to the multilayer setting by accounting for the additional complexity from multilayers and the low-rank structure of layers' weight matrices.

**Theorem 1.** *Let $\{\widetilde{\eta}_k\}_{k=1}^{\widetilde{K}}$ be the output of Algorithm 1. Suppose the mutually independent adjacency tensor sequences $\{\mathbf{A}(t)\}_{t\in[T]}, \{\mathbf{A}'(t)\}_{t\in[T]}, \{\mathbf{B}(t)\}_{t\in[T]}, \{\mathbf{B}'(t)\}_{t\in[T]} \subset \{0,1\}^{n\times n\times L}$ are generated according to Definition 2 and satisfy Model 1, Assumptions 1(i), (ii) and 2. Assume the threshold $\tau$ is chosen such that $c_{\tau,1}n\sqrt{L}\log^{3/2}(T) < \tau < c_{\tau,2}\kappa^2\Delta$, where $c_{\tau,1}, c_{\tau,2} > 0$ are sufficiently large and small absolute constants, respectively. We have that*

$$\mathbb{P}\Big\{\widetilde{K} = K \text{ and } |\widetilde{\eta}_k - \eta_k| \leq \epsilon_k, \forall k \in [K]\Big\} \geq 1 - CT^{-c}, \quad \text{where } \epsilon_k = C_\epsilon \frac{\log(T)}{\kappa_k^2},$$

*and $C_\epsilon, C, c > 0$ are absolute constants.*

Theorem 1 implies that, with probability tending to 1 as $T \to \infty$, the estimated number of change points satisfies $\widetilde{K} = K$ and the relative localization error vanishes: $\max_{k\in[K]} \Delta^{-1}|\widetilde{\eta}_k - \eta_k| \leq C_\epsilon \Delta^{-1}\kappa^{-2}\log(T) \to 0$ by Assumption 2. This establishes the consistency of Algorithm 1 in both detecting and localizing all change points.

**Remark 1.** *Compared to Wang et al. (2021), which established minimax-optimal localization rates for single-layer networks, our work extends these guarantees to more complex multilayer settings without sacrificing accuracy. In contrast, Wang et al. (2025) focused on the online setting and obtained a localization rate of order $\kappa^{-2}(d^2 m_{\max} + nd + Lm_{\max})\log(\Delta/\alpha)$, where $\alpha$ controls the Type-I error rate. Our approach, by comparison, achieves a substantially sharper rate of order $\kappa_k^{-2}\log(T)$.*

## 3 LIMITING DISTRIBUTIONS

Inference on change points is generally more challenging than establishing high-probability bounds on localization errors. To address this, we introduce a final refinement step, inspired by approaches such as those in Madrid Padilla et al. (2023); Xue et al. (2024); Xu et al. (2024).

Let $\{\mathbf{A}(t)\}_{t\in[T]}$ and $\{\mathbf{B}(t)\}_{t\in[T]}$ be independent samples as defined in Definition 2. Let $\{\widetilde{\eta}_k\}_{k=1}^{K}$ be the output of Algorithm 1 with $0 = \widetilde{\eta}_0 < \widetilde{\eta}_1 < \cdots < \widetilde{\eta}_{\widetilde{K}} < \widetilde{\eta}_{\widetilde{K}+1} = T$. For each $k \in [\widetilde{K}]$, define the final estimators as

$$\widehat{\eta}_k = \underset{\tilde{s}_k < t < \tilde{e}_k}{\arg\min} \, \mathcal{Q}_k(t) = \underset{\tilde{s}_k < t < \tilde{e}_k}{\arg\min} \sum_{u=\tilde{s}_k+1}^{t} \|\mathbf{A}(u) - \widehat{\mathbf{P}}^{\widetilde{\eta}_{k-1},\widetilde{\eta}_k}\|_{\mathrm{F}}^2 + \sum_{u=t+1}^{\tilde{e}_k} \|\mathbf{A}(u) - \widehat{\mathbf{P}}^{\widetilde{\eta}_k,\widetilde{\eta}_{k+1}}\|_{\mathrm{F}}^2, \quad (5)$$

where $(\tilde{s}_k, \tilde{e}_k] = ((\widetilde{\eta}_{k-1} + \widetilde{\eta}_k)/2, (\widetilde{\eta}_k + \widetilde{\eta}_{k+1})/2]$ and

$$\widehat{\mathbf{P}}^{\widetilde{\eta}_{k-1},\widetilde{\eta}_k} = \text{TH-PCA}(\mathbf{B}^{\widetilde{\eta}_{k-1},\widetilde{\eta}_k}, (d, d, m^{\widetilde{\eta}_{k-1},\widetilde{\eta}_k}), 1, 0), \quad (6)$$

with TH-PCA detailed in Algorithm 2, $\mathbf{B}^{\cdot,\cdot}$ defined in (2) and $m_{\widetilde{\eta}^{k-1},\widetilde{\eta}_k}$ defined in Assumption 1(iii).

**Theorem 2.** *Let $\{\mathbf{A}(t)\}_{t\in[T]}, \{\mathbf{A}'(t)\}_{t\in[T]}, \{\mathbf{B}(t)\}_{t\in[T]}, \{\mathbf{B}'(t)\}_{t\in[T]} \subset \{0,1\}^{n\times n\times L}$ be mutually independent adjacency tensor sequences generated according to Definition 2 and satisfying Model 1, Assumptions 1 and 2. Let $\{\widehat{\eta}_k\}_{k=1}^{\widetilde{K}}$ be defined in (5) with $\{\widetilde{\eta}_k\}_{k=1}^{\widetilde{K}}$ obtained from Algorithm 1, using a threshold $\tau$ satisfying condition stated in Theorem 1.*

*For $k \in [K]$, if $\kappa_k \to 0$, as $T \to \infty$, then when $T \to \infty$, we have $|\widehat{\eta}_k - \eta_k| = O_p(\kappa_k^{-2})$ and*

$$\kappa_k^2(\widehat{\eta}_k - \eta_k) \xrightarrow{\mathcal{D}} \arg\min_{r \in \mathbb{R}} \mathcal{P}'_k(r), \quad where \quad \mathcal{P}'_k(r) = \begin{cases} -r + 2\sigma_{k,k}\mathbb{B}_1(-r), & r < 0, \\ 0, & r = 0, \\ r + 2\sigma_{k,k+1}\mathbb{B}_2(r), & r > 0, \end{cases}$$

*for $r \in \mathbb{Z}$. Here, $\mathbb{B}_1(r)$ and $\mathbb{B}_2(r)$ are independent standard Brownian motions, and for any $k' \in \{k, k+1\}$, $\sigma_{k,k'}^2 = \mathrm{Var}\left(\langle \boldsymbol{\Psi}_k, \mathbf{E}_{k'}(1) \rangle\right)$, where $\boldsymbol{\Psi}_k$ is the normalized jump tensor (Model 1), and $\mathbf{E}_{k'}(t) = \mathbf{A}_{k'}(t) - \mathbf{P}(\eta_{k'})$ with $\{\mathbf{A}_{k'}(t)\}_{t \in \mathbb{Z}} \overset{\text{i.i.d.}}{\sim} \mathrm{MRDPG}(\{X_i\}_{i=1}^n, \{W_{(l)}(\eta_{k'})\}_{l \in [L]})$.*

Theorem 2 establishes the localization error bounds and limiting distributions for the refined change point estimators in the vanishing jump regime ($\kappa_k \to 0$). In particular, it shows the uniform tightness $\kappa_k^2 |\widehat{\eta}_k - \eta_k| = O_p(1)$, which improves upon Theorem 1 by a logarithmic factor and guarantees the existence of limiting distributions. To the best of our knowledge, Theorem 2 is the first to derive limiting distributions for change point estimators in network data. These limiting distributions are associated with a two-sided Brownian motion. Results for the non-vanishing jump regime ($\kappa_k \to \rho_k > 0$) are deferred to Appendix A.

### 3.1 CONFIDENCE INTERVAL CONSTRUCTION

Using Theorem 2, we construct data-driven $(1 - \alpha)$ confidence intervals for $\eta_k$, $k \in [K]$, in the vanishing regime, for a user-specified confidence level $\alpha \in (0, 1)$ as follows.

**Step 1: Estimate the jump size and normalized jump tensor.** Compute the estimated jump size $\hat{\kappa}_k = \|\widehat{\mathbf{P}}^{\widetilde{\eta}_k, \widetilde{\eta}_{k+1}} - \widehat{\mathbf{P}}^{\widetilde{\eta}_{k-1}, \widetilde{\eta}_k}\|_{\mathrm{F}}$ and the estimated normalized jump tensor $\widehat{\boldsymbol{\Psi}}_k = \hat{\kappa}_k^{-1}(\widehat{\mathbf{P}}^{\widetilde{\eta}_k, \widetilde{\eta}_{k+1}} - \widehat{\mathbf{P}}^{\widetilde{\eta}_{k-1}, \widetilde{\eta}_k})$ where $\widehat{\mathbf{P}}^{\cdot, \cdot}$ is defined in (6).

**Step 2: Estimate the variances.** For each $k' \in \{k, k+1\}$, compute

$$\hat{\sigma}_{k,k'}^2 = \frac{1}{\widetilde{\eta}_{k'} - \widetilde{\eta}_{k'-1} - 1} \sum_{t=\widetilde{\eta}_{k'-1}+1}^{\widetilde{\eta}_{k'}} \left(\langle \widehat{\boldsymbol{\Psi}}_k, \mathbf{A}(t) - \widehat{\mathbf{P}}^{\widetilde{\eta}_{k'-1}, \widetilde{\eta}_{k'}} \rangle\right)^2.$$

**Step 3: Simulate limiting distributions.** Let $B \in \mathbb{N}^+$ and $M \in \mathbb{R}^+$. For each $b \in [B]$, let

$$\hat{u}_k^{(b)} = \arg\min_{r \in (-M, M)} \widehat{\mathcal{P}}'_k(r), \quad where \quad \widehat{\mathcal{P}}'_k(r) = \begin{cases} -r + \frac{2\hat{\sigma}_{k,k}}{\sqrt{T}}\sum_{i=\lceil Tr \rceil}^{-1} z_i^{(b)}, & r < 0, \\ 0, & r = 0, \\ r + \frac{2\hat{\sigma}_{k,k+1}}{\sqrt{T}}\sum_{i=1}^{\lfloor Tr \rfloor} z_i^{(b)}, & r > 0, \end{cases}$$

with independent standard Gaussian random variables $\{z_i^{(b)}\}_{i=-\lfloor TM \rfloor}^{\lceil TM \rceil}$.

**Step 4: Construct the confidence interval.** Let $\hat{q}_{\alpha/2}, \hat{q}_{1-\alpha/2}$ be empirical quantiles of $\{\hat{u}_k^{(b)}\}_{b=1}^B$. The $(1 - \alpha)$ confidence interval for $\eta_k$ is given by

$$\left[\widehat{\eta}_k - \frac{\hat{q}_{1-\alpha/2}}{\hat{\kappa}_k^2}\mathbb{1}\{\hat{\kappa}_k \neq 0\}, \widehat{\eta}_k - \frac{\hat{q}_{\alpha/2}}{\hat{\kappa}_k^2}\mathbb{1}\{\hat{\kappa}_k \neq 0\}\right].$$

The empirical performance of this procedure is evaluated in Section 4.1.

## 4 NUMERICAL EXPERIMENTS

### 4.1 SIMULATION STUDIES

To evaluate the performance of our method (Algorithm 1) for change point detection and localization, we compare it to gSeg (Chen and Zhang, 2015) and kerSeg (Song and Chen, 2024). For the competitors, we consider two input types: networks (nets.) and their layer-wise Frobenius norms (frob.). For gSeg, we construct the similarity graph using the minimum spanning tree and apply the original edge-count scan statistics. For kerSeg, we use the kernel-based scan statistics fGKCP$_1$.

For both methods, we set the significance level $\alpha = 0.05$. Our proposed method is referred to as CPDmrdpg. Following Wang et al. (2025), we use relatively large Tucker ranks as inputs to TH-PCA (Algorithm 2) for robustness, setting $r_1 = r_2 = 15$ and $r_3 = L$ to compute the refined scan statistics (Definition 5). Based on Theorem 1, we set the threshold $\tau = c_{\tau,1} n \sqrt{L} \log^{3/2}(T)$ with $c_{\tau,1} = 0.1$. We also assess the confidence intervals constructed utilizing the procedure in Section 3.1, a capability not supported by the competitors. We set $B = 500$ and $M = T$ as suggested by Xu et al. (2024).

To assess sensitivity to tuning parameters, we vary the threshold constant $c_{\tau,1} \in \{0.05, 0.08, 0.10, 0.12, 0.15, 0.20, 0.25\}$ and input ranks $r \in \{10, 15, 20\}$. We further conduct additional simulations to evaluate the robustness of our method under temporal dependence, high-frequency change points and randomly located changes. In addition, we compare our approach with existing dynamic multilayer network approaches (Wang et al., 2025), which are designed for online settings, as well as with deep-learning-based approaches (Li et al., 2024). All results are reported in Appendix G.1.

Performance is quantified using the following metrics: $(i)$ Absolute error: $|\widehat{K} - K|$ where $\widehat{K}$ and $K$ denote the numbers of estimated and true change points, respectively; $(ii)$ One-sided Hausdorff distances (see Section 1.2): $d(\widehat{\mathcal{C}}|\mathcal{C})$ and $d(\mathcal{C}|\widehat{\mathcal{C}})$ where $\widehat{\mathcal{C}}$ and $\mathcal{C}$ denote the sets of estimated and true change points, respectively; $(iii)$ Time segment coverage: $C(\mathcal{G}, \mathcal{G}') = T^{-1} \sum_{\mathcal{A} \in \mathcal{G}} |\mathcal{A}| \cdot \max_{\mathcal{A}' \in \mathcal{G}'} |\mathcal{A} \cap \mathcal{A}'| / |\mathcal{A} \cup \mathcal{A}'|$ where $\mathcal{G}$ and $\mathcal{G}'$ denote the partitions of the time span $[1, T]$ into intervals between consecutive true and estimated change points, respectively.

Throughout, we set the time horizon to $T = 200$ and the number of layers to $L = 4$, and consider node sizes $n \in \{50, 100\}$. Each setting is evaluated over 100 Monte Carlo trials. We consider two network models: the Dirichlet distribution model (DDM) and the multilayer stochastic block model (MSBM), with structural changes specified in each scenario. In the DDM, we generate latent positions $\{X_i\}_{i=1}^n \cup \{Y_i\}_{i=1}^n \overset{\text{i.i.d.}}{\sim} \text{Dirichlet}(\mathbf{1}_d)$ with $d = 5$ and $\mathbf{1}_d \in \mathbb{R}^d$ denoting the all-one vector. For each time $t$, we sample weight matrices $\{W_{(l)}(t)\}_{l=1}^L \subset \mathbb{R}^{d \times d}$ with entries $(W_{(l)}(t))_{u,v} \sim \text{Uniform}((\rho_t L + l)/(4L), (\rho_t L + l + 1)/(4L))$. The edge probabilities are given by $\mathbf{P}_{i,j,l}(t) = X_i^\top W_{(l)}(t) Y_j$ and the adjacency entries are sampled as $\mathbf{A}_{i,j,l}(t) \sim \text{Bernoulli}(\mathbf{P}_{i,j,l}(t))$. In the MSBM, the edge probability tensor $\mathbf{P}_{i,j,l}(t) \in [0,1]^{n \times n \times L}$ is defined as $\mathbf{P}_{i,j,l}(t) = p_{1,l}$ if nodes $i, j \in \mathcal{B}_c$ for some $c \in [C_t]$, and $p_{2,l}$ otherwise, where $\{\mathcal{B}_c\}_{c \in [C_t]}$ partitions the nodes into $C_t$ communities. The connection probabilities are drawn from $p_{1,l} \sim \text{Uniform}((3L+l-1)/(4L), (3L+l)/(4L))$ and $p_{2,l} \sim \text{Uniform}((2L + l - 1)/(4L), (2L + l)/(4L))$. The adjacency tensor $\mathbf{A}(t) \in \{0, 1\}^{n \times n \times L}$ is then sampled $\mathbf{A}_{i,j,l}(t) \overset{\text{ind.}}{\sim} \text{Bernoulli}(\mathbf{P}_{i,j,l}(t))$.

**Scenario 1.** We consider the DDM with $K = 2$ change points at $t \in \{70, 140\}$, yielding 3 time segments $\{\mathcal{A}_i\}_{i=1}^3$. We set $\rho_t = 2$ for $t \in \mathcal{A}_1 \cup \mathcal{A}_3$, and $\rho_t = 3$ with reversed layer order for $t \in \mathcal{A}_2$.

**Scenario 2.** We consider the MSBM with $K = 5$ change points at $t \in \{20, 60, 80, 160, 180\}$, resulting in 6 time segments $\{\mathcal{A}_i\}_{i=1}^6$. We let $\{\mathcal{B}_c(t)\}_{c \in [C_t]}$ be evenly-sized communities and specify the changes as follows: $C_t = 4$ for $t \in \mathcal{A}_1$, $C_t = 2$ for $t \in \mathcal{A}_2$, $C_t = 4$ for $t \in \mathcal{A}_3$, $C_t = 4$ with reversed layer order for $t \in \mathcal{A}_4$, $C_t = 3$ for $t \in \mathcal{A}_5$ and $C_t = 4$ for $t \in \mathcal{A}_6$.

**Scenario 3.** We consider the MSBM with $K = 3$ change points at $t \in \{50, 100, 150\}$, yielding 4 time segments $\{\mathcal{A}_i\}_{i=1}^4$. The number of communities is fixed at $C_t = 3$ but in the first layer, the the community sizes vary across segments $(0.3n, 0.4n, 0.3n)$ in $\mathcal{A}_1 \cup \mathcal{A}_4$, $(0.4n, 0.3n, 0.3n)$ in $\mathcal{A}_2$ and $(0.5n, 0.3n, 0.2n)$ in $\mathcal{A}_3$. The remaining layers retain equal-sized communities.

**Scenario 4.** We consider the MSBM with $K = 5$ change points at $t \in \{20, 60, 80, 160, 180\}$, resulting in 6 time segments $\{\mathcal{A}_i\}_{i=1}^6$. The number of communities is fixed at $C_t = 4$ with equal-sized partitions, while the connection probabilities vary across segments. Specifically, for $\epsilon = 0.1$, we let $p_{1,l} \sim \text{Uniform}(0.5 \cdot [0.21 + \delta_t \cdot \epsilon], 0.5 \cdot [0.25 + \delta_t \cdot \epsilon])$ and $p_{2,l} \sim \text{Uniform}(0.21 + \delta_t \cdot \epsilon, 0.25 + \delta_t \cdot \epsilon)$, where $\delta_t = 0$ for $t \in \mathcal{A}_1 \cup \mathcal{A}_5$, $\delta_t = 1$ for $t \in \mathcal{A}_2 \cup \mathcal{A}_4 \cup \mathcal{A}_6$ and $\delta_t = 2$ for $t \in \mathcal{A}_3$.

The changes in **Scenarios 1** and **4** follow Model 1, while those in **Scenarios 2** and **3** do not, allowing us to assess the robustness of our methods. Table 1 presents results all four scenarios. Across most scenarios, our method demonstrates the strongest overall performance, nearly accurately estimating both the number and locations of change points, and remaining robust even when Model 1

Table 1: Means of evaluation metrics for Scenarios 1–4.

| | | | $n=50$ | | | | $n=100$ | | |
|---|---|---|---|---|---|---|---|---|---|
| Scenario | Method | $|\widehat{K}-K|\downarrow$ | $d(\widehat{\mathcal{C}}|\mathcal{C})\downarrow$ | $d(\mathcal{C}|\widehat{\mathcal{C}})\downarrow$ | $C(\mathcal{G},\mathcal{G}')\uparrow$ | $|\widehat{K}-K|\downarrow$ | $d(\widehat{\mathcal{C}}|\mathcal{C})\downarrow$ | $d(\mathcal{C}|\widehat{\mathcal{C}})\downarrow$ | $C(\mathcal{G},\mathcal{G}')\uparrow$ |
| | CPDmrdpg | 0.01 | 0.00 | 0.42 | 99.86% | 0.00 | 0.00 | 0.00 | 100% |
| | gSeg (nets.) | 1.09 | Inf | Inf | 52.82% | 1.12 | Inf | Inf | 52.62% |
| 1 | kerSeg (nets.) | 0.10 | 0.00 | 3.12 | 99.13% | 0.12 | 0.00 | 2.82 | 99.17% |
| | gSeg (frob.) | 0.52 | Inf | Inf | 90.12% | 0.47 | Inf | Inf | 88.71% |
| | kerSeg (frob.) | 0.26 | 0.00 | 5.76 | 98.35% | 0.30 | 0.00 | 6.07 | 98.11% |
| | CPDmrdpg | 0.00 | 0.00 | 0.00 | 100% | 0.00 | 0.00 | 0.00 | 100% |
| | gSeg (nets.) | 1.60 | Inf | Inf | 67.68% | 1.58 | Inf | Inf | 69.24% |
| 2 | kerSeg (nets.) | 0.15 | 0.00 | 1.53 | 99.32% | 0.16 | 0.00 | 1.81 | 99.31% |
| | gSeg (frob.) | 0.23 | Inf | Inf | 97.71% | 0.16 | 0.04 | 1.65 | 99.17% |
| | kerSeg (frob.) | 0.35 | 0.11 | 3.43 | 98.37% | 0.40 | 0.02 | 4.42 | 97.81% |
| | CPDmrdpg | 0.19 | 9.64 | 0.14 | 95.11% | 0.00 | 0.02 | 0.02 | 99.98% |
| | gSeg (nets.) | 0.98 | Inf | Inf | 68.93% | 0.69 | Inf | Inf | 80.10% |
| 3 | kerSeg (nets.) | 0.16 | 0.18 | 2.06 | 98.90% | 0.17 | 0.00 | 3.26 | 99.16% |
| | gSeg (frob.) | 0.92 | Inf | Inf | 66.78% | 0.79 | Inf | Inf | 72.11% |
| | kerSeg (frob.) | 0.82 | 48.52 | 5.11 | 73.55% | 0.79 | 48.82 | 4.75 | 73.80% |
| | CPDmrdpg | 0.00 | 0.02 | 0.02 | 99.98% | 0.00 | 0.00 | 0.00 | 100% |
| | gSeg (nets.) | 5.00 | Inf | Inf | 0.00% | 4.98 | Inf | Inf | 0.77% |
| 4 | kerSeg (nets.) | 0.36 | 0.14 | 2.65 | 98.56% | 0.34 | 0.08 | 2.93 | 98.47% |
| | gSeg (frob.) | 1.53 | Inf | Inf | 74.92% | 1.86 | Inf | Inf | 68.57% |
| | kerSeg (frob.) | 0.40 | 0.05 | 3.71 | 98.12% | 0.42 | 0.06 | 2.93 | 98.63% |

Table 2: The 95% confidence interval coverage (average length) for change points across all scenarios.

| $n$ | Scenario 1 | Scenario 2 | Scenario 3 | Scenario 4 |
|---|---|---|---|---|
| 100 | 100% (0.003) | 100% (0.106) | 76.67% (1.528) | 100% (0.605) |
| 150 | 100% (0.001) | 100% (0.029) | 95.33% (0.653) | 100% (0.294) |

is violated. For gSeg, Frobenius norm (frob.) inputs yield better results than networks (nets.), while kerSeg performs better with networks, benefiting from its high-dimensional kernel-based design. Although both competitors exhibit low Hausdorff distances $d(\widehat{\mathcal{C}}|\mathcal{C})$, their higher reverse distances $d(\mathcal{C}|\widehat{\mathcal{C}})$ and frequent errors in estimating the number of change points suggest they often detect spurious change points.

Table 2 reports the coverage and average lengths of the confidence intervals constructed via the procedure in Section 3.1 for node size $n \in \{100, 150\}$. The proposed method generally achieves strong coverage with reasonably narrow intervals. Coverage is lower in **Scenario 3**, where violations of Model 1 and relatively small, layer-specific changes pose greater challenges. The performance improves with larger $n$ as the change magnitudes $\kappa_k$ increase.

## 4.2 REAL DATA EXPERIMENTS

Our analysis incorporates two real data sets, the worldwide agricultural trade network data set presented here and the U.S. air transport network data set in Appendix G.2.

**The worldwide agricultural trade network data** are available from Food and Agricultural Organization of the United Nations (2022). The dataset comprises annual multilayer networks from 1986 to 2020 ($T = 35$), with nodes representing countries and layers representing agricultural products. A directed edge within a layer indicates the trade relation between two countries of a specific agricultural product. We use the top $L = 4$ agricultural products by the trade volume and the $n = 75$ most active countries based on import/export volume. Tuning parameters follow the setup described in Section 4.1.

Table 3 summarizes the change points detected by the proposed and competing methods for the worldwide agricultural trade network data. Notably, the gSeg method fails to detect any change points after 2010, regardless of input type. Meanwhile, the kerSeg method detects change points in 1990 and 1992, which are temporally too close. In contrast, our proposed method (CPDmrdpg) identifies four major change points (1991, 1999, 2005, and 2013) that align well with known geopolitical and policy-related events. Furthermore, confidence intervals, which may be constructed for the pro-

Table 3: Detected change points for the worldwide agricultural trade network data.

| Method | Detected change points |
|---|---|
| CPDmrdpg | 1991, 1999, 2005, 2013 |
| gSeg (nets.) | 1993, 2002, 2010 |
| kerSeg (nets.) | 1990, 1992, 1999, 2005, 2012 |
| gSeg (frob.) | 1993, 2002, 2009 |
| kerSeg (frob.) | 1990, 1992, 1997, 2003, 2012 |

Table 4: Detected change point from Algorithm 1 and $95\%$ confidence intervals via Section 3.1 for the worldwide agricultural trade network data.

| Detected change points | Time point | Confidence interval |
|---|---|---|
| 1991 | 6 | $(5.97, 6.03)$ |
| 1999 | 14 | $(13.98, 14.02)$ |
| 2005 | 20 | $(17.97, 18.05)$ |
| 2013 | 28 | $(25.99, 26.06)$ |

posed method via the procedure from Section 3.1, are presented in Table 4 at a $95\%$ confidence level.

The 1991 change point aligns with the German reunification and the dissolution of the Soviet Union, both of which triggered major political shifts that significantly affected the trade dynamics. The 1999 change point corresponds to the World Trade Organization's (WTO) Third Ministerial Conference, a key moment in debates on globalization, particularly regarding agricultural subsidies and tariff reductions, with developing nations demanding fairer trade terms. The 2005 change point marks a WTO agreement to eliminate agricultural export subsidies, promoting greater equity in global markets. Finally, the 2013 change point corresponds to the adoption of the WTO's Bali Package, the first fully endorsed multilateral agreement, which introduced the Trade Facilitation Agreement and key provisions on food security and tariff quota administration, significantly impacting agricultural trade.

## 5 CONCLUSION

In this paper, we study offline change point localization and inference in dynamic multilayer networks — a setting that, to the best of our knowledge, has not been previously addressed. We propose a two-stage algorithm with consistency guarantees for estimating both the number and locations of change points. We further develop local refinement procedures, derive limiting distributions and introduce a data-driven method for constructing confidence intervals for the true change points.

The current framework assumes temporal independence, but it can be extended to incorporate temporal dependence structures (e.g. Padilla et al., 2022; Cho and Owens, 2023); see Appendix B for details on the framework and corresponding adjustments to the theoretical analysis.

Several limitations of this work remain open for future research. First, the assumption $\Delta = \Theta(T)$ precludes frequent change points. This could be relaxed using alternative selection strategies such as the narrowest-over-threshold approach (Baranowski et al., 2019) instead of the greedy selection in this paper. Second, our inference procedure is limited to vanishing jumps. It would be interesting to explore practical procedures for the non-vanishing regime, potentially building on bootstrap methods (e.g. Cho and Kirch, 2022).

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

APPENDIX

All technical details are deferred to the Appendix. Appendix A establishes the limiting distributions in the non-vanishing regime, while Appendix B discusses the extension to incorporate temporal dependence and Appendix C presents the extension to allow changes in latent positions. Additional algorithms used in our procedures are provided in Appendix D. The proof of Theorem 1 is given in Appendix E, while the proofs of the limiting distribution results, including Theorem 2 in the main text and Theorem 3 in Appendix A are presented in Appendix F. Further details and results for Section 4 are collected in Appendix G.

## A   LIMITING DISTRIBUTIONS IN THE NON-VANISHING REGIME

**Theorem 3.** *Let* $\{\mathbf{A}(t)\}_{t\in[T]}, \{\mathbf{A}'(t)\}_{t\in[T]}, \{\mathbf{B}(t)\}_{t\in[T]}, \{\mathbf{B}'(t)\}_{t\in[T]} \subset \{0,1\}^{n\times n\times L}$ *be mutually independent adjacency tensor sequences generated according to Definition 2 and satisfying Model 1, Assumptions 1 and 2. Let* $\{\widehat{\eta}_k\}_{k=1}^{\widetilde{K}}$ *be defined in (5) with* $\{\widetilde{\eta}_k\}_{k=1}^{\widetilde{K}}$ *obtained from Algorithm 1, using a threshold* $\tau$ *satisfying condition stated in Theorem 1.*

*For* $k \in [K]$, *if* $\kappa_k \to \rho_k$, *as* $T \to \infty$, *with* $\rho_k > 0$ *being an absolute constant, then when* $T \to \infty$, *we have* $|\widehat{\eta}_k - \eta_k| = O_p(1)$ *and*

$$\widehat{\eta}_k - \eta_k \xrightarrow{\mathcal{D}} \arg\min_{r\in\mathbb{Z}} \mathcal{P}_k(r), \quad where \quad \mathcal{P}_k(r) = \begin{cases} -r\rho_k^2 - 2\rho_k \sum_{t=r+1}^{0} \langle \boldsymbol{\Psi}_k, \mathbf{E}_k(t) \rangle, & r < 0, \\ 0, & r = 0, \\ r\rho_k^2 + 2\rho_k \sum_{t=1}^{r} \langle \boldsymbol{\Psi}_k, \mathbf{E}_{k+1}(t) \rangle, & r > 0, \end{cases}$$

*for* $r \in \mathbb{Z}$. *Here, the normalized jump tensor* $\boldsymbol{\Psi}_k$ *is defined in Model 1, and for any* $k \in [K+1]$ *and* $t \in \mathbb{Z}$, $\mathbf{E}_k(t) = \mathbf{A}_k(t) - \mathbf{P}(\eta_k)$ *with* $\{\mathbf{A}_k(t)\}_{t\in\mathbb{Z}} \overset{\text{i.i.d.}}{\sim} \text{MRDPG}(\{X_i\}_{i=1}^n, \{W_{(l)}(\eta_k)\}_{l\in[L]})$.

The proof of Theorem 3 is given in Appendix F.

Similar to Theorem 2, Theorem 3 establishes the uniform tightness $\kappa_k^2 |\widehat{\eta}_k - \eta_k| = O_p(1)$ and further derives the limiting distributions of the refined change point estimators defined in (5), which are associated with a two-sided random walk.

## B   EXTENSION TO TEMPORAL DEPENDENCE

To incorporate temporal dependence, we modify the data-generating process in Definition 2 by introducing exponentially decaying correlations governed by a parameter $\pi \in [0, 1]$. Specifically, at time points $t \in \{\eta_k + 1\}_{k=1}^{K}$, adjacency tensors are sampled independently: for any $1 \le i \le j \le n$ and $l \in [L]$,
$$\mathbf{A}_{i,j,l}(t) \sim \text{Bernoulli}(\mathbf{P}_{i,j,l}(t)),$$
where $\mathbf{P}_{i,j,l}(t) = X_i^\top W_{(l)}(t) X_j$. For $k \in [K]$, $t \in [\eta_{k+1}] \backslash [\eta_k + 1]$, $1 \le i \le j \le n$ and $l \in [L]$, edges evolve as follows:

$$\mathbf{A}_{i,j,l}(t+1) \begin{cases} = \mathbf{A}_{i,j,l}(t), & \text{with probability } \pi, \\ \sim \text{Bernoulli}(\mathbf{P}_{i,j,l}(\eta_k + 1)), & \text{with probability } 1 - \pi. \end{cases}$$

When $\pi = 0$, the framework reduces to the independent case. For $\pi > 0$, the adjacency tensors exhibit dependence across time, requiring modifications to the theoretical analysis.

**Adjustments to theoretical analysis.** Temporal dependence introduces correlations across time, which invalidate the standard concentration inequalities used in our original analysis. To address this, we adapt our results using Lemma 14 from Padilla et al. (2022), which extends Bernstein's inequality to account for weak temporal dependence.

Two main components of our theoretical framework require modification:

**Proof of Proposition 4:** We redefine the event

$$\mathcal{A}(s,t,e) = \Big\{ \big| \langle \widetilde{\mathbf{A}}^{s,e}(t), \widetilde{\mathbf{B}}^{s,e}(t) \rangle - \|\widetilde{\mathbf{P}}^{s,e}(t)\|_{\text{F}}^2 \big|$$
$$\le C_{\mathcal{A}} \log(T) \big( \|\widetilde{\mathbf{P}}^{s,e}(t)\|_{\text{F}} + \sqrt{(1-\pi)^{-1} \log(T)} n\sqrt{L} \big) \Big\}.$$

By modifying the proof of Lemma S.4 in Wang et al. (2021) and applying Lemma 14 in Padilla et al. (2022), we obtain that $\mathbb{P}(\mathcal{A}(s,t,e)^c) \leq C_1 T^{-c_1}$.

**Proof of Theorem 1:** By revising the proof of Theorem 4 in Wang et al. (2025) and applying Lemma 14 in Padilla et al. (2022), together with Assumption 1 $(i)$ and $(ii)$ and Lemma 5 in Wang et al. (2025), for any $t \in (s,e)$, we have that

$$\mathbb{P}\left\{\left\|\widehat{\mathbf{P}}^{s_k,e_k}(t) - \widetilde{\mathbf{P}}^{s_k,e_k}(t)\right\|_F \leq C_1\sqrt{(d^2 m_t^{s,e} + nd + Lm_t^{s,e})(1-\pi)^{-1}\log(T)}\right\} \geq 1 - T^{-c_1}.$$

Then by revising the proof of Lemma 5 to account for temporal dependence via Lemma 14 in Padilla et al. (2022), and for any $\varepsilon > 0$, we establish that

$$\mathbb{P}\left\{\left|\frac{1}{\sqrt{e_k - s_k}}\sum_{t=s_k+1}^{e_k}\left\langle\mathbf{P}(t) - \mathbf{A}'(t), \widehat{\mathbf{P}}^{s_k,e_k}(b_k)/\|\widehat{\mathbf{P}}^{s_k,e_k}(b_k)\|_F\right\rangle\right| \geq \varepsilon\right\}$$

$$\leq 2\exp\left\{-c_4\frac{\varepsilon^2}{(1-\pi)^{-1} + \varepsilon/(c_3\kappa_k\sqrt{\Delta})}\right\}.$$

Combining these modifications, the signal-to-noise ratio (SNR) condition becomes

$$\kappa\sqrt{\Delta(1-\pi)} \geq C_{\mathrm{SNR}}\log(T)\sqrt{nL^{1/2} + d^2 m_{\max} + nd + Lm_{\max}},$$

Under this condition, we have that

$$\mathbb{P}\left\{\widetilde{K} = K \text{ and } |\widetilde{\eta}_k - \eta_k| \leq \epsilon_k, \forall k \in [K]\right\} \geq 1 - CT^{-c}, \quad \text{where } \epsilon_k = C_\epsilon\frac{\log(T)}{(1-\pi)\kappa_k^2}.$$

Compared to the independent case ($\pi = 0$) as shown in Theorem 1, the signal-to-noise ratio is stronger by a factor of $(1-\pi)^{-1/2}$ and the localization rate worsens by a factor of $(1-\pi)^{-1}$, reflecting the impact of the temporal dependence.

## C  EXTENSION TO CHANGES IN LATENT POSITIONS

In this section, we outline an extension that allows latent positions $X(t)$ to change across segments.

We redefine the D-MRDPG model as

$$\{\mathbf{A}(t)\}_{t=1}^T \sim \text{D-MRDPGs}\big(\{X(t)\}_{t=1}^T, \{W_{(l)}(t)\}_{l\in[L]}\big),$$

where both latent positions $X(t) \in \mathbb{R}^{n\times d}$ and layer weight matrices $\{W_{(l)}(t)\}_{l\in[L]} \subset \mathbb{R}^{d\times d}$ are allowed to vary over time. Assume that there exist change points

$$0 = \eta_0 < \eta_1 < \cdots < \eta_K < T = \eta_{K+1}$$

such that for $t \in [T-1]$,

$$\{X(t)\} \cup \{(W_{(l)}(t)\}_{l=1}^L \neq \{X(t+1)\} \cup \{W_{(l)}(t+1)\}_{l=1}^L$$

if and only if $t \in \{\eta_k\}_{k=1}^K$.

Let $\mathbf{P}(t)$ denote the probability tensor at time $t$. We retain the definition of the $k$-th jump size,

$$\kappa_k = \|\mathbf{P}(\eta_{k+1}) - \mathbf{P}(\eta_k)\|_{\mathrm{F}},$$

where $\mathbf{P}(t)$ is the probability tensor at time $t$.

For any $0 \leq s < t < e \leq T$, let $\widetilde{\mathbf{P}}^{s,e}(t)$ denote the expected CUSUM-transformed tensor as in (2) of the main text. To guarantee the Tucker low-rank structure required for TH-PCA, Assumption 1 must be modified accordingly.

**Modified assumptions:** $(i)$ **Mode-1 full-rank condition.** Assume that

$$\text{rank}(\mathcal{M}_1(\widetilde{\mathbf{P}}^{s,e}(t))) = d, \quad \frac{\sigma_1(\mathcal{M}_1(\widetilde{\mathbf{P}}^{s,e}(t)))}{\sigma_d(\mathcal{M}_1(\widetilde{\mathbf{P}}^{s,e}(t)))} \leq C_\sigma, \quad \sigma_d(\mathcal{M}_1(\widetilde{\mathbf{P}}^{s,e}(t))) \geq C_{\mathrm{gap}}\sqrt{n}.$$

When $X(t)$ is time-invariant, this reduces to Assumption 1 $(i)$.

$(ii)$ **Mode-3 low-rank condition.** Let $m_t^{s,e} = \text{rank}(\widetilde{Q}^{s,e}(t))$. Assume

$$\frac{\sigma_1(\mathcal{M}_3(\widetilde{\mathbf{P}}^{s,e}(t)))}{\sigma_{m_t^{s,e}}(\mathcal{M}_3(\widetilde{\mathbf{P}}^{s,e}(t)))} \leq C_\sigma, \quad \sigma_{m_t^{s,e}}(\mathcal{M}_3(\widetilde{\mathbf{P}}^{s,e}(t))) \geq C_{\text{gap}}.$$

This parallels Assumption 1 $(ii)$.

These conditions ensure the necessary low-rankness of the expected CUSUM tensors. We acknowledge that, compared with the original assumptions, the extended conditions, particularly $(i)$, are less direct or transparent in terms of the explicit structural properties of the network model.

Under the modified assumptions above, and provided the signal-to-noise ratio condition

$$\kappa\sqrt{\Delta} \geq C_{\text{SNR}} \log(T)\sqrt{nL^{1/2} + d^2 m_{\max} + nd + L m_{\max}},$$

where $m_{\max} = \max_{k \in [K+1]} \text{rank}\big(\mathcal{M}_3(\mathbf{P}(\eta_k))\big)$, the same two-stage algorithm continues to achieve the same localization rate as in Theorem 1. This shows that our methodology naturally extends to settings with time-varying latent positions without altering the core statistical guarantees.

## D ADDITIONAL ALGORITHMS

We present the tensor heteroskedastic principal component analysis (TH-PCA) algorithm introduced in Han et al. (2022), incorporating an additional truncation step, in Algorithm 2. Its subroutine, the heteroskedastic principal component analysis (H-PCA) algorithm proposed by Zhang et al. (2022), is provided in Algorithm 3.

---

**Algorithm 2** Tensor heteroskedastic principal component analysis, TH-PCA$(\mathbf{A}, (r_1, r_2, r_3), \tau_1, \tau_2)$

---

**INPUT:** Tensor $\mathbf{A} \in \mathbb{R}^{p_1 \times p_2 \times p_3}$, ranks $r_1, r_2, r_3 \in \mathbb{N}^+$, thresholds $\tau_1, \tau_2 \geq 0$
    **for** $s \in [3]$ **do**
        $\widehat{U}_s \leftarrow$ H-PCA$(\mathcal{M}_s(\mathbf{A})\mathcal{M}_s(\mathbf{A})^\top, r_s)$     ▷ See Algorithm 3 for H-PCA and Section 1.2 for $\mathcal{M}_s(\mathbf{A})$
    **end for**
    $\widetilde{\mathbf{P}} \leftarrow \mathbf{A} \times_1 \widehat{U}_1 \widehat{U}_1^\top \times_2 \widehat{U}_2 \widehat{U}_2^\top \times_3 \widehat{U}_3 \widehat{U}_3^\top$     ▷ See Section 1.2 for $\times_s$
    **for** $\{i, j, l\} \in [p_1] \times [p_2] \times [p_3]$ **do**
        $\widehat{\mathbf{P}}_{i,j,l} \leftarrow \min\big\{\tau_1, \max\{-\tau_2, \widetilde{\mathbf{P}}_{i,j,l}\}\big\}$
    **end for**
**OUTPUT:** $\widehat{\mathbf{P}} \in \mathbb{R}^{p_1 \times p_2 \times p_3}$

---

**Algorithm 3** Heteroskedastic principal component analysis, H-PCA$(\Sigma, r)$

---

**INPUT:** Matrix $\Sigma \in \mathbb{R}^{n \times n}$, rank $r \in \mathbb{N}^+$.
**Initialise:** $\widehat{\Sigma}^{(0)} \leftarrow \Sigma, \text{diag}\big(\widehat{\Sigma}^{(0)}\big) \leftarrow 0, T \leftarrow 5 \log\{\sigma_{\min}(\Sigma)/n\}$
    **for** $t \in \{0\} \cup [T-1]$ **do**
        Singular value decomposition $\widehat{\Sigma}^{(t)} = \sum_{i=1}^n \sigma^{i,(t)}\mathbf{u}^{i,(t)}(\mathbf{v}^{i,(t)})^\top, \quad \sigma^{1,(t)} \geq \cdots \geq \sigma^{n,(t)} \geq 0$
        $\widetilde{\Sigma}^{(t)} \leftarrow \sum_{i=1}^r \sigma^{i,(t)}\mathbf{u}^{i,(t)}\big(\mathbf{v}^{i,(t)}\big)^\top$
        $\widehat{\Sigma}^{(t+1)} \leftarrow \widehat{\Sigma}^{(t)}, \text{diag}\big(\widehat{\Sigma}^{(t+1)}\big) \leftarrow \text{diag}\big(\widetilde{\Sigma}^{(t)}\big)$
    **end for**
    $U \leftarrow (\mathbf{u}^1, \ldots, \mathbf{u}^r)$ from top-$r$ left singular vectors of $\widehat{\Sigma}^{(T)}$
**OUTPUT:** $U \in \mathbb{R}^{n \times r}$

---

## E PROOF OF THEOREM 1

The proof of Theorem 1 is in Appendix E.1 with all necessary auxiliary results in Appendix E.2.

### E.1    PROOF OF THEOREM 1

*Proof.* We first define the event

$$\mathcal{A} = \left\{ \widetilde{K} = K \text{ and } |b_k - \eta_k| \leq \tilde{\epsilon}, \, \forall k \in [K] \right\}, \quad \text{where } \tilde{\epsilon} = C_{\tilde{\epsilon}} \log(T) \left\{ \frac{n\sqrt{L} \log^{1/2}(T)}{\kappa^2} + \frac{\sqrt{\Delta}}{\kappa} \right\},$$

where $\{b_k\}_{k=1}^{\widetilde{K}}$ are preliminary change point estimates obtained from **Stage I** in Algorithm 1. Then by Proposition 4, it holds that

$$\mathbb{P}\{\mathcal{A}\} \geq 1 - C_0 T^{-c_0}$$

and $C_{\tilde{\epsilon}}, C_0, c_0 > 0$ are absolute constants. Since $\{\mathbf{A}'(t)\}_{t=1}^{T} \cup \{\mathbf{B}'(t)\}_{t=1}^{T}$ are independent of $\{\mathbf{A}(t)\}_{t=1}^{T} \cup \{\mathbf{B}(t)\}_{t=1}^{T}$, the distribution of $\{\mathbf{A}'(t)\}_{t=1}^{T} \cup \{\mathbf{B}'(t)\}_{t=1}^{T}$ remains unaffected under the conditioning on the event $\mathcal{A}$. All subsequent analysis in this proof is carried out under the event $\mathcal{A}$. Consequently, we can derive that

$$|b_k - \eta_k| \leq \tilde{\epsilon} \leq \Delta/6, \quad \forall k \in [K], \tag{7}$$

where the last inequality follows from Assumption 2 and the fact that $C_{\text{SNR}}$ is a sufficiently large constant.

**Step 1.** We first establish that for any $k \in [K]$, each working interval $(s_k, e_k)$ contains exactly one true change point, namely $\eta_k$, and the two endpoints are well separated.

From (7), we obtain that $\eta_k \in [b_{k-1}, b_{k+1}]$,

$$\eta_k - b_{k-1} \geq \eta_k - \eta_{k-1} - |\eta_{k-1} - b_{k-1}| \geq \Delta - \Delta/6 \geq 5\Delta/6,$$

and

$$b_{k+1} - \eta_k \geq \eta_{k+1} - \eta_k - |\eta_{k+1} - b_{k+1}| \geq \Delta - \Delta/6 \geq 5\Delta/6.$$

Similarly, we can derive that

$$\min\{b_k - b_{k-1}, \, b_{k+1} - b_k\} \geq 2\Delta/3.$$

As a result, the working interval

$$(s_k, e_k] = \big( b_{k-1} + (b_k - b_{k-1})/2, \, b_{k+1} - (b_{k+1} - b_k)/2 \big],$$

contains exactly one change point $\eta_k$. For any $t \in (s_k, e_k)$, denote $m_t^{s_k, e_k} = \text{rank}(\widetilde{Q}^{s_k, e_k}(t))$ with $\widetilde{Q}^{s_k, e_k}(t)$ defined in (3). Then we have that

$$m_t^{s_k, e_k} \leq m_k + m_{k+1} \leq 2m_{\max}, \tag{8}$$

where $m_k = \text{rank}\big(Q(\eta_k)\big)$ with $Q(\eta_k)$ defined in (4), and $m_{\max} = \max_{k \in [K+1]} m_k$.

In addition, we have that

$$b_k - s_k = (b_k - b_{k-1})/2 \geq \Delta/3,$$

and

$$e_k - b_k = (b_{k+1} - b_k)/2 \geq \Delta/3.$$

Therefore,

$$\min\{e_k - b_k, b_k - s_k\} \geq \Delta/3. \tag{9}$$

**Step 2.** We now show that the population statistics $\widetilde{\mathbf{P}}^{s_k, e_k}(b_k)$ provide a sufficiently strong signal within each working interval $(s_k, e_k]$.

By Lemma 6, it holds that

$$\left\| \widetilde{\mathbf{P}}^{s_k, e_k}(t) \right\|_{\mathrm{F}}^2 = \begin{cases} \frac{t - s_k}{(e_k - s_k)(e_k - t)} (e_k - \eta_k)^2 \kappa_k^2, & s_k < t \leq \eta_k, \\ \frac{e_k - t}{(e_k - s_k)(t - s_k)} (\eta_k - s_k)^2 \kappa_k^2, & \eta_k < t < e_k. \end{cases}$$

Define the scaling factor

$$\widetilde{\Delta}_k = \sqrt{\frac{(b_k - s_k)(e_k - b_k)}{e_k - s_k}}.$$

Assuming without loss of generality that $b_k \leq \eta_k$ and using (9), we obtain

$$\widetilde{\Delta}_k^2 \geq \frac{\min\{b_k - s_k,\ e_k - b_k\}}{2} \geq \frac{\Delta}{6}. \tag{10}$$

Thus, we have that

$$\left\|\widetilde{\mathbf{P}}^{s_k,e_k}(b_k)\right\|_{\mathrm{F}}^2 = \frac{b_k - s_k}{(e_k - s_k)(e_k - b_k)}(e_k - \eta_k)^2\,\kappa_k^2 = \widetilde{\Delta}_k^2\left(\frac{e_k - \eta_k}{e_k - b_k}\right)^2 \kappa_k^2$$

$$= \widetilde{\Delta}_k^2\left(1 - \frac{\eta_k - b_k}{e_k - b_k}\right)^2 \kappa_k^2 \geq \frac{\Delta}{6}(1 - \frac{\Delta/6}{\Delta/3})^2\kappa_k^2 = \Delta\kappa_k^2/24, \tag{11}$$

where the first inequality follows from (7), (9) and (10).

**Step 3.** Note that each entry of the tensor $\widetilde{\mathbf{B}'}^{s,e}(b_k)$ is independently $c_\sigma$-sub-Gaussian distributed with mean tensor $\mathbb{E}\{\widetilde{\mathbf{B}'}^{s,e}(b_k)\} = \widetilde{\mathbf{P}}^{s_k,e_k}(\eta_k)$ and an absolute constant $c_\sigma > 0$. By Theorem 4 and Lemma 5 in Wang et al. (2025), and Assumption 1 $(i)$ and $(ii)$, for any $t \in (s,e)$, it holds that

$$\mathbb{P}\left\{\left\|\widehat{\mathbf{P}}^{s_k,e_k}(t) - \widetilde{\mathbf{P}}^{s_k,e_k}(t)\right\|_F \leq C_1\sqrt{(d^2 m_t^{s,e} + nd + Lm_t^{s,e})\log(T)}\right\} \geq 1 - T^{-c_1},$$

for some constants $C_1 > 0$ and $c_1 > 3$. By (8), we can derive that

$$\mathbb{P}\left\{\left\|\widehat{\mathbf{P}}^{s_k,e_k}(t) - \widetilde{\mathbf{P}}^{s_k,e_k}(t)\right\|_F \leq C_2\sqrt{(d^2 m_{\max} + nd + Lm_{\max})\log(T)}\right\} \geq 1 - T^{-c_1},$$

where $C_2 > 0$ is a constant. Define the event

$$\mathcal{B} = \left\{ \sup_{\substack{0 \leq s_k < t < e_k \leq T \\ (s_k,e_k)\ \text{contains only one change point } \eta_k}} \left\|\widehat{\mathbf{P}}^{s_k,e_k}(t) - \widetilde{\mathbf{P}}^{s_k,e_k}(t)\right\|_F \leq \right.$$

$$\left. C_{\mathcal{B}}\sqrt{(d^2 m_{\max} + nd + Lm_{\max})\log(T)}\right\}. \tag{12}$$

with a constant $C_{\mathcal{B}} > 0$. By the union bound argument, it holds that

$$\mathbb{P}\{\mathcal{B}\} \geq 1 - T^{-c_{\mathcal{B}}},$$

with a constant $c_{\mathcal{B}} > 0$. By the event $\mathcal{B}$ and the triangle equality, we have that

$$\left\|\widehat{\mathbf{P}}^{s_k,e_k}(b_k)\right\|_{\mathrm{F}} \geq \left\|\widetilde{\mathbf{P}}^{s_k,e_k}(b_k)\right\|_{\mathrm{F}} - C_{\mathcal{B}}\sqrt{(d^2 m_{\max} + nd + Lm_{\max})\log(T)} \geq \kappa_k\sqrt{\Delta}/48, \tag{13}$$

where the last inequality follows from (11), Assumption 2 and the fact that $C_{\mathrm{SNR}}$ is a sufficiently large constant. As a consequence,

$$2\left\langle \widetilde{\mathbf{P}}^{s_k,e_k}(b_k)/\|\widetilde{\mathbf{P}}^{s_k,e_k}(b_k)\|_{\mathrm{F}}, \widehat{\mathbf{P}}^{s_k,e_k}(b_k)/\|\widehat{\mathbf{P}}^{s_k,e_k}(b_k)\|_{\mathrm{F}}\right\rangle$$

$$=2-\left\|\frac{\widetilde{\mathbf{P}}^{s_k,e_k}(b_k)}{\|\widetilde{\mathbf{P}}^{s_k,e_k}(b_k)\|_{\mathrm{F}}} - \frac{\widehat{\mathbf{P}}^{s_k,e_k}(b_k)}{\|\widehat{\mathbf{P}}^{s_k,e_k}(b_k)\|_{\mathrm{F}}}\right\|_{\mathrm{F}}^2$$

$$=2-\left\|\frac{\left(\widetilde{\mathbf{P}}^{s_k,e_k}(b_k) - \widehat{\mathbf{P}}^{s_k,e_k}(b_k)\right)\|\widetilde{\mathbf{P}}^{s_k,e_k}(b_k)\|_{\mathrm{F}}}{\|\widetilde{\mathbf{P}}^{s_k,e_k}(b_k)\|_{\mathrm{F}}\|\widehat{\mathbf{P}}^{s_k,e_k}(b_k)\|_{\mathrm{F}}}\right.$$

$$\left.+\frac{\widetilde{\mathbf{P}}^{s_k,e_k}(b_k)\left(\|\widehat{\mathbf{P}}^{s_k,e_k}(b_k)\|_{\mathrm{F}} - \|\widetilde{\mathbf{P}}^{s_k,e_k}(b_k)\|_{\mathrm{F}}\right)}{\|\widetilde{\mathbf{P}}^{s_k,e_k}(b_k)\|_{\mathrm{F}}\|\widehat{\mathbf{P}}^{s_k,e_k}(b_k)\|_{\mathrm{F}}}\right\|_{\mathrm{F}}^2$$

$$\geq 2-\left(\frac{\|\widetilde{\mathbf{P}}^{s_k,e_k}(b_k) - \widehat{\mathbf{P}}^{s_k,e_k}(b_k)\|_{\mathrm{F}}\|\widetilde{\mathbf{P}}^{s_k,e_k}(b_k)\|_{\mathrm{F}}}{\|\widetilde{\mathbf{P}}^{s_k,e_k}(b_k)\|_{\mathrm{F}}\|\widehat{\mathbf{P}}^{s_k,e_k}(b_k)\|_{\mathrm{F}}}\right.$$

$$\left.+\frac{\|\widetilde{\mathbf{P}}^{s_k,e_k}(b_k)\|_{\mathrm{F}}\|\widehat{\mathbf{P}}^{s_k,e_k}(b_k) - \widetilde{\mathbf{P}}^{s_k,e_k}(b_k)\|_{\mathrm{F}}}{\|\widetilde{\mathbf{P}}^{s_k,e_k}(b_k)\|_{\mathrm{F}}\|\widehat{\mathbf{P}}^{s_k,e_k}(b_k)\|_{\mathrm{F}}}\right)^2$$

$$=2-\left(\frac{\left(\|\widetilde{\mathbf{P}}^{s_k,e_k}(b_k)\|_{\mathrm{F}} + \|\widetilde{\mathbf{P}}^{s_k,e_k}(b_k)\|_{\mathrm{F}}\right)\|\widehat{\mathbf{P}}^{s_k,e_k}(b_k) - \widetilde{\mathbf{P}}^{s_k,e_k}(b_k)\|_{\mathrm{F}}}{\|\widetilde{\mathbf{P}}^{s_k,e_k}(b_k)\|_{\mathrm{F}}\|\widehat{\mathbf{P}}^{s_k,e_k}(b_k)\|_{\mathrm{F}}}\right)^2$$

$$\geq 2-4\left(\frac{\|\widetilde{\mathbf{P}}^{s_k,e_k}(b_k) - \widehat{\mathbf{P}}^{s_k,e_k}(b_k)\|_{\mathrm{F}}}{\min\{\|\widetilde{\mathbf{P}}^{s_k,e_k}(b_k)\|_{\mathrm{F}}, \|\widehat{\mathbf{P}}^{s_k,e_k}(b_k)\|_{\mathrm{F}}\}}\right)^2$$

$$\geq 2-4\frac{48^2 C_{\mathcal{B}}^2(d^2 m_t^{s,e} + nd + L m_t^{s,e})\log(T)}{\kappa_k^2 \Delta} \geq 1,$$

where the first inequality follows from the reverse triangle inequality, the third inequality follows from the definition of the event $\mathcal{B}$, (11) and (13), and the final inequality follows from Assumption 2 and the fact that $C_{\mathrm{SNR}}$ is a sufficiently large constant. Therefore,

$$\left\langle \widetilde{\mathbf{P}}^{s_k,e_k}(b_k), \widehat{\mathbf{P}}^{s_k,e_k}(b_k)/\|\widehat{\mathbf{P}}^{s_k,e_k}(b_k)\|_{\mathrm{F}}\right\rangle \geq \|\widetilde{\mathbf{P}}^{s_k,e_k}(b_k)\|_{\mathrm{F}}/2 \geq \kappa_k\sqrt{\Delta/96}, \qquad (14)$$

where the last inequality follows from (11).

**Step 4.** Since $\{\mathbf{A}'(t)\}_{t=1}^T$ is independent of $\{\mathbf{B}'(t)\}_{t=1}^T$, the distribution of $\{\mathbf{A}'(t)\}_{t=1}^T$ remain unaffected under the conditioning on the event $\mathcal{B}$. By the truncation in the construction of $\widehat{\mathbf{P}}^{s_k,e_k}(b_k)$ stated in Algorithm 2, we have that

$$\|\widehat{\mathbf{P}}^{s_k,e_k}(b_k)\|_\infty \leq \sqrt{\frac{(e_k - b_k)(b_k - s_k)}{(e_k - s_k)}}$$

Combined with (13), it follows that

$$(e_k - s_k)^{-1/2}\|\widehat{\mathbf{P}}^{s_k,e_k}(b_k)\|_\infty/\|\widehat{\mathbf{P}}^{s_k,e_k}(b_k)\|_{\mathrm{F}} \leq \frac{1}{c_3\kappa_k\sqrt{\Delta}},$$

for some constant $c_3 > 0$. Applying Lemma 5, we obtain for any $\varepsilon > 0$

$$\mathbb{P}\left\{\left|\frac{1}{\sqrt{e_k - s_k}}\sum_{t=s_k+1}^{e_k}\left\langle \mathbf{P}(t) - \mathbf{A}'(t), \widehat{\mathbf{P}}^{s_k,e_k}(b_k)/\|\widehat{\mathbf{P}}^{s_k,e_k}(b_k)\|_{\mathrm{F}}\right\rangle\right| \geq \varepsilon\right\}$$

$$\leq 2\exp\left\{-c_4\frac{\varepsilon^2}{1 + \varepsilon/(c_3\kappa_k\sqrt{\Delta})}\right\}.$$

where $c_4 > 0$ is a constant. Choosing $\varepsilon = C_3\sqrt{\log(T)}$ for a large enough constant $C_3 > 0$, and applying Assumption 2 and the fact that $C_{\mathrm{SNR}}$ is a sufficiently large constant, we finally derive that

$$\mathbb{P}\left\{\left|\frac{1}{\sqrt{e_k - s_k}}\sum_{t=s_k+1}^{e_k}\left\langle \mathbf{P}(t) - \mathbf{A}'(t), \widehat{\mathbf{P}}^{s_k,e_k}(b_k)/\|\widehat{\mathbf{P}}^{s_k,e_k}(b_k)\|_{\mathrm{F}}\right\rangle\right| \geq C_3\sqrt{\log(T)}\right\} \leq 2T^{-c_5}, \tag{15}$$

where $c_5 > 3$ is a constant. A similar argument also demonstrates that

$$\mathbb{P}\left\{\left|\left\langle \widetilde{\mathbf{P}}^{s_k,e_k}(t) - \widetilde{\mathbf{A}'}^{s_k,e_k}(t), \widehat{\mathbf{P}}^{s_k,e_k}(b_k)/\|\widehat{\mathbf{P}}^{s_k,e_k}(b_k)\|_{\mathrm{F}}\right\rangle\right| \geq C_3\sqrt{\log(T)}\right\} \leq 2T^{-c_5}. \tag{16}$$

**Step 5.** We now consider the univariate time series defined for all $t \in (s_k, e_k)$ as

$$y(t) = \left\langle \mathbf{A}'(t), \widehat{\mathbf{P}}^{s_k,e_k}(b_k)/\|\widehat{\mathbf{P}}^{s_k,e_k}(b_k)\|_{\mathrm{F}}\right\rangle$$

and

$$y^{s_k,e_k}(t) = \left\langle \widetilde{\mathbf{A}'}^{s_k,e_k}(t), \widehat{\mathbf{P}}^{s_k,e_k}(b_k)/\|\widehat{\mathbf{P}}^{s_k,e_k}(b_k)\|_{\mathrm{F}}\right\rangle.$$

Conditional on the event $\mathcal{B}$, define the corresponding mean functions

$$f(t) = \mathbb{E}(y(t)) = \left\langle \mathbf{P}(t), \widehat{\mathbf{P}}^{s_k,e_k}(b_k)/\|\widehat{\mathbf{P}}^{s_k,e_k}(b_k)\|_{\mathrm{F}}\right\rangle,$$

and

$$f^{s_k,e_k}(t) = \mathbb{E}(y^{s_k,e_k}(t)) = \left\langle \widetilde{\mathbf{P}}^{s_k,e_k}(t), \widehat{\mathbf{P}}^{s_k,e_k}(b_k)/\|\widehat{\mathbf{P}}^{s_k,e_k}(b_k)\|_{\mathrm{F}}\right\rangle.$$

The function $f(t)$ is a piecewise constant on $(s_k, e_k]$ with a single change point at $\eta_k$. Using (14), we obtain that

$$\left|f^{s_k,e_k}(\eta_k)|\right| \geq \kappa_k\sqrt{\Delta/96},$$

Moreover, from (15), (16) and an union bound argument, we have that

$$\mathbb{P}\left\{\sup_{\substack{0\leq s_k < t < e_k \leq T \\ (s_k,e_k) \text{ contains only one change point } \eta_k}}\left|\frac{1}{\sqrt{e_k - s_k}}\sum_{t=s_k+1}^{e_k}(y(t) - f(t))\right|\right.$$

$$\left. \geq C_3\sqrt{\log(T)}\right\} \leq 2T^{-c_6}$$

and

$$\mathbb{P}\left\{\sup_{\substack{0\leq s_k < t < e_k \leq T \\ (s_k,e_k) \text{ contains only one change point } \eta_k}}\left|y^{s_k,e_k}(t) - f^{s_k,e_k}(t)\right| \geq C_3\sqrt{\log(T)}\right\} \leq 2T^{-c_6},$$

for some constant $c_6 > 0$. Applying Lemma 12 from Wang et al. (2017) with $\lambda = C\sqrt{\log(T)}$, it follows that the estimated change point $\widetilde{\eta}_k = \arg\max_{s_k < t < e_k}|y^{s_k,e_k}(t)|$ is an undetected change point and satisfies for a large enough constant $C_5 > 0$,

$$|\widetilde{\eta}_k - \eta_k| \leq C_5\frac{\log(T)}{\kappa_k^2},$$

which completes the proof.

$\square$

### E.2 ADDITIONAL RESULTS

**Proposition 4.** *Let $\{b_k\}_{k=1}^{\widetilde{K}}$ denote the output of **Stage I** in Algorithm 1 applied to two independent adjacency tensor sequences $\{\mathbf{A}(t)\}_{t\in[T]}, \{\mathbf{B}(t)\}_{t\in[T]} \subset \{0,1\}^{n\times n\times L}$, generated according to Definition 2 and satisfying Model 1 and Assumption 2. Suppose the threshold $\tau$ is chosen such that*

$$c_{\tau,1} n\sqrt{L}\log^{3/2}(T) < \tau < c_{\tau,2}\kappa^2\Delta, \tag{17}$$

*where $c_{\tau,1}, c_{\tau,2} > 0$ are sufficiently large and small absolute constants, respectively.*

*Then, it holds that*

$$\mathbb{P}\Big\{\widetilde{K} = K \text{ and } |b_k - \eta_k| \leq \tilde{\epsilon}, \, \forall k \in [K]\Big\} \geq 1 - CT^{-c},$$

*where*

$$\tilde{\epsilon} = C_{\tilde{\epsilon}}\log(T)\Big\{\frac{n\sqrt{L}\log^{1/2}(T)}{\kappa^2} + \frac{\sqrt{\Delta}}{\kappa}\Big\},$$

*and $C_{\tilde{\epsilon}}, C, c > 0$ are absolute constants.*

*Proof.* The proof presented here is a minor modification of Theorem 1 in Wang et al. (2021). For completeness, we include the full details below.

For $0 \leq s < t < e \leq T$, we define the event

$$\mathcal{A}(s,t,e) = \Big\{\big|\langle \widetilde{\mathbf{A}}^{s,e}(t), \widetilde{\mathbf{B}}^{s,e}(t)\rangle - \|\widetilde{\mathbf{P}}^{s,e}(t)\|_{\mathrm{F}}^2\big| \leq C_{\mathcal{A}}\log(T)\big(\|\widetilde{\mathbf{P}}^{s,e}(t)\|_{\mathrm{F}} + \log^{1/2}(T)n\sqrt{L}\big)\Big\},$$

where $\widetilde{\mathbf{P}}^{s,e}(t)$ is defined in (2) and $C_{\mathcal{A}} > 0$ is a constant. Due to Lemma S.4 in Wang et al. (2021), it holds that $\mathbb{P}(\mathcal{A}(s,t,e)^c) \leq C_1 T^{-c_1}$ for some constants $C_1 > 0$ and $c_1 > 3$. By an union bound argument, it holds that

$$\mathbb{P}(\mathcal{A}) = \mathbb{P}\Big\{\bigcup_{0 \leq s < t < e \leq T} \mathcal{A}(s,t,e)\Big\} \geq 1 - C_1 T^{3-c_1}.$$

All subsequent analysis in this proof is carried out under the event $\mathcal{A}$.

We aim to demonstrate that, conditioned on the event $\mathcal{A}$ and assuming that the algorithm has accurately detected and localized change points so far, the procedure will also successfully identify any remaining undetected change point, if one exists, and estimate its location within an error of $\tilde{\epsilon}$. To this end, it suffices to consider an arbitrary time interval $0 \leq s < e \leq T$ that satisfies

$$\eta_{r-1} \leq s < \eta_r < \cdots < \eta_{r+q} < e \leq \eta_{r+q+1}, \quad q \geq -1,$$

and

$$\max\big\{\min\{\eta_r - s, s - \eta_{r-1}\}, \min\{\eta_{r+q+1} - e, e - \eta_{r+q}\}\big\} \leq \tilde{\epsilon},$$

where $q = -1$ indicates that there is no change point contained in $(s, e)$ and

$$\tilde{\epsilon} = C_{\tilde{\epsilon}}\log(T)\Big\{\frac{n\sqrt{L}\log^{1/2}(T)}{\kappa^2} + \frac{\sqrt{\Delta}}{\kappa}\Big\}.$$

with an absolute constant $C_{\tilde{\epsilon}} > 0$. By Assumption 2, we have that

$$\tilde{\epsilon} \leq C_{\tilde{\epsilon}}\Delta\Big(\frac{1}{C_{\mathrm{SNR}}^2\log^{1/2}(T)} + \frac{1}{C_{\mathrm{SNR}}\sqrt{nL^{1/2}}}\Big) \leq \Delta/64, \tag{18}$$

where the final inequality follows that $C_{\mathrm{SNR}}$ is large enough. Consequently, for any change point $\eta_k$, it must be that either $|\eta_k - s| \leq \tilde{\epsilon}$ or $|\eta_k - s| \geq \Delta - \tilde{\epsilon} \geq 3\Delta/4$. This implies that if $\min\{|e - \eta_k|, |\eta_k - s|\} \leq \tilde{\epsilon}$, then $\eta_k$ corresponds to a change point that has already been detected and estimated within an error of at most $\tilde{\epsilon}$ during the previous induction step. We refer to a change point $\eta_k$ in $(s, e)$ as undetected if

$$\min\{\eta_k - s, \eta_k - e\} \geq 3\Delta/4.$$

To complete the induction step, it suffices to show that $\mathrm{SBS}\big((s,e), \tau, \mathcal{J}\big)$ satisfies the following tow properties: $(i)$ It does not detect any new change point within $(s, e)$ if all change points in that

interval have already been detected; and $(ii)$ It detects a point $b$ in $(s, e)$ such that $|\eta_k - b| \leq \tilde{\epsilon}$ if there exists at least one previously undetected change point $\eta_k$ in $(s, e)$.

**Step 1.** Assume that there are no undetected change points within the interval $(s, e)$. Then, for any interval $(\alpha', \beta'] \in \mathcal{J}$ with $(\alpha', \beta'] \subseteq (s, e]$, one of the following scenarios must hold: $(i)$ The interval $(\alpha', \beta')$ contains no change points; $(ii)$ The interval $(\alpha', \beta')$ contains exactly one change point $\eta_k$ and $\min\{\eta_k - \alpha', \beta' - \eta_k\} \leq \tilde{\epsilon}$; $(iii)$ The interval $(\alpha', \beta')$ contains two change points $\eta_k$ and $\eta_{k+1}$, and $\max\{\eta_k - \alpha', \beta' - \eta_{k+1}\} \leq \tilde{\epsilon}$.

We focus on analyzing the scenario $(iii)$, as the other two scenarios are similar and more straightforward. If scenario $(iii)$ holds, then by (18), we have

$$\tilde{\epsilon} \leq \Delta/64 \leq (\beta' - \alpha')/64,$$

This implies that the interval

$$(\alpha, \beta] = \left(\alpha' + 64^{-1}(\beta' - \alpha'), \beta' - 64^{-1}(\beta' - \alpha')\right],$$

contains no change points. Note that $\widetilde{\mathbf{P}}^{\alpha,\beta}(t) = 0$ for all $t \in (\alpha, \beta)$, since there are no true change points within $(\alpha, \beta)$. Moreover, by the event $\mathcal{A}$,

$$\max_{\alpha < t < \beta} \left\langle \widetilde{\mathbf{A}}^{\alpha,\beta}(t), \widetilde{\mathbf{B}}^{\alpha,\beta}(t) \right\rangle \leq C_{\mathcal{A}} n \sqrt{L} \log^{3/2}(T).$$

Therefore, if the input parameter $\tau$ satisfies

$$\tau > C_{\mathcal{A}} n \sqrt{L} \log^{3/2}(T),$$

Algorithm 1 will correctly reject the existence of undetected change points.

**Step 2.** Now suppose there exists a change point $\eta_k \in (s, e)$ such that

$$\min\{\eta_k - s, \eta_k - e\} \geq 3\Delta/4.$$

Let $a_{\mathcal{I}}, b_{\mathcal{I}}$ and $\mathcal{I}^*$ be as defined in the procedure $\mathrm{SBS}\left((s, e), \tau, \mathcal{J}\right)$. Denote $\mathcal{I}^* = (\alpha^{*\prime}, \beta^{*\prime}]$. By Lemma 8 in Madrid Padilla et al. (2022), for any change point $\eta_k \in (s, e)$ satisfying $\min\{\eta_k - s, e - \eta_k\} \geq 3\Delta/4$, there exists an interval $(\alpha', \beta'] \subseteq (s, e]$ containing only one $\eta_k$ such that

$$\eta_k - 3\Delta/4 \leq \alpha' \leq \eta_k - 3\Delta/16 \quad \text{and} \quad \eta_k + 3\Delta/16 \leq \beta' \leq \eta_k + 3\Delta/4.$$

Since $(\alpha, \beta] = [\alpha' + (\beta' - \alpha')/64, \beta' - (\beta' - \alpha')/64]$, we have

$$\eta_k - \Delta 3/4 \leq \alpha \leq \eta_k - \Delta/8 \quad \text{and} \quad \eta_k + \Delta/8 \leq \beta \leq \eta_k + \Delta 3/4.$$

On the event $\mathcal{A}$, it holds that

$$\left\langle \widetilde{\mathbf{A}}^{\alpha,\beta}(\eta_k), \widetilde{\mathbf{B}}^{\alpha,\beta}(\eta_k) \right\rangle \geq \|\widetilde{\mathbf{P}}^{\alpha,\beta}(\eta_k)\|_{\mathrm{F}}^2 - C_{\mathcal{A}} \log(T)\left\{\log^{1/2}(T) n \sqrt{L} + \|\widetilde{\mathbf{P}}^{\alpha,\beta}(\eta_k)\|_{\mathrm{F}}\right\}. \quad (19)$$

Furthermore, by Lemma 6, it hold that

$$\|\widetilde{\mathbf{P}}^{\alpha,\beta}(\eta_k)\|_{\mathrm{F}}^2 = \frac{(\eta_k - \alpha)(\beta - \eta_k)}{\beta - \alpha} \kappa_k^2.$$

Then we can derive that

$$\|\widetilde{\mathbf{P}}^{\alpha,\beta}(\eta_k)\|_{\mathrm{F}}^2 \geq \kappa_k^2 \Delta/16 \quad \text{and} \quad \|\widetilde{\mathbf{P}}^{\alpha,\beta}(\eta_k)\|_{\mathrm{F}}^2 \leq 3\kappa_k^2 \Delta/4. \quad (20)$$

Combining (19) and (20), and using Assumption 2 along with fact that $C_{\mathrm{SNR}}$ is a sufficiently large constant, we obtain

$$\left\langle \widetilde{\mathbf{A}}^{\alpha,\beta}(\eta_k), \widetilde{\mathbf{B}}^{\alpha,\beta}(\eta_k) \right\rangle \geq \kappa_k^2 \Delta/16 - \kappa_k^2 \Delta/64 - \kappa_k^2 \Delta/64 \geq \kappa_k^2 \Delta/32.$$

By the definition of $\mathcal{I}^*$, it follows that

$$a_{\mathcal{I}^*} = \left\langle \tilde{A}^{\alpha^*,\beta^*}(b_{\mathcal{I}^*}), \tilde{B}^{\alpha^*,\beta^*}(b_{\mathcal{I}^*}) \right\rangle \geq (\kappa_{\max}^{s,e})^2 \Delta/32, \quad (21)$$

where

$$\kappa_{\max}^{s,e} = \max\left\{\kappa_k \colon \min\{\eta_k - s, e - \eta_k\} \geq 3\Delta/4\right\}.$$

Therefore, if the threshold $\tau$ satisfies
$$\tau < \kappa^2 \Delta/32,$$
Algorithm 1 will consistently detect the existence of any previously undetected change points within the interval.

**Step 3.** Suppose there exists at least one undetected change point $\eta_k \in (s, e)$ such that
$$\min\{\eta_k - s, \eta_k - e\} \geq 3\Delta/4.$$

Let $a_{\mathcal{I}}, b_{\mathcal{I}}$ and $\mathcal{I}^*$ be defined according to the procedure $\text{SBS}\big((s, e), \tau, \mathcal{J}\big)$, and denote $\mathcal{I}^* = (\alpha^{*\prime}, \beta^{*\prime}]$. To complete the induction step, it suffices to establish that there exists an undetected change point $\eta_k \in (\alpha^{*\prime}, \beta^{*\prime})$ satisfying

$$\min\{\eta_k - \alpha^{*\prime}, \beta^{*\prime} - \eta_k\} \geq 3\Delta/4, \tag{22}$$

and that

$$|b_{\mathcal{I}^*} - \eta_k| \leq \tilde{\epsilon}. \tag{23}$$

**Step 3.1. Proof of** (22)**.** Let

$$(\alpha^*, \beta^*] = (\alpha^{*\prime} + (\beta^{*\prime} - \alpha^{*\prime})/64, \beta^{*\prime} - (\beta^{*\prime} - \alpha^{*\prime})/64]. \tag{24}$$

Assume for contradiction that

$$\max_{\alpha^* < t < \beta^*} \big\| \widetilde{\mathbf{P}}^{\alpha^*, \beta^*}(t) \big\|_{\mathrm{F}}^2 < (\kappa_{\max}^{s,e})^2 \Delta/64. \tag{25}$$

Then on the event $\mathcal{A}$, we obtain

$$\max_{\alpha^* < t < \beta^*} \big\langle \widetilde{A}^{\alpha^*, \beta^*}(t), \widetilde{B}^{\alpha^*, \beta^*}(t) \big\rangle$$

$$\leq \max_{\alpha^* < t < \beta^*} \big\| \widetilde{\mathbf{P}}^{\alpha^*, \beta^*}(t) \big\|_{\mathrm{F}}^2 + C_{\mathcal{A}} \log(T) \max_{\alpha^* < t < \beta^*} \big\| \widetilde{\mathbf{P}}^{\alpha^*, \beta^*}(t) \big\|_{\mathrm{F}} + C_{\mathcal{A}} \log^{3/2}(T) n \sqrt{L}$$

$$< (\kappa_{\max}^{s,e})^2 \Delta/64 + C_{\mathcal{A}} \log(T) \kappa_{\max}^{s,e} \sqrt{\Delta}/8 + C_{\mathcal{A}} \log^{3/2}(T) n \sqrt{L}$$

$$\leq (\kappa_{\max}^{s,e})^2 \Delta/64 + (\kappa_{\max}^{s,e})^2 \Delta/128 + (\kappa_{\max}^{s,e})^2 \Delta/128 = (\kappa_{\max}^{s,e})^2 \Delta/32,$$

where the second inequality follows from (25), the third inequality follows from Assumption 2 and the fact that $C_{\mathrm{SNR}}$ is a large enough constant. This contradicts the inequality (21). Thus,

$$\max_{\alpha^* < t < \beta^*} \big\| \widetilde{\mathbf{P}}^{\alpha^*, \beta^*}(t) \big\|_{\mathrm{F}}^2 \geq (\kappa_{\max}^{s,e})^2 \Delta/64. \tag{26}$$

Now, observe $(i)$ If $(\alpha^*, \beta^*)$ contains at least two change points, then $\beta^* - \alpha^* \geq \Delta$. $(ii)$ If it contains exactly one change point $\eta_k$, but $\min\{\eta_k - \alpha^*, \beta^* - \eta_k\} < \Delta/64$, then by Lemma 6, we would have

$$\max_{\alpha^* < t < \beta^*} \big\| \widetilde{P}^{\alpha^*, \beta^*}(t) \big\|_{\mathrm{F}}^2 = \big\| \widetilde{P}^{\alpha^*, \beta^*}(\eta_k) \big\|_{\mathrm{F}}^2 = \frac{(\eta_k - \alpha^*)(\beta^* - \eta_k)}{\beta^* - \alpha^*} \kappa_k^2$$

$$\leq \min\{\beta^* - \eta_k, \eta_k - \alpha^*\} \kappa_k^2 < (\kappa_{\max}^{s,e})^2 \Delta/64,$$

contradicting (26). Therefore, it has to be the case that $\min\{\eta_k - \alpha^*, \beta^* - \eta_k\} \geq \Delta/64$. Moreover, by (24), it holds that
$$\beta^{*,\prime} - \alpha^{*,\prime} \geq \beta^* - \alpha^* \geq \Delta/64.$$
Then, using Assumption 2 and the the fact that $C_{\mathrm{SNR}}$ is large enough, we have that

$$\tilde{\epsilon} \leq C_{\tilde{\epsilon}} \Delta \left( \frac{1}{C_{\mathrm{SNR}}^2 \log^{1/2 + 2\xi}(T)} + \frac{1}{C_{\mathrm{SNR}} \sqrt{n L^{1/2}} \log^{1+\xi}(T)} \right) \leq (\beta^{*,\prime} - \alpha^{*,\prime})/64,$$

Hence, by a similar argument as in **Step 1**, no previously detected change point lies in $(\alpha^*, \beta^*)$. Note that by (21), there is at least one undetected change point in $(\alpha^*, \beta^*)$.

**Step 3.2. Proof of** (23)**.** To this end, we apply Lemma S.5 in Wang et al. (2021). Define

$$\lambda = \max_{\alpha^* < t < \beta^*} \left| \big\langle \widetilde{\mathbf{A}}^{\alpha^*, \beta^*}(t), \widetilde{\mathbf{B}}^{\alpha^*, \beta^*}(t) \big\rangle - \big\| \widetilde{\mathbf{P}}^{\alpha^*, \beta^*}(t) \big\|_{\mathrm{F}}^2 \right|. \tag{27}$$

From (26), Assumption 2 and the fact that $C_{\mathrm{SNR}}$ is a sufficiently large constant, it follows that

$$c_2 \max_{\alpha^* < t < \beta^*} \left\| \widetilde{\mathbf{P}}^{\alpha^*,\beta^*}(t) \right\|_{\mathrm{F}}^2 / 2 \geq \max \Big\{ C_{\mathcal{A}} \log(T) \max_{\alpha^* < t < \beta^*} \left\| \widetilde{\mathbf{P}}^{\alpha^*,\beta^*}(t) \right\|_{\mathrm{F}},$$
$$C_{\mathcal{A}} \log^{3/2}(T) n\sqrt{L} \Big\}, \tag{28}$$

where $c_2 > 0$ is a small enough constant. By the definition of the event $\mathcal{A}$, we obtain

$$\lambda \leq C_{\mathcal{A}} \log(T) \Big\{ \max_{\alpha^* < t < \beta^*} \left\| \widetilde{\mathbf{P}}^{\alpha^*,\beta^*}(t) \right\|_{\mathrm{F}} + \log^{1/2}(T) n\sqrt{L} \Big\} \leq c_2 \max_{\alpha^* < t < \beta^*} \left\| \widetilde{\mathbf{P}}^{\alpha^*,\beta^*}(t) \right\|_{\mathrm{F}}^2, \tag{29}$$

where the last inequality follows from (28). Note that (21), (27) and (29) verify conditions (2), (3), (4) of Lemma S.5 in Wang et al. (2021), respectively. Therefore, Lemma S.5 in Wang et al. (2021) implies that there exists an undetected change point $\eta_k$ within $(s, e)$ such that

$$|\eta_k - b_{\mathcal{I}^*}| \leq \frac{C_3 \Delta \lambda}{\|\widetilde{P}^{\alpha^*,\beta^*}(\eta_k)\|_{\mathrm{F}}^2} \quad \text{and} \quad \|\widetilde{P}^{\alpha^*,\beta^*}(\eta_k)\|_{\mathrm{F}}^2 \geq c_4 \max_{\alpha^* \leq t \leq \beta^*} \|\widetilde{P}^{\alpha^*,\beta^*}(t)\|_{\mathrm{F}}^2, \tag{30}$$

where $C_3, c_4 > 0$ are constants. Then combining (29) and (30), we can derive that

$$\begin{aligned}
|\eta_k - b_{\mathcal{I}^*}| &\leq \frac{C_3 C_{\mathcal{A}} \Delta \log(T) \big\{ \max_{\alpha^* < t < \beta^*} \|\widetilde{\mathbf{P}}^{\alpha^*,\beta^*}(t)\|_{\mathrm{F}} + \log^{1/2}(T) n\sqrt{L} \big\}}{c_4 \max_{\alpha^* \leq t \leq \beta^*} \|\widetilde{\mathbf{P}}^{\alpha^*,\beta^*}(t)\|_{\mathrm{F}}^2} \\
&= \frac{C_3 C_{\mathcal{A}} \log(T)}{c_4} \left\{ \frac{\Delta}{\max_{\alpha^* < t < \beta^*} \|\widetilde{\mathbf{P}}^{\alpha^*,\beta^*}(t)\|_{\mathrm{F}}} + \frac{\Delta \log^{1/2}(T) n\sqrt{L}}{\max_{\alpha^* \leq t \leq \beta^*} \|\widetilde{P}^{\alpha^*,\beta^*}(t)\|_{\mathrm{F}}^2} \right\} \\
&\leq \frac{C_3 C_{\mathcal{A}} \log(T)}{c_4} \left\{ \frac{8\sqrt{\Delta}}{\kappa} + \frac{64 \log^{1/2}(T) n\sqrt{L}}{\kappa^2} \right\} \\
&\leq C_5 \log(T) \left\{ \frac{\sqrt{\Delta}}{\kappa} + \frac{\log^{1/2}(T) n\sqrt{L}}{\kappa^2} \right\},
\end{aligned}$$

where the second inequality follows form (26) and $C_5 > 0$ is an constant. This completes the induction step and therefore, the proof.

$\square$

**Lemma 5.** *Let $\{\mathbf{A}(t)\}_{t\in[T]}$ follow D-MRDPGs as in Definition 2. Let $\mathbf{V} \in \mathbb{R}^{n \times n \times L}$ and $\{w_t\}_{t=1}^T \subset \mathbb{R}$ satisfy $\sum_{t=1}^T w_t^2 = 1$. Then for any $\varepsilon > 0$, it holds that*

$$\mathbb{P}\left( \left| \sum_{t=1}^T w_t \big\langle \mathbf{A}(t) - \mathbf{P}(t), \mathbf{V}/\|\mathbf{V}\|_{\mathrm{F}} \big\rangle \right| \geq \varepsilon \right) \leq 2 \exp\left( -c \frac{\varepsilon^2}{1 + \varepsilon \|\mathbf{V}\|_{\mathrm{F}}^{-1} \|\mathbf{V}\|_{\infty} \max_{1 \leq t \leq T} |w_t|} \right),$$

*where $c > 0$ is an absolute constant and $\|\mathbf{V}\|_{\infty} = \max_{i,j\in[n],l\in[L]} |V_{i,j,l}|$.*

*Proof.* By definition of the tensor inner product, we have that

$$\sum_{t=1}^T w_t \big\langle \mathbf{A}(t) - \mathbf{P}(t), \mathbf{V}/\|\mathbf{V}\|_{\mathrm{F}} \big\rangle = \sum_{t=1}^T \sum_{i=1}^n \sum_{j=1}^n \sum_{l=1}^L \|\mathbf{V}\|_{\mathrm{F}}^{-1} w_t \mathbf{V}_{i,j,l} \Big\{ \big(\mathbf{A}(t)\big)_{i,j,l} - \big(\mathbf{P}(t)\big)_{i,j,l} \Big\}.$$

We can derive that

$$\sum_{t=1}^T \sum_{i=1}^n \sum_{j=1}^n \sum_{l=1}^L \|\mathbf{V}\|_{\mathrm{F}}^{-2} w_t^2 \mathbf{V}_{i,j,l}^2 = 1,$$

and

$$\max_{t\in[T],i,j\in[n],l\in[L]} \Big\{ \|\mathbf{V}\|_{\mathrm{F}}^{-1} w_t \mathbf{V}_{i,j,l} \Big\} \leq \|\mathbf{V}\|_{\mathrm{F}}^{-1} \|\mathbf{V}\|_{\infty} \max_{1 \leq t \leq T} |w_t|.$$

Since

$$\big\{ \big(\mathbf{A}(t)\big)_{i,j,l} - \big(\mathbf{P}(t)\big)_{i,j,l} \big\}_{i,j\in[n],l\in[L],t\in[T]}$$

are mutually independent centered Bernoulli random variables By Bernstein inequality (e.g. Theorem 2.8.2 in Vershynin, 2018), it holds with an absolute constant $c_0 > 0$ that

$$\mathbb{P}\left( \left| \sum_{t=1}^{T} w_t \langle \mathbf{A}(t) - \mathbf{P}(t), \mathbf{V}/\|\mathbf{V}\|_{\mathrm{F}} \rangle \right| \geq \varepsilon \right) \leq 2 \exp\left( -c_0 \frac{\varepsilon^2}{1 + \varepsilon \|\mathbf{V}\|_{\mathrm{F}}^{-1} \|\mathbf{V}\|_{\infty} \max_{1 \leq t \leq T} |w_t|} \right),$$

which completes the proof.

$\square$

**Lemma 6.** *Suppose the adjacency tensor sequence $\{\mathbf{B}(t)\}_{t \in [T]} \subset \{0,1\}^{n \times n \times L}$ is generated according to Definition 2 and satisfy Model 1. For any $0 \leq s < t < e \leq T$, let $\widetilde{\mathbf{P}}^{s,e}(t)$ be defined as in (2). If $(s,e)$ contains exactly one change point $\eta_k$, then for any $t \in (s,e)$*

$$\left\| \widetilde{\mathbf{P}}^{s,e}(t) \right\|_{\mathrm{F}}^2 = \begin{cases} \frac{t-s}{(e-s)(e-t)}(e - \eta_k)^2 \kappa_k^2, & s < t \leq \eta_k, \\ \frac{e-t}{(e-s)(t-s)}(\eta_k - s)^2 \kappa_k^2, & \eta_k < t < e. \end{cases}$$

*Proof.* This follows directly the definition of $\widetilde{\mathbf{P}}^{s,e}(t)$ in (2). $\square$

## F PROOFS OF THEOREMS 2 AND 3

*Proof.* **Step 1.** Preliminary bounds. We first define the event

$$\mathcal{A} = \left\{ \widetilde{K} = K \text{ and } |\widetilde{\eta}_k - \eta_k| \leq \epsilon_k, \, \forall k \in [K] \right\}, \quad \text{where} \quad \epsilon_k = C_\epsilon \frac{\log(T)}{\kappa_k^2}.$$

Then by Theorem 1, it holds that

$$\mathbb{P}\{\mathcal{A}\} \geq 1 - C_0 T^{-c_0},$$

and $C_{\tilde{\epsilon}}, C_0, c_0 > 0$ are absolute constants. Since $\mathcal{A}$ holds with probability tending to 1 as $T \to \infty$, we condition the remainder of the proof on $\mathcal{A}$.

From $\mathcal{A}$, Assumption 2 and the fact that $C_{\mathrm{SNR}}$ is a sufficiently large constant, we have for all $k \in [K]$ that $\eta_k \in [\widetilde{\eta}_{k-1}, \widetilde{\eta}_{k+1}]$,

$$\eta_k - \widetilde{\eta}_{k-1} \geq \eta_k - \eta_{k-1} - |\eta_{k-1} - \widetilde{\eta}_{k-1}| \geq \Delta - C_\epsilon \frac{\log(T)}{\kappa^2} \geq \Delta - \Delta/6 = 5\Delta/6, \quad (31)$$

and

$$\widetilde{\eta}_{k+1} - \eta_k \geq \eta_{k+1} - \eta_k - |\eta_{k+1} - \widetilde{\eta}_{k+1}| \geq \Delta - C_\epsilon \frac{\log(T)}{\kappa^2} \geq \Delta - \Delta/6 = 5\Delta/6. \quad (32)$$

Similarly, we can derive that for any $k \in [K+1]$

$$\widetilde{\eta}_k - \widetilde{\eta}_{k-1} \geq 2\Delta/3. \quad (33)$$

As a result, the working interval

$$(\tilde{s}_k, \tilde{e}_k] = (\widetilde{\eta}_{k-1} + (\widetilde{\eta}_k - \widetilde{\eta}_{k-1})/2, \, \widetilde{\eta}_{k+1} - (\widetilde{\eta}_{k+1} - \widetilde{\eta}_k)/2],$$

contains exactly one change point $\eta_k$.

Next, by Theorem 4 and Lemma 5 in Wang et al. (2025), and Assumption 1 $(i)$ and $(iii)$, for any $k \in [K+1]$, we have

$$\left\| \widehat{\mathbf{P}}^{\widetilde{\eta}_{k-1}, \widetilde{\eta}_k} - \mathbf{P}^{\widetilde{\eta}_{k-1}, \widetilde{\eta}_k} \right\|_{\mathrm{F}} = O_p\left( \sqrt{\frac{(d^2 m_{\widetilde{\eta}_{k-1}, \widetilde{\eta}_k} + nd + L m_{\widetilde{\eta}_{k-1}, \widetilde{\eta}_k}) \log(T)}{\widetilde{\eta}_k - \widetilde{\eta}_{k-1}}} \right),$$

for some absolute constant $C_1 > 0$. For any $k \in [K+1]$, by (31) and (32), each interval $(\widetilde{\eta}_{k-1}, \widetilde{\eta}_k)$ contains at most two true change points $\eta_k - 1$ and $\eta_k$. Consequently, we have that

$$m_{\widetilde{\eta}_{k-1}, \widetilde{\eta}_k} \leq m_{k-1} + m_k + m_{k+1} \leq 3 m_{\max}.$$

Thus, it holds that

$$\left\|\widehat{\mathbf{P}}^{\widetilde{\eta}_{k-1},\widetilde{\eta}_k} - \mathbf{P}^{\widetilde{\eta}_{k-1},\widetilde{\eta}_k}\right\|_{\mathrm{F}} \le C_2 \sqrt{\frac{(d^2 m_{\max} + nd + Lm_{\max})\log(T)}{\widetilde{\eta}_k - \widetilde{\eta}_{k-1}}}, \quad \forall k \in [K+1], \quad (34)$$

with an absolute constant $C_2 > 0$.

**Step 2.** Characterization of bias. From (31) and (32), for any $k \in [K]$, the interval $(\widetilde{\eta}_{k-1}, \widetilde{\eta}_{k+1})$ may contain one, two or three change points. One example that contains three change points is illustrated in the figure below. We analyze the biases in this case. The analyses for the other scenarios are similar but simpler and therefore omitted.

In the following, we analyze three types of bias terms. Denote $\alpha_T = \log(T)$, then $\alpha_T \to \infty$ as $T \to \infty$. Observe that

$$\begin{aligned}
&\left\|\mathbf{P}(\eta_k) - \mathbf{P}^{\widetilde{\eta}_{k-1},\widetilde{\eta}_k}\right\|_{\mathrm{F}} \\
=&\left\|\mathbf{P}(\eta_k) - \frac{\eta_{k-1} - \widetilde{\eta}_{k-1}}{\widetilde{\eta}_k - \widetilde{\eta}_{k-1}}\mathbf{P}(\eta_{k-1}) - \frac{\eta_k - \eta_{k-1}}{\widetilde{\eta}_k - \widetilde{\eta}_{k-1}}\mathbf{P}(\eta_k) - \frac{\widetilde{\eta}_k - \eta_k}{\widetilde{\eta}_k - \widetilde{\eta}_{k-1}}\mathbf{P}(\eta_{k+1})\right\|_{\mathrm{F}} \\
\le&\frac{\eta_{k-1} - \widetilde{\eta}_{k-1}}{\widetilde{\eta}_k - \widetilde{\eta}_{k-1}}\left\|\mathbf{P}(\eta_k) - \mathbf{P}(\eta_{k-1})\right\|_{\mathrm{F}} + \frac{\widetilde{\eta}_k - \eta_k}{\widetilde{\eta}_k - \widetilde{\eta}_{k-1}}\left\|\mathbf{P}(\eta_{k+1}) - \mathbf{P}(\eta_k)\right\|_{\mathrm{F}} \\
\le&\frac{3C_\epsilon \log(T)}{2\Delta\kappa_{k-1}^2}\kappa_{k-1} + \frac{3C_\epsilon \log(T)}{2\Delta\kappa_k^2}\kappa_k \le \alpha_T^{-1}\kappa_k,
\end{aligned} \quad (35)$$

where the second inequity follows from the event $\mathcal{A}$ and (33), and the last inequality follows from Assumption 2 and the fact that $C_{\mathrm{SNR}}$ is a large enough constant. Similarly, we have that

$$\begin{aligned}
\left\|\mathbf{P}(\eta_{k+1}) - \mathbf{P}^{\widetilde{\eta}_k,\widetilde{\eta}_{k+1}}\right\|_{\mathrm{F}} =&\left\|\mathbf{P}(\eta_{k+1}) - \frac{\eta_{k+1} - \widetilde{\eta}_k}{\widetilde{\eta}_{k+1} - \widetilde{\eta}_k}\mathbf{P}(\eta_{k+1}) - \frac{\widetilde{\eta}_{k+1} - \eta_{k+1}}{\widetilde{\eta}_{k+1} - \widetilde{\eta}_k}\mathbf{P}(\eta_{k+2})\right\|_{\mathrm{F}} \\
=&\frac{\widetilde{\eta}_{k+1} - \eta_{k+1}}{\widetilde{\eta}_{k+1} - \widetilde{\eta}_k}\left\|\mathbf{P}(\eta_{k+1}) - \mathbf{P}(\eta_{k+2})\right\|_{\mathrm{F}} \\
\le&\frac{3C_\epsilon \log(T)}{2\Delta\kappa_{k+1}^2}\kappa_{k+1} \le \alpha_T^{-1}\kappa_k,
\end{aligned} \quad (36)$$

and

$$\begin{aligned}
&\left\|\mathbf{P}(\eta_{k+1}) - \mathbf{P}^{\widetilde{\eta}_{k-1},\widetilde{\eta}_k}\right\|_{\mathrm{F}} \\
=&\left\|\mathbf{P}(\eta_{k+1}) - \frac{\eta_{k-1} - \widetilde{\eta}_{k-1}}{\widetilde{\eta}_k - \widetilde{\eta}_{k-1}}\mathbf{P}(\eta_{k-1}) - \frac{\eta_k - \eta_{k-1}}{\widetilde{\eta}_k - \widetilde{\eta}_{k-1}}\mathbf{P}(\eta_k) - \frac{\widetilde{\eta}_k - \eta_k}{\widetilde{\eta}_k - \widetilde{\eta}_{k-1}}\mathbf{P}(\eta_{k+1})\right\|_{\mathrm{F}} \\
\le&\frac{\eta_{k-1} - \widetilde{\eta}_{k-1}}{\widetilde{\eta}_k - \widetilde{\eta}_{k-1}}\left\|\mathbf{P}(\eta_{k+1}) - \mathbf{P}(\eta_{k-1})\right\|_{\mathrm{F}} + \frac{\eta_k - \eta_{k-1}}{\widetilde{\eta}_k - \widetilde{\eta}_{k-1}}\left\|\mathbf{P}(\eta_{k+1}) - \mathbf{P}(\eta_k)\right\|_{\mathrm{F}} \\
\le&\frac{3C_\epsilon \log(T)}{2\Delta\kappa_{k-1}^2}(\kappa_{k-1} + \kappa_k) + \kappa_k \le \alpha_T^{-1}\kappa_k + \kappa_k \le C_3\kappa_k,
\end{aligned} \quad (37)$$

for some constant $C_3 > 0$.

**Step 3.** Uniform tightness of $\kappa_k^2|\widehat{\eta}_k - \eta_k|$. In this step, we show that $\kappa_k^2|\widehat{\eta}_k - \eta_k| = O_p(1)$. Let $r = \widehat{\eta}_k - \eta_k$ and without loss of generality, assume $r \ge 0$. Our goal is to establish that

$$r\kappa_k^2 = O_p(1)$$

If $r\kappa_k^2 < 1$, the conclusion holds trivially. Thus, for the remainder of the argument, we assume that $r\kappa_k^2 \ge 1$. Since $\widehat{\eta}_k = \eta_k + r$, it follows that

$$\mathcal{Q}_k(\eta_k + r) - \mathcal{Q}_k(\eta_k) \le 0.$$

Now observe that

$$\mathcal{Q}_k(\eta_k + r) - \mathcal{Q}_k(\eta_k) = \sum_{t=\eta_k+1}^{\eta_k+r} \left\| \mathbf{A}(t) - \widehat{\mathbf{P}}^{\widetilde{\eta}_{k-1},\widetilde{\eta}_k} \right\|_{\mathrm{F}}^2 - \left\| \mathbf{A}(t) - \widehat{\mathbf{P}}^{\widetilde{\eta}_k,\widetilde{\eta}_{k+1}} \right\|_{\mathrm{F}}^2$$

$$= \sum_{t=\eta_k+1}^{\eta_k+r} \left\{ \left\| \mathbf{A}(t) - \widehat{\mathbf{P}}^{\widetilde{\eta}_{k-1},\widetilde{\eta}_k} \right\|_{\mathrm{F}}^2 - \left\| \mathbf{A}(t) - \mathbf{P}^{\widetilde{\eta}_{k-1},\widetilde{\eta}_k} \right\|_{\mathrm{F}}^2 \right\}$$

$$- \sum_{t=\eta_k+1}^{\eta_k+r} \left\{ \left\| \mathbf{A}(t) - \widehat{\mathbf{P}}^{\widetilde{\eta}_k,\widetilde{\eta}_{k+1}} \right\|_{\mathrm{F}}^2 - \left\| \mathbf{A}(t) - \mathbf{P}^{\widetilde{\eta}_k,\widetilde{\eta}_{k+1}} \right\|_{\mathrm{F}}^2 \right\}$$

$$+ \sum_{t=\eta_k+1}^{\eta_k+r} \left\{ \left\| \mathbf{A}(t) - \mathbf{P}^{\widetilde{\eta}_{k-1},\widetilde{\eta}_k} \right\|_{\mathrm{F}}^2 - \left\| \mathbf{A}(t) - \mathbf{P}(\eta_k) \right\|_{\mathrm{F}}^2 \right\}$$

$$- \sum_{t=\eta_k+1}^{\eta_k+r} \left\{ \left\| \mathbf{A}(t) - \mathbf{P}^{\widetilde{\eta}_k,\widetilde{\eta}_{k+1}} \right\|_{\mathrm{F}}^2 - \left\| \mathbf{A}(t) - \mathbf{P}(\eta_{k+1}) \right\|_{\mathrm{F}}^2 \right\}$$

$$+ \sum_{t=\eta_k+1}^{\eta_k+r} \left\{ \left\| \mathbf{A}(t) - \mathbf{P}(\eta_k) \right\|_{\mathrm{F}}^2 - \left\| \mathbf{A}(t) - \mathbf{P}(\eta_{k+1}) \right\|_{\mathrm{F}}^2 \right\}$$

$$= I - II + III - IV + V.$$

Therefore, we have that

$$V \leq -I + II - III + IV \leq |I| + |II| + |III| + |IV|. \tag{38}$$

**Step 3.1.** Order of magnitude of $I$. We start by analyzing the term

$$I = \sum_{t=\eta_k+1}^{\eta_k+r} \left\{ \left\| \mathbf{A}(t) - \widehat{\mathbf{P}}^{\widetilde{\eta}_{k-1},\widetilde{\eta}_k} \right\|_{\mathrm{F}}^2 - \left\| \mathbf{A}(t) - \mathbf{P}^{\widetilde{\eta}_{k-1},\widetilde{\eta}_k} \right\|_{\mathrm{F}}^2 \right\}$$

$$= \sum_{t=\eta_k+1}^{\eta_k+r} \left\| \widehat{\mathbf{P}}^{\widetilde{\eta}_{k-1},\widetilde{\eta}_k} - \mathbf{P}^{\widetilde{\eta}_{k-1},\widetilde{\eta}_k} \right\|_{\mathrm{F}}^2 - 2 \sum_{t=\eta_k+1}^{\eta_k+r} \left\langle \mathbf{A}(t) - \mathbf{P}^{\widetilde{\eta}_{k-1},\widetilde{\eta}_k}, \widehat{\mathbf{P}}^{\widetilde{\eta}_{k-1},\widetilde{\eta}_k} - \mathbf{P}^{\widetilde{\eta}_{k-1},\widetilde{\eta}_k} \right\rangle$$

$$= \sum_{t=\eta_k+1}^{\eta_k+r} \left\| \widehat{\mathbf{P}}^{\widetilde{\eta}_{k-1},\widetilde{\eta}_k} - \mathbf{P}^{\widetilde{\eta}_{k-1},\widetilde{\eta}_k} \right\|_{\mathrm{F}}^2 - 2 \sum_{t=\eta_k+1}^{\eta_k+r} \left\langle \mathbf{A}(t) - \mathbf{P}(\eta_{k+1}), \widehat{\mathbf{P}}^{\widetilde{\eta}_{k-1},\widetilde{\eta}_k} - \mathbf{P}^{\widetilde{\eta}_{k-1},\widetilde{\eta}_k} \right\rangle$$

$$- 2 \sum_{t=\eta_k+1}^{\eta_k+r} \left\langle \mathbf{P}(\eta_{k+1}) - \mathbf{P}^{\widetilde{\eta}_{k-1},\widetilde{\eta}_k}, \widehat{\mathbf{P}}^{\widetilde{\eta}_{k-1},\widetilde{\eta}_k} - \mathbf{P}^{\widetilde{\eta}_{k-1},\widetilde{\eta}_k} \right\rangle$$

$$= I.1 - 2I.2 - 2I.3. \tag{39}$$

By (34), we have that

$$\left\| \widehat{\mathbf{P}}^{\widetilde{\eta}_{k-1},\widetilde{\eta}_k} - \mathbf{P}^{\widetilde{\eta}_{k-1},\widetilde{\eta}_k} \right\|_{\mathrm{F}}^2 \leq C_2^2 \frac{(d^2 m_{\max} + nd + L m_{\max}) \log(T)}{\widetilde{\eta}_k - \widetilde{\eta}_{k-1}}$$

$$\leq 3 C_2^2 \frac{(d^2 m_{\max} + nd + L m_{\max}) \log(T)}{2\Delta} \leq \alpha_T^{-1} \kappa_k^2, \tag{40}$$

where the second inequality is by (33) and the last inequality follows from Assumption 2 and the fact that $C_{\mathrm{SNR}}$ is a sufficiently large constant. This yields that

$$|I.1| = \sum_{t=\eta_k+1}^{\eta_k+r} \left\| \widehat{\mathbf{P}}^{\widetilde{\eta}_{k-1},\widetilde{\eta}_k} - \mathbf{P}^{\widetilde{\eta}_{k-1},\widetilde{\eta}_k} \right\|_{\mathrm{F}}^2 = O_p(r \alpha_T^{-1} \kappa_k^2), \tag{41}$$

We now turn to the term $I.2$ in (39). By Lemma 5 and (40), we obtain that

$$|I.2| = O_p\left( r^{1/2} \left\| \widehat{\mathbf{P}}^{\widetilde{\eta}_{k-1},\widetilde{\eta}_k} - \mathbf{P}^{\widetilde{\eta}_{k-1},\widetilde{\eta}_k} \right\|_{\mathrm{F}} \right) = O_p(r^{1/2} \alpha_T^{-1/2} \kappa_k). \tag{42}$$

Next, by the Cauchy–Schwarz inequality, we derive that

$$|I.3| \leq r \big\| \mathbf{P}(\eta_{k+1}) - \mathbf{P}^{\widetilde{\eta}_{k-1}, \widetilde{\eta}_k} \big\|_{\mathrm{F}} \big\| \widehat{\mathbf{P}}^{\widetilde{\eta}_{k-1}, \widetilde{\eta}_k} - \mathbf{P}^{\widetilde{\eta}_{k-1}, \widetilde{\eta}_k} \big\|_{\mathrm{F}} = O_p(r \alpha_T^{-1/2} \kappa_k^2), \qquad (43)$$

where the last inequality follows from (37) and (40).

Combining (39), (41), (42) and (43), we conclude that

$$(I) = o_p(r \kappa_k^2 + r^{1/2} \kappa_k). \qquad (44)$$

**Step 3.2.** Order of magnitude of $II$. We now analyze the term

$$
\begin{aligned}
II &= \sum_{t=\eta_k+1}^{\eta_k+r} \left\{ \big\| \mathbf{A}(t) - \widehat{\mathbf{P}}^{\widetilde{\eta}_k, \widetilde{\eta}_{k+1}} \big\|_{\mathrm{F}}^2 - \big\| \mathbf{A}(t) - \mathbf{P}^{\widetilde{\eta}_k, \widetilde{\eta}_{k+1}} \big\|_{\mathrm{F}}^2 \right\} \\
&= \sum_{t=\eta_k+1}^{\eta_k+r} \big\| \widehat{\mathbf{P}}^{\widetilde{\eta}_k, \widetilde{\eta}_{k+1}} - \mathbf{P}^{\widetilde{\eta}_k, \widetilde{\eta}_{k+1}} \big\|_{\mathrm{F}}^2 - 2 \sum_{t=\eta_k+1}^{\eta_k+r} \left\langle \mathbf{A}(t) - \mathbf{P}^{\widetilde{\eta}_k, \widetilde{\eta}_{k+1}}, \widehat{\mathbf{P}}^{\widetilde{\eta}_k, \widetilde{\eta}_{k+1}} - \mathbf{P}^{\widetilde{\eta}_k, \widetilde{\eta}_{k+1}} \right\rangle \\
&= \sum_{t=\eta_k+1}^{\eta_k+r} \big\| \widehat{\mathbf{P}}^{\widetilde{\eta}_k, \widetilde{\eta}_{k+1}} - \mathbf{P}^{\widetilde{\eta}_k, \widetilde{\eta}_{k+1}} \big\|_{\mathrm{F}}^2 - 2 \sum_{t=\eta_k+1}^{\eta_k+r} \left\langle \mathbf{A}(t) - \mathbf{P}(\eta_{k+1}), \widehat{\mathbf{P}}^{\widetilde{\eta}_k, \widetilde{\eta}_{k+1}} - \mathbf{P}^{\widetilde{\eta}_k, \widetilde{\eta}_{k+1}} \right\rangle \\
&\quad - 2 \sum_{t=\eta_k+1}^{\eta_k+r} \left\langle \mathbf{P}(\eta_{k+1}) - \mathbf{P}^{\widetilde{\eta}_k, \widetilde{\eta}_{k+1}}, \widehat{\mathbf{P}}^{\widetilde{\eta}_k, \widetilde{\eta}_{k+1}} - \mathbf{P}^{\widetilde{\eta}_k, \widetilde{\eta}_{k+1}} \right\rangle \\
&= II.1 - 2II.2 - 2II.3. \qquad (45)
\end{aligned}
$$

By (34), we have that

$$
\begin{aligned}
\big\| \widehat{\mathbf{P}}^{\widetilde{\eta}_k, \widetilde{\eta}_{k+1}} - \mathbf{P}^{\widetilde{\eta}_k, \widetilde{\eta}_{k+1}} \big\|_{\mathrm{F}}^2 &\leq C_2^2 \frac{(d^2 m_{\max} + nd + L m_{\max}) \log(T)}{\widetilde{\eta}_{k+1} - \widetilde{\eta}_k} \\
&\leq 3 C_2^2 \frac{(d^2 m_{\max} + nd + L m_{\max}) \log(T)}{2\Delta} \leq \alpha_T^{-1} \kappa_k^2, \qquad (46)
\end{aligned}
$$

where the second inequality follows from (33) and the last inequality follows from Assumption 2 and the fact that $C_{\mathrm{SNR}}$ is a sufficiently large constant. It then follows that

$$|II.1| = \sum_{t=\eta_k+1}^{\eta_k+r} \big\| \widehat{\mathbf{P}}^{\widetilde{\eta}_k, \widetilde{\eta}_{k+1}} - \mathbf{P}^{\widetilde{\eta}_k, \widetilde{\eta}_{k+1}} \big\|_{\mathrm{F}}^2 = O_p(r \alpha_T^{-1} \kappa_k^2), \qquad (47)$$

To control $II.2$, by Lemma 5 and (46), we obtain that

$$|II.2| = O_p\left( r^{1/2} \big\| \widehat{\mathbf{P}}^{\widetilde{\eta}_k, \widetilde{\eta}_{k+1}} - \mathbf{P}^{\widetilde{\eta}_k, \widetilde{\eta}_{k+1}} \big\|_{\mathrm{F}} \right) = O_p(r^{1/2} \alpha_T^{-1/2} \kappa_k). \qquad (48)$$

Next, by the Cauchy–Schwarz inequality, we derive that

$$|II.3| \leq r \big\| \mathbf{P}(\eta_{k+1}) - \mathbf{P}^{\widetilde{\eta}_k, \widetilde{\eta}_{k+1}} \big\|_{\mathrm{F}} \big\| \widehat{\mathbf{P}}^{\widetilde{\eta}_{k-1}, \widetilde{\eta}_k} - \mathbf{P}^{\widetilde{\eta}_k, \widetilde{\eta}_{k+1}} \big\|_{\mathrm{F}} = O_p(r \alpha_T^{-3/2} \kappa_k^2), \qquad (49)$$

where the last inequality follows from (36) and (46).

Combining (45), (47), (48) and (49), we conclude that

$$|II| = o_p(r \kappa_k^2 + r^{1/2} \kappa_k). \qquad (50)$$

**Step 3.3.** Order of magnitude of $III$. We now analyze the term

$$
\begin{aligned}
III &= \sum_{t=\eta_k+1}^{\eta_k+r} \left\{ \left\| \mathbf{A}(t) - \mathbf{P}^{\widetilde{\eta}_{k-1},\widetilde{\eta}_k} \right\|_{\mathrm{F}}^2 - \left\| \mathbf{A}(t) - \mathbf{P}(\eta_k) \right\|_{\mathrm{F}}^2 \right\} \\
&= \sum_{t=\eta_k+1}^{\eta_k+r} \left\| \mathbf{P}(\eta_k) - \mathbf{P}^{\widetilde{\eta}_{k-1},\widetilde{\eta}_k} \right\|_{\mathrm{F}}^2 - 2 \sum_{t=\eta_k+1}^{\eta_k+r} \left\langle \mathbf{A}(t) - \mathbf{P}(\eta_k), \mathbf{P}^{\widetilde{\eta}_{k-1},\widetilde{\eta}_k} - \mathbf{P}(\eta_k) \right\rangle \\
&= \sum_{t=\eta_k+1}^{\eta_k+r} \left\| \mathbf{P}(\eta_k) - \mathbf{P}^{\widetilde{\eta}_{k-1},\widetilde{\eta}_k} \right\|_{\mathrm{F}}^2 - 2 \sum_{t=\eta_k+1}^{\eta_k+r} \left\langle \mathbf{P}(\eta_{k+1}) - \mathbf{P}(\eta_k), \mathbf{P}^{\widetilde{\eta}_{k-1},\widetilde{\eta}_k} - \mathbf{P}(\eta_k) \right\rangle \\
&\quad - 2 \sum_{t=\eta_k+1}^{\eta_k+r} \left\langle \mathbf{A}(t) - \mathbf{P}(\eta_{k+1}), \mathbf{P}^{\widetilde{\eta}_{k-1},\widetilde{\eta}_k} - \mathbf{P}(\eta_k) \right\rangle \\
&= III.1 - 2III.2 - 2III.3.
\end{aligned}
\tag{51}
$$

From (35), we obtain that

$$
|III.1| = O_p(r\alpha_T^{-2}\kappa_k^2). \tag{52}
$$

Using the Cauchy–Schwarz inequality and again (35), we have that

$$
|III.2| \le r \left\| \mathbf{P}(\eta_{k+1}) - \mathbf{P}(\eta_k) \right\|_{\mathrm{F}} \left\| \mathbf{P}^{\widetilde{\eta}_{k-1},\widetilde{\eta}_k} - \mathbf{P}(\eta_k) \right\|_{\mathrm{F}} = O_p(r\alpha_T^{-1}\kappa_k^2). \tag{53}
$$

To bound $III.3$, by Lemma 5 and (35), we get that

$$
|III.3| = O_p\left( r^{1/2} \left\| \mathbf{P}^{\widetilde{\eta}_{k-1},\widetilde{\eta}_k} - \mathbf{P}(\eta_k) \right\|_{\mathrm{F}} \right) = O_p(r^{1/2}\alpha_T^{-1/2}\kappa_k). \tag{54}
$$

Combining (51), (52), (53) and (54), we conclude that

$$
|III| = o_p(r\kappa_k^2 + r^{1/2}\kappa_k). \tag{55}
$$

**Step 3.4.** Order of magnitude of $IV$. Consider the term

$$
\begin{aligned}
IV &= \sum_{t=\eta_k+1}^{\eta_k+r} \left\{ \left\| \mathbf{A}(t) - \mathbf{P}^{\widetilde{\eta}_k,\widetilde{\eta}_{k+1}} \right\|_{\mathrm{F}}^2 - \left\| \mathbf{A}(t) - \mathbf{P}(\eta_{k+1}) \right\|_{\mathrm{F}}^2 \right\} \\
&= \sum_{t=\eta_k+1}^{\eta_k+r} \left\| \mathbf{P}(\eta_{k+1}) - \mathbf{P}^{\widetilde{\eta}_k,\widetilde{\eta}_{k+1}} \right\|_{\mathrm{F}}^2 - 2 \sum_{t=\eta_k+1}^{\eta_k+r} \left\langle \mathbf{A}(t) - \mathbf{P}(\eta_{k+1}), \mathbf{P}^{\widetilde{\eta}_k,\widetilde{\eta}_{k+1}} - \mathbf{P}(\eta_{k+1}) \right\rangle \\
&= IV.1 - 2IV.2.
\end{aligned}
\tag{56}
$$

By (36), we derive that

$$
|IV.1| = O_p(r\alpha_T^{-2}\kappa_k^2). \tag{57}
$$

By Lemma 5 and (36), we have that

$$
|IV.2| = O_p\left( r^{1/2} \left\| \mathbf{P}(\eta_{k+1}) - \mathbf{P}^{\widetilde{\eta}_k,\widetilde{\eta}_{k+1}} \right\|_{\mathrm{F}} \right) = O_p(r^{1/2}\alpha_T^{-1/2}\kappa_k). \tag{58}
$$

Combining (56), (57) and (58), we conclude that

$$
|IV| = o_p(r\kappa_k^2 + r^{1/2}\kappa_k). \tag{59}
$$

**Step 3.5.** Order of magnitude of $V$. We now analyze the final term

$$
\begin{aligned}
V &= \sum_{t=\eta_k+1}^{\eta_k+r} \left\| \mathbf{A}(t) - \mathbf{P}(\eta_k) \right\|_{\mathrm{F}}^2 - \left\| \mathbf{A}(t) - \mathbf{P}(\eta_{k+1}) \right\|_{\mathrm{F}}^2 \\
&= \sum_{t=\eta_k+1}^{\eta_k+r} \left\| \mathbf{P}(\eta_k) - \mathbf{P}(\eta_{k+1}) \right\|_{\mathrm{F}}^2 - 2 \sum_{t=\eta_k+1}^{\eta_k+r} \left\langle \mathbf{A}(t) - \mathbf{P}(\eta_{k+1}), \mathbf{P}(\eta_k) - \mathbf{P}(\eta_{k+1}) \right\rangle \\
&= r\kappa_k^2 - 2V.1
\end{aligned}
\tag{60}
$$

Using Lemma 5, we obtain that

$$|V.1| = O_p\left(r^{1/2}\big\|\mathbf{P}(\eta_k) - \mathbf{P}(\eta_{k+1})\big\|_{\mathrm{F}}\right) = O_p(r^{1/2}\kappa_k). \tag{61}$$

**Step 3.6:** Combining (38), (44), (50), (55), (59), (60) and (61) we have for all $r\kappa_k^2 \geq 1$ that

$$r\kappa_k^2 = O_p(1).$$

**Step 4.** Limiting Distributions. For any $t \in (\tilde{s}_k, \tilde{e}_k)$, define

$$\widetilde{\mathcal{Q}}_k(t) = \sum_{u=\tilde{s}_k+1}^{t} \|\mathbf{A}(u) - \mathbf{P}(\eta_k)\|_{\mathrm{F}}^2 + \sum_{u=t+1}^{\tilde{e}_k} \|\mathbf{A}(u) - \mathbf{P}(\eta_{k+1})\|_{\mathrm{F}}^2.$$

Note that the term $V$ defined in (38) satisfies

$$V = \widetilde{\mathcal{Q}}_k(\eta_k + r) - \widetilde{\mathcal{Q}}_k(\eta_k),$$

and hence by (38), (44), (50), (55), (59) and $r\kappa_k^2 = O_p(1)$, we have that

$$\left|\mathcal{Q}_k(\eta_k + r) - \mathcal{Q}_k(\eta_k) - \left\{\widetilde{\mathcal{Q}}_k(\eta_k + r) - \widetilde{\mathcal{Q}}_k(\eta_k)\right\}\right| \leq |I| + |II| + |III| + |IV| \xrightarrow{p} 0.$$

Therefore, by Slutsky's theorem, it suffices to derive the limiting distributions of $\widetilde{\mathcal{Q}}_k(\eta_k + r) - \widetilde{\mathcal{Q}}_k(\eta_k)$ as $T \to \infty$. We consider the two scenarios for $\kappa_k$.

**Non-vanishing scenario.** Suppose $\kappa_k \to \rho_k$, as $T \to \infty$, with $\rho_k > 0$ being an absolute constant. For $r < 0$, we have that

$$\begin{aligned}
\widetilde{\mathcal{Q}}_k(\eta_k + r) - \widetilde{\mathcal{Q}}_k(\eta_k) &= \sum_{t=\eta_k+r+1}^{\eta_k} \left\{\big\|\mathbf{A}(t) - \mathbf{P}(\eta_{k+1})\big\|_{\mathrm{F}}^2 - \big\|\mathbf{A}(t) - \mathbf{P}(\eta_k)\big\|_{\mathrm{F}}^2\right\} \\
&= \sum_{t=\eta_k+r+1}^{\eta_k} \big\|\mathbf{P}(\eta_k) - \mathbf{P}(\eta_{k+1})\big\|_{\mathrm{F}}^2 \\
&\quad - 2\sum_{t=\eta_k+1}^{\eta_k+r} \big\langle\mathbf{A}(t) - \mathbf{P}(\eta_k), \mathbf{P}(\eta_{k+1}) - \mathbf{P}(\eta_k)\big\rangle \\
&\xrightarrow{\mathcal{D}} -r\rho_k^2 - 2\rho_k \sum_{t=r+1}^{0} \big\langle\mathbf{\Psi}_k, \mathbf{E}_k(t)\big\rangle,
\end{aligned} \tag{62}$$

with $\mathbf{\Psi}_k$ defined in Model 1, and for any $k \in [K+1]$ and $t \in \mathbb{Z}$, $\mathbf{E}_k(t) = \mathbf{A}_k(t) - \mathbf{P}(\eta_k)$ with $\{\mathbf{A}_k(t)\}_{t\in\mathbb{Z}} \overset{\text{i.i.d.}}{\sim} \mathrm{MRDPG}(\{X_i\}_{i=1}^n, \{W_{(l)}(\eta_k)\}_{l\in[L]})$.

For $r > 0$, we have that when $T \to \infty$,

$$\begin{aligned}
\widetilde{\mathcal{Q}}_k(\eta_k + r) - \widetilde{\mathcal{Q}}_k(\eta_k) &= \sum_{t=\eta_k+1}^{\eta_k+r} \left\{\big\|\mathbf{A}(t) - \mathbf{P}(\eta_k)\big\|_{\mathrm{F}}^2 - \big\|\mathbf{A}(t) - \mathbf{P}(\eta_{k+1})\big\|_{\mathrm{F}}^2\right\} \\
&= \sum_{t=\eta_k+1}^{\eta_k+r} \big\|\mathbf{P}(\eta_k) - \mathbf{P}(\eta_{k+1})\big\|_{\mathrm{F}}^2 \\
&\quad + 2\sum_{t=\eta_k+1}^{\eta_k+r} \big\langle\mathbf{A}(t) - \mathbf{P}(\eta_{k+1}), \mathbf{P}(\eta_{k+1}) - \mathbf{P}(\eta_k)\big\rangle \\
&\xrightarrow{\mathcal{D}} r\rho_k^2 + 2\rho_k \sum_{t=1}^{r} \big\langle\mathbf{\Psi}_k, \mathbf{E}_{k+1}(t)\big\rangle.
\end{aligned} \tag{63}$$

By Slutsky's theorem and the argmin continuous mapping theorem (see e.g. Theorem 3.2.2 in Wellner et al., 2013), we obtain

$$\widehat{\eta}_k - \eta_k \xrightarrow{\mathcal{D}} \arg\min \mathcal{P}_k(r),$$

which completes the proof of part Theorem 3.

**Vanishing scenario.** Let $m = \kappa_k^{-2}$, noting that $m \to \infty$ as $T \to \infty$. For $r > 0$, we have that

$$\widetilde{\mathcal{Q}}_k(\eta_k + rm) - \widetilde{\mathcal{Q}}_k(\eta_k)$$

$$= \sum_{t=\eta_k+1}^{\eta_k+rm} \left\{ \left\| \mathbf{A}(t) - \mathbf{P}(\eta_k) \right\|_{\mathrm{F}}^2 - \left\| \mathbf{A}(t) - \mathbf{P}(\eta_{k+1}) \right\|_{\mathrm{F}}^2 \right\}$$

$$= \sum_{t=\eta_k+1}^{\eta_k+rm} \left\| \mathbf{P}(\eta_k) - \mathbf{P}(\eta_{k+1}) \right\|_{\mathrm{F}}^2 + 2 \sum_{t=\eta_k+1}^{\eta_k+rm} \left\langle \mathbf{A}(t) - \mathbf{P}(\eta_{k+1}), \mathbf{P}(\eta_{k+1}) - \mathbf{P}(\eta_k) \right\rangle$$

$$= r + \frac{2}{\sqrt{m}} \sum_{t=\eta_k+1}^{\eta_k+rm} \left\langle \mathbf{A}(t) - \mathbf{P}(\eta_{k+1}), \mathbf{\Psi}_k \right\rangle.$$

By the functional central limit theorem, we have that when $T \to \infty$,

$$\frac{1}{\sqrt{m}} \sum_{t=\eta_k+1}^{\eta_k+rm} \left\langle \mathbf{A}(t) - \mathbf{P}(\eta_{k+1}), \mathbf{\Psi}_k \right\rangle \xrightarrow{\mathcal{D}} \sigma_{k,k+1} \mathbb{B}_1(r),$$

where $\mathbb{B}_1(r)$ is a standard Brownian motion and for any $k \in [K]$ and $k' \in \{k, k+1\}$, $\sigma_{k,k'}^2 = \mathrm{Var}\left( \langle \mathbf{\Psi}_k, \mathbf{E}_{k'}(1) \rangle \right)$. Consequently, as $T \to \infty$

$$\widetilde{\mathcal{Q}}_k(\eta_k + rm) - \widetilde{\mathcal{Q}}_k(\eta_k) \xrightarrow{\mathcal{D}} r + 2\sigma_{k,k+1} \mathbb{B}_1(r).$$

Similarly, for $r < 0$, we have that when $T \to \infty$

$$\widetilde{\mathcal{Q}}_k(\eta_k + rm) - \widetilde{\mathcal{Q}}_k(\eta_k) \xrightarrow{D} -r + 2\sigma_{k,k} \mathbb{B}_2(-r),$$

where $\mathbb{B}_2(r)$ is a standard Brownian motion. Applying Slutsky's theorem and the argmin continuous mapping theorem (see e.g. Theorem 3.2.2 in Wellner et al., 2013), we conclude that

$$\kappa_k^2(\widehat{\eta}_k - \eta_k) \xrightarrow{\mathcal{D}} \arg\min \mathcal{P}_k'(r),$$

which completes the proof of Theorem 2.

$\square$

# G  ADDITIONAL DETAILS AND RESULTS IN SECTION 4

All experiments were run on a CPU with 16GB RAM. For each synthetic scenario with node size $n = 100$, number of layers $L = 4$ and time span $T = 200$, the compute time is about 10 hours to localize the change points and to construct the confidence intervals over 100 Monte Carlo trials. For each real data experiment, the computation time is approximately 15 minutes to perform change point localization and confidence interval construction.

## G.1  ADDITIONAL RESULTS IN SECTION 4.1

**Sensitivity analysis:** We examine the sensitivity of Algorithm 1 to the threshold constant $c_{\tau,1}$. Guided by Theorem 1, the threshold is set as $\tau = c_{\tau,1} n \sqrt{L} \log^{3/2}(T)$. The constant $c_{\tau,1}$ is calibrated by evaluating the false positive rate under an MSBM with no change points (four equally sized communities; see Section 4.1). We find that $c_{\tau,1} = 0.1$ yields a false detection rate of about 1%, demonstrating effective control. Smaller values ($c_{\tau,1} \in 0.05, 0.08$) increased false detections, while larger values ($c_{\tau,1} \in 0.12, 0.15, 0.20$) reduced detection power.

Table 5: Means of evaluation metrics for dynamic networks simulated from Scenario 1, varying $c_{\tau,1}$.

| $n$ | $c_{\tau,1}$ | $|\widehat{K} - K| \downarrow$ | $d(\widehat{\mathcal{C}}|\mathcal{C}) \downarrow$ | $d(\mathcal{C}|\widehat{\mathcal{C}}) \downarrow$ | $C(\mathcal{G}, \mathcal{G}') \uparrow$ |
|---|---|---|---|---|---|
|    | 0.25 | 0.00 | 0.00 | 0.00 | 100% |
|    | 0.20 | 0.00 | 0.00 | 0.00 | 100% |
|    | 0.15 | 0.00 | 0.00 | 0.00 | 100% |
| 50 | 0.12 | 0.00 | 0.00 | 0.00 | 100% |
|    | 0.10 | 0.01 | 0.00 | 0.42 | 99.86% |
|    | 0.08 | 0.25 | 0.00 | 6.68 | 97.80% |
|    | 0.05 | 5.18 | 0.00 | 52.86 | 67.50% |
|    | 0.25 | 0.00 | 0.00 | 0.00 | 100% |
|    | 0.20 | 0.00 | 0.00 | 0.00 | 100% |
|    | 0.15 | 0.00 | 0.00 | 0.00 | 100% |
| 100 | 0.12 | 0.00 | 0.00 | 0.00 | 100% |
|    | 0.10 | 0.00 | 0.00 | 0.00 | 100% |
|    | 0.08 | 0.15 | 0.00 | 4.98 | 98.54% |
|    | 0.05 | 5.02 | 0.00 | 53.84 | 67.56% |

Table 6: Means of evaluation metrics for dynamic networks simulated from Scenario 2, varying $c_{\tau,1}$.

| $n$ | $c_{\tau,1}$ | $|\widehat{K} - K| \downarrow$ | $d(\widehat{\mathcal{C}}|\mathcal{C}) \downarrow$ | $d(\mathcal{C}|\widehat{\mathcal{C}}) \downarrow$ | $C(\mathcal{G}, \mathcal{G}') \uparrow$ |
|---|---|---|---|---|---|
|    | 0.25 | 0.00 | 0.00 | 0.00 | 100% |
|    | 0.20 | 0.00 | 0.00 | 0.00 | 100% |
|    | 0.15 | 0.00 | 0.00 | 0.00 | 100% |
| 50 | 0.12 | 0.00 | 0.00 | 0.00 | 100% |
|    | 0.10 | 0.00 | 0.00 | 0.00 | 100% |
|    | 0.08 | 0.02 | 0.00 | 0.64 | 99.68% |
|    | 0.05 | 3.79 | 0.00 | 28.46 | 75.43% |
|    | 0.25 | 0.00 | 0.00 | 0.00 | 100% |
|    | 0.20 | 0.00 | 0.00 | 0.00 | 100% |
|    | 0.15 | 0.00 | 0.00 | 0.00 | 100% |
| 100 | 0.12 | 0.00 | 0.00 | 0.00 | 100% |
|    | 0.10 | 0.00 | 0.00 | 0.00 | 100% |
|    | 0.08 | 0.05 | 0.00 | 1.14 | 99.38% |
|    | 0.05 | 3.53 | 0.00 | 28.60 | 76.50% |

Tables 5–8 report results for **Scenarios 1-4** with $c_{\tau,1} \in \{0.05, 0.08, 0.10, 0.12, 0.15, 0.20, 0.25\}$. These results demonstrate that our method is relatively robust against the choices of $c_{\tau,1}$.

We also assess sensitivity to the input ranks $r_1 = r_2 = r \in \{10, 15, 20\}$ with results shown in Table 9. We find that the method remains robust across these choices.

**Frequent change points:** To assess performance under increasingly frequent changes, we conduct simulations with varying numbers of change points $K \in \{2, 7, 12\}$. The results are summarized in Table 10. As shown in Table 10, although localization accuracy decreases slightly as $K$ increases, all change points are consistently detected, demonstrating the robustness of the method even in high-frequency settings.

**Random change points:** To evaluate performance under randomly located structural changes, we conduct simulations in which change point locations are randomly sampled in each iteration rather than equally spaced. The number of change points is set to two for Scenario 1 and five for Scenario 2. The results are summarized in Table 11. We conclude that even with randomly located change points, our method demonstrates good and robust performance, outperforming the competitors in most settings.

Table 7: Means of evaluation metrics for dynamic networks simulated from Scenario 3, varying $c_{\tau,1}$.

| $n$ | $c_{\tau,1}$ | $|\widehat{K}-K|\downarrow$ | $d(\widehat{\mathcal{C}}|\mathcal{C})\downarrow$ | $d(\mathcal{C}|\widehat{\mathcal{C}})\downarrow$ | $C(\mathcal{G},\mathcal{G}')\uparrow$ |
|---|---|---|---|---|---|
|  | 0.25 | 1.00 | 50.00 | 0.00 | 75.00% |
|  | 0.20 | 0.96 | 48.00 | 0.00 | 76.00% |
|  | 0.15 | 0.64 | 32.00 | 0.00 | 84.00% |
| 50 | 0.12 | 0.39 | 19.58 | 0.08 | 90.17% |
|  | 0.10 | 0.19 | 9.64 | 0.14 | 95.11% |
|  | 0.08 | 0.09 | 4.30 | 0.50 | 97.61% |
|  | 0.05 | 4.27 | 0.36 | 32.54 | 71.55% |
|  | 0.25 | 0.43 | 21.50 | 0.00 | 89.25% |
|  | 0.20 | 0.15 | 7.52 | 0.02 | 96.23% |
|  | 0.15 | 0.00 | 0.02 | 0.02 | 99.98% |
| 100 | 0.12 | 0.00 | 0.02 | 0.02 | 99.98% |
|  | 0.10 | 0.00 | 0.02 | 0.02 | 99.98% |
|  | 0.08 | 0.06 | 0.02 | 1.08 | 99.57% |
|  | 0.05 | 4.04 | 0.02 | 32.76 | 73.62% |

Table 8: Means of evaluation metrics for dynamic networks simulated from Scenario 4, varying $c_{\tau,1}$.

| $n$ | $c_{\tau,1}$ | $|\widehat{K}-K|\downarrow$ | $d(\widehat{\mathcal{C}}|\mathcal{C})\downarrow$ | $d(\mathcal{C}|\widehat{\mathcal{C}})\downarrow$ | $C(\mathcal{G},\mathcal{G}')\uparrow$ |
|---|---|---|---|---|---|
|  | 0.25 | 2.67 | 83.20 | 0.00 | 62.47% |
|  | 0.20 | 1.19 | 28.40 | 0.00 | 85.63% |
|  | 0.15 | 0.13 | 2.60 | 0.00 | 98.67% |
| 50 | 0.12 | 0.01 | 0.22 | 0.02 | 99.88% |
|  | 0.10 | 0.00 | 0.02 | 0.02 | 99.98% |
|  | 0.08 | 0.00 | 0.02 | 0.02 | 99.98% |
|  | 0.05 | 0.75 | 0.02 | 11.94 | 93.36% |
|  | 0.25 | 0.01 | 0.20 | 0.00 | 99.90% |
|  | 0.20 | 0.00 | 0.00 | 0.00 | 100% |
|  | 0.15 | 0.00 | 0.00 | 0.00 | 100% |
| 100 | 0.12 | 0.00 | 0.00 | 0.00 | 100% |
|  | 0.10 | 0.00 | 0.00 | 0.00 | 100% |
|  | 0.08 | 0.00 | 0.00 | 0.00 | 100% |
|  | 0.05 | 0.89 | 0.00 | 12.46 | 92.55% |

**Temporal dependence:** We further evaluate robustness under temporal dependence by modifying **Scenario 1**. Instead of sampling $\mathbf{A}_{i,j,l}(t) \sim \text{Bernoulli}(\mathbf{P}_{i,j,l}(t))$ independently across time, we generate temporally dependent edges as follows:

$$\mathbf{A}_{i,j,l}(t+1) \sim \begin{cases} \text{Bernoulli}\big((1-\mathbf{P}_{i,j,l}(t+1))\pi + \mathbf{P}_{i,j,l}(t+1)\big), & \mathbf{A}_{i,j,l}(t) = 1, \\ \text{Bernoulli}\big(\mathbf{P}_{i,j,l}(t+1)(1-\pi)\big), & \mathbf{A}_{i,j,l}(t) = 0; \end{cases}$$

with $\pi \in \{0.005, 0.01, 0.02\}$ controlling dependence strength. Results in Table 12 show that our method maintains strong performance when temporal dependence is weak ($\pi \leq 0.01$) and degrades gradually as dependence increases.

**Comparisons with other competitors:** The method of Wang et al. (2025), denoted CPDonline, is designed for online change point detection in dynamic multilayer networks. The method of Li et al. (2024), denoted AutoCPD, is deep learning–based and typically requires myriad labeled data indicating whether a time window contains a change point to train a classifier, which is not available in our fully unsupervised setting.

Nevertheless, we conduct simulations comparing our method with CPDonline and AutoCPD. For AutoCPD, we trained the method using simulated labeled data and then applied it to a testing dataset. As shown in Table 13, our method (CPDmrdpg) outperforms CPDonline and AutoCPD across nearly all evaluation metrics for both small and moderate-sized networks.

Table 9: Means of evaluation metrics for dynamic networks simulated from Scenarios 1 and 3 with $n = 50, L = 4, r_3 = L$, varying input ranks $r_1 = r_2 = r$.

| Scenario | $r$ | $|\widehat{K} - K| \downarrow$ | $d(\widehat{\mathcal{C}}|\mathcal{C}) \downarrow$ | $d(\mathcal{C}|\widehat{\mathcal{C}}) \downarrow$ | $C(\mathcal{G}, \mathcal{G}') \uparrow$ |
|---|---|---|---|---|---|
|   | 10 | 0.00 | 0.00 | 0.00 | 100% |
| 1 | 15 | 0.01 | 0.00 | 0.42 | 99.86% |
|   | 20 | 0.00 | 0.00 | 0.00 | 100% |
|   | 10 | 0.15 | 7.80 | 0.30 | 95.96% |
| 3 | 15 | 0.19 | 9.64 | 0.14 | 95.11% |
|   | 20 | 0.05 | 2.70 | 0.20 | 98.55% |

Table 10: Means of evaluation metrics for dynamic networks simulated from Scenario 1 with $n = 100$ and $L = 8$, varying $K$.

| $K$ | $|\widehat{K} - K| \downarrow$ | $d(\widehat{\mathcal{C}}|\mathcal{C}) \downarrow$ | $d(\mathcal{C}|\widehat{\mathcal{C}}) \downarrow$ | $C(\mathcal{G}, \mathcal{G}') \uparrow$ |
|---|---|---|---|---|
| 2 | 0.00 | 0.00 | 0.00 | 100% |
| 7 | 0.00 | 1.00 | 1.00 | 96.08% |
| 12 | 0.00 | 1.00 | 1.00 | 94.19% |

### G.2 ADDITIONAL DETAILS AND RESULTS IN SECTION 4.2

This section provides a detailed analysis of the U.S. air transportation network data, evaluates the performance of competing methods (introduced in Section 4.1), and presents the constructed confidence intervals using the procedure in Section 3.1.

**The U.S. air transportation network data** consist of monthly data from January 2015 to June 2022 ($T = 90$) and are available from Bureau of Transportation Statistics (2022). Each node corresponds to an airport and each layer represents a commercial airline. A directed edge in a given layer indicates a direct flight operated by a specific commercial airline between two airports. We choose the $L = 4$ airlines with the highest flight volumes and the $n = 50$ airports with the most departures and arrivals.

Table 14 reports the detected change points for all methods. Although the kerSeg method using networks as input demonstrates a good performance in the simulation study, it detects an excessive number of change points in this real data experiment, making the results unreliable and raising concerns about false positives. Similarly, the kerSeg method that uses layer-wise Frobenius norms as input has detected change points that are too close, yielding clusters of change points that could potentially be grouped together. On the contrary, the gSeg method that uses the Frobenius norms as input detects too few change points, while the gSeg method using networks as input has detected too many change points. The proposed CPDmrdpg method (Algorithm 1) identifies five change points that align well with known disruptions and policy changes in the aviation sector, as discussed below.

The change point in December 2015 coincides with increased regulatory scrutiny over airline consolidation, following concerns raised by the American Antitrust Institute about reduced market competition after a series of mergers. The June 2017 change point aligns with the proposal of the Aviation Innovation, Reform and Reauthorization Act, which advocated for privatizing air traffic control and influenced route planning among carriers. Moreover, the February 2019 change point follows the U.S. government shutdown (December 2018 - January 2019), which caused Transportation Security Administration staffing shortages and significant operational disruptions, prompting stabilization efforts in the months that followed. Lastly, the most significant structural disruptions emerged in February 2020 and February 2021, aligning with the initial shock and continued fallout of the COVID-19 pandemic, which triggered widespread flight cancellations, demand collapse and structural reconfiguration in the aviation industry.

Table 15 reports the detected change point from Algorithm 1 and the $95\%$ confidence intervals constructed using the procedure in Section 3.1.

Table 11: Means of evaluation metrics for networks simulated from Scenarios 1 and 2 under randomly located change points.

| $n$ | Scenario | Method | $|\widehat{K} - K| \downarrow$ | $d(\widehat{\mathcal{C}}|\mathcal{C}) \downarrow$ | $d(\mathcal{C}|\widehat{\mathcal{C}}) \downarrow$ | $C(\mathcal{G}, \mathcal{G}') \uparrow$ |
|---|---|---|---|---|---|---|
| 50 | 1 | CPDmrdpg | 0.05 | 2.58 | 0.77 | 98.84% |
| | | gSeg (nets.) | 1.40 | Inf | Inf | 31.82% |
| | | kerSeg (nets.) | 0.25 | Inf | Inf | 94.53% |
| | | gSeg (frob.) | 0.60 | Inf | Inf | 83.10% |
| | | kerSeg (frob.) | 0.35 | Inf | Inf | 93.93% |
| | 2 | CPDmrdpg | 0.59 | 7.48 | 0.94 | 95.87% |
| | | gSeg (nets.) | 1.90 | 35.55 | 5.65 | 82.96% |
| | | kerSeg (nets.) | 0.25 | 3.05 | 1.25 | 98.23% |
| | | gSeg (frob.) | 0.50 | Inf | Inf | 93.15% |
| | | kerSeg (frob.) | 0.60 | 0.45 | 10.20 | 96.43% |
| 100 | 1 | CPDmrdpg | 0.05 | 2.57 | 1.41 | 98.54% |
| | | gSeg (nets.) | 1.40 | Inf | Inf | 29.26% |
| | | kerSeg (nets.) | 0.20 | Inf | Inf | 94.25% |
| | | gSeg (frob.) | 0.65 | Inf | Inf | 78.30% |
| | | kerSeg (frob.) | 0.40 | Inf | Inf | 96.68% |
| | 2 | CPDmrdpg | 0.43 | 5.03 | 0.95 | 96.80% |
| | | gSeg (nets.) | 1.75 | Inf | Inf | 80.47% |
| | | kerSeg (nets.) | 0.40 | 2.15 | 1.55 | 98.81% |
| | | gSeg (frob.) | 0.35 | 0.90 | 2.80 | 98.06% |
| | | kerSeg (frob.) | 0.30 | 0.25 | 4.80 | 97.97% |

Table 12: Means of evaluation metrics for dynamic networks simulated from Scenario 1 with $n = 50$, varying levels of temporal dependence $\pi$.

| $\pi$ | $|\widehat{K} - K| \downarrow$ | $d(\widehat{\mathcal{C}}|\mathcal{C}) \downarrow$ | $d(\mathcal{C}|\widehat{\mathcal{C}}) \downarrow$ | $C(\mathcal{G}, \mathcal{G}') \uparrow$ |
|---|---|---|---|---|
| 0.000 | 0.01 | 0.00 | 0.43 | 99.86% |
| 0.005 | 0.05 | 0.00 | 0.20 | 99.90% |
| 0.010 | 0.60 | 0.00 | 15.00 | 93.90% |
| 0.020 | 5.50 | 0.00 | 57.00 | 62.60% |

Table 13: Means of evaluation metrics for dynamic networks simulated from Scenario 1, comparing CPDmrdpg with competing methods.

| $n$ | Method | $|\widehat{K} - K| \downarrow$ | $d(\widehat{\mathcal{C}}|\mathcal{C}) \downarrow$ | $d(\mathcal{C}|\widehat{\mathcal{C}}) \downarrow$ | $C(\mathcal{G}, \mathcal{G}') \uparrow$ |
|---|---|---|---|---|---|
| 50 | CPDmrdpg | 0.01 | 0.00 | 0.42 | 99.86% |
| | CPDonline | 0.00 | 3.00 | 3.00 | 95.13% |
| | AutoCPD | 0.00 | 0.69 | 0.69 | 99.12% |
| 100 | CPDmrdpg | 0.00 | 0.00 | 0.00 | 100% |
| | CPDonline | 0.00 | 2.00 | 2.00 | 97.05% |
| | AutoCPD | 0.97 | 0.89 | 16.17 | 88.29% |

Table 14: Detected change points for the U.S. air transportation network data.

| Method | Detected change points |
|---|---|
| CPDmrdpg | 2015-12, 2017-06, 2019-02, 2020-02, 2021-02 |
| gSeg (nets.) | 2015-11, 2016-10, 2017-09, 2018-09, 2019-09, 2020-10, 2021-08 |
| kerSeg (nets.) | 2015-11, 2016-03, 2016-10, 2017-05, 2017-09, 2018-05, 2018-10 |
| | 2019-03, 2019-09, 2020-03, 2020-10, 2021-03, 2021-09 |
| gSeg (frob.) | 2015-11, 2020-01, 2021-03 |
| kerSeg (frob.) | 2015-11, 2017-10, 2020-01, 2021-03, 2021-05, 2021-09, 2022-01 |

Table 15: Detected change point from Algorithm 1 and $95\%$ confidence intervals via Section 3.1 for the U.S. air transportation network data.

| Detected change points | Time point | Confidence interval |
|---|---|---|
| 2015-12 | 12 | $(11.55, 12.41)$ |
| 2017-06 | 30 | $(28.79, 30.98)$ |
| 2019-02 | 50 | $(49.67, 53.22)$ |
| 2020-02 | 62 | $(59.66, 60.36)$ |
| 2021-02 | 74 | $(73.58, 74.27)$ |

