# OpenReview forum: "Change Point Localization and Inference in Dynamic Multilayer Networks"
_ICLR.cc/2026/Conference — ICLR 2026 Poster_

### Official Review · Reviewer_USG5 · 2025-10-24

**Soundness:** 2
**Presentation:** 3
**Contribution:** 2
**Rating:** 4
**Confidence:** 4

**Summary:**

This paper develops a two-stage CUSUM pipeline for change-point detection in dynamic multilayer random dot product graphs. Stage-1 performs seeded/binary segmentation with a tensor CUSUM; Stage-2 applies a low-rank TH-PCA/matched-filter refinement that exploits shared latent positions across layers, yielding a $\sqrt{L}$ SNR gain and $O(\kappa^{-2})$ localization rate under large minimal spacing and global layer-weight changes. Proofs establish consistency and limit laws; simulations and small real data are provided.

**Strengths:**

The paper is clearly written; the model, assumptions, and the two-stage algorithm are easy to follow.

Clean multilayer insight: the Stage-2 projection/matched filter makes precise how low rank and layer pooling provide $\sqrt{L}$ SNR gains.

Technical development is careful (ensuring consistency and limiting distribution for the refined estimator); a standardized pipeline (screen then refine) is appealing.

**Weaknesses:**

1. $\textbf{Scope (extension vs.~novelty):}$ The contribution appears closely aligned with prior multilayer RDPG pipelines [1] that (i) \emph{screen} for candidate changes via CUSUM/binary segmentation–type statistics and then (ii) refine via a low-rank projection step (and the target is similar). In the current draft, the primary difference seems to be a regime shift (online $\rightarrow$ offline) while keeping the same modeling assumptions (multilayer RDPG/GRDPG) and the same screen–then–refine structure and target. As written, this reads more like an extension to the offline setting than a fully new methodological framework.


2. $\textbf{Narrow regime.}$ The methodology focuses on time-invariant latent positions with only changes in global layer weights. This means that the communities of nodes remain consistent over time; in this scenario, the change points appear to be narrowly defined, as they cannot disrupt the community structure of the networks. Many practical settings involve node-localized changes alongside shifts in community structure, where the latent positions $X(t)$ should also be able to vary (See [2] for an example). The current scope does not account for these regimes.


3. $\textbf{Spacing dependence.}$ The paper assumes a strong gap at the order $T$--with a gap at the order of $T$ and the number of change points is $O(1)$, spacing-free $\ell_0$ approach can already achieve $\textbf{exact selection}$ and the same $O(\kappa^{-2})$ localization approach without a minimal spacing condition. In contrast, using binary segmentation requires setting up the minimal spacing window, which is not necessary and could be misleading. For example. The paper effectively must generate evenly spaced change points. If, instead, change times are drawn i.i.d.\ from a distribution—e.g.,
$
\eta_1,\ldots,\eta_K \sim \mathrm{Unif} \{ 2,\ldots,T \},
$
the minimal spacing
$
\Delta_{\min}:=\min_{1\le k\le K+1}\bigl(\eta_k-\eta_{k-1}\bigr)
$
typically collapses. This destroys the large-spacing assumption required by a CUSUM screen and makes any fixed spacing window ill-posed. By contrast, spacing-free $\(\ell_0\)$ approaches remain optimal/consistent in this setting and do not require enforcing even spacing.


For the $\ell_0$ approach to estimation of dynamic (possible multilayer) networks, I refer to [2], but I believe more literature can be found.

For comparison between CUSUM and $\ell_0$ approach, I refer to [3]. See Section 4, 'However, compared to the $l_0$-penalization methods, WBS is computationally more expensive and involves more tuning parameters.''.

4. Simulations use (nearly) evenly spaced changes, reinforcing the strong-spacing assumption; real-data experiments also use very small $n,T,L$, This leaves the reader with the impression that the method hinges on large spacing and may not scale or adapt to realistic, short-gap or node–localized scenarios.

[1] Wang, Fan, et al. "Multilayer random dot product graphs: Estimation and online change point detection." Journal of the Royal Statistical Society Series B: Statistical Methodology (2025): qkaf051.

[2] Zhao, P., Bhattacharya, A., Pati, D., & Mallick, B. K. (2022). Factorized fusion shrinkage for dynamic relational data. arXiv preprint arXiv:2210.00091.

[3] ''Wang, Daren, Yi Yu, and Alessandro Rinaldo. "Univariate mean change point detection: Penalization, CUSUM and optimality." Electronic Journal of Statistics 14 (2020): 1917-1961.

**Questions:**

1. Comparison with $\ell_0$ approaches. Please include a discussion comparing against spacing-free $\ell_0$ methods. Proper references should be cited and compared; I list some, but I think more can likely be found.

2. I think including a vectorized version of multiplayer networks and then using binary segmentation is worth being included as a baseline to see how much the improvement in stage 2 is.

3. To demonstrate robustness beyond the current (near-)even spacing and global-weight-jump setting, please add stress tests that randomly change times i.i.d. from a distribution.


The paper has very strong assumptions, and the model is not very applicable given those assumptions. However, the paper is technically solid, and the TH-PCA refinement plus distributional results are valuable. I would not oppose acceptance if proper questions are addressed or at least discussed.

---

> ### Author Response · Authors · 2025-11-19
>
> We thank the reviewer for the insightful and constructive comments.
>
> **Comparison with [1].** We first emphasize that online and offline change point detection are fundamentally different tasks. Online detection aims to identify, as quickly as possible, after a change has occurred, using only past and current data. Offline localization, in contrast, typically involves multiple change points and seeks to estimate both their total number and precise locations by leveraging the entire dataset.
>
> While [1] studied online change point detection in multilayer networks, their algorithm differs substantially from ours. In their method (see Section 3 of their paper), at each time $t$, a low-rank tensor-based estimator is used to compute averages over segments, and the Frobenius norm of the difference is compared against a pre-specified threshold. Exceeding this threshold triggers a detection; otherwise, the procedure moves to $t+1$. They derive a localization bound of order
> $$
> \kappa^{-2} \big(d^2m_{\max} + nd + Lm_{\max}\big) \log(\Delta/\alpha),
> $$
> where $\kappa$ is magnitude of the change, $n$ the number of nodes, $L$ the number of layers,  $d$ the latent dimension, $m_{\max}$ a rank parameter, $\alpha$ the Type-I error level and $\Delta$ the location of the true change point.
>
> In contrast, the offline problem requires globally optimal localization and correct estimation of the total number of change points, tasks that online sequential methods are not designed to address. Our paper develops a novel two-stage procedure tailored specifically to the offline setting.
> Stage I performs coarse localization via seeded binary segmentation with CUSUM statistics and Stage II refines these candidates using localized scan statistics built upon tensor-based low-rank estimation. This yields a substantially sharper localization rate of order
> $$
> \kappa_k^{-2}\log(T),
> $$
> and we further guarantee that, with high probability, the estimated number of change points satisfies
> $$
> \widetilde{K} = K.
> $$
> Beyond localization, our work also derives limiting distributions for the refined estimators and constructs fully data-driven confidence intervals, results that, to the best of our knowledge, have not previously appeared in the network literature.
>
> We will include this clarification in the final manuscript.
>
>
> **Time-invariant latent positions.** We thank the reviewer for raising this point. We clarify the modeling scope from two complementary angles.
>
> First, our model does allow certain forms of latent-position changes. Although our main focus is on changes in global layer weights, the model formulation naturally includes cases where the effective latent structure of the network changes across segments. For example, when $L = 1$ and $d = 3$, consider
> $$
> W_{(1)}(\eta) =
> \begin{pmatrix}
> 1 & 0 & 0 \\\\
> 0 & 0 & 0 \\\\
> 0 & 0 & 1
> \end{pmatrix}
> \quad\text{and}\quad
> W_{(1)}(\eta+1) =
> \begin{pmatrix}
> 0 & 0 & 0 \\\\
> 0 & 1 & 0 \\\\
> 0 & 0 & 1
> \end{pmatrix}.
> $$
> For a latent position vector $X_1 = (X_{1,1}, X_{1,2}, X_{1,3})^{\top} \in \mathbb{R}^3$, it holds that
> $$
> X_1^{\top}W_{(1)}(\eta)X_1 = X_{1,1}X_{1,1} + X_{1,3}X_{1,3}, \quad X_1^{\top} W_{(1)}(\eta+1)X_1 = X_{1,2}X_{1,2} + X_{1,3}X_{1,3}.
> $$
> This means that the first two coordinates interchange their roles across the change point $\eta$, while the third coordinate remains invariant. Thus, although the node-level latent vector still lies in $\mathbb{R}^3$, the effective latent structure governing the edge probabilities changes between the two segments. This example illustrates that our modeling framework can accommodate certain forms of latent-position shifts, not only changes in global layer weights.
>
> Second, our procedure is empirically robust to moderate latent-position perturbations. In Scenario 2 of Section 4.1, we introduce perturbations to the latent positions across segments. The proposed method continues to perform well in this setting, suggesting that our procedure is not overly restrictive and can tolerate moderate deviations from the time-invariant latent position assumption.
>
>
>
>
>
> **References**
>
> [1] Fan Wang, Wanshan Li, Oscar Hernan Madrid Padilla, Yi Yu, and Alessandro Rinaldo. Multilayer random dot product graphs: estimation and online change point detection. Journal of the Royal Statistical Society, Series B, 2025.

---

> ### Author Response · Authors · 2025-11-19
>
> **Spacing constraints and comparison with  $\ell_0$-based approaches.** We thank the reviewer for raising these points. We first clarify that our method employs seeded binary segmentation (SBS) introduced in [3], rather than wild binary segmentation (WBS). SBS has a substantially lower computational cost, for example, $O(T \log (T))$ in the univariate case, because it leverages a predetermined collection of seeded intervals, making it well suited for large $T$ and high-dimensional network settings.
>
>
>
> Regarding $\ell_0$-based approaches, we agree that $\ell_0$-penalized methods do not require a minimal spacing condition. However, they typically incur a higher computational cost than SBS. For example, in the univariate case, dynamic programming as introduced in [1] incurs a cost of $O(T^2)$, and even more advanced procedures such as the pruned exact linear time (PELT) algorithm [2] retain a worst-case complexity of $O(T^2)$. These approaches become computationally demanding for long sequences or high-dimensional network data, whereas SBS offers a far more scalable alternative.
>
>
> The minimal spacing requirement in our method arises from the greedy selection rule, which selects the interval with the maximal CUSUM above a predefined threshold. This requirement is not intrinsic to SBS itself. In the univariate setting, the SBS framework in [3] incorporates a narrowest-over-threshold (NOT) selection rule, which identifies the shortest interval whose CUSUM statistic exceeds a predefined threshold and relaxes spacing constraints while retaining optimal localization guarantees.
>
>
>
> In our implementation, we adopt the greedy selection rule and therefore require a minimal spacing condition. In principle, replacing the greedy step with a NOT-type rule, as in [3], would remove the need for evenly spaced change points. However, extending NOT in our dynamic multilayer network setting is substantially more challenging, and preliminary analysis indicates that such an adaptation yields a localization rate worse than the $\kappa_k^{-2}\log(T)$ rate achieved with the greedy rule.  We will incorporate this discussion and the relevant references into the final manuscript.
>
>
> **Comparison with vectorized approaches.**
> Our method employs a two-stage approach, where Stage I utilizes a vectorized multilayer network representation combined with SBS. As established in Proposition 4 (Section D.2 of the Appendix), the change point localization error rate after Stage I is of the order
> $$
>       \log(T) \bigg( \frac{n \sqrt{L} \log^{1/2}(T)}{\kappa_k^2} + \frac{\sqrt{\Delta}}{\kappa_k}  \bigg),
> $$
> where $\Delta$ denotes the minimal spacing between consecutive change points. Stage II then applies tensor-based low-rank estimation to refine these preliminary estimates. The localization error rate after Stage II is significantly improved to
> $$
> \frac{\log(T)}{\kappa_k^{2}}.
> $$
> We will include this discussion in the final manuscript.
>
>
> **Simulations with randomly change point locations.** As suggested by the reviewer,  we conducted additional simulations for Scenarios 1 and 2,  where the change point locations were randomly sampled in each iteration rather than fixed at equally spaced positions.  The results, summarized in Tables 1 and 2, show that our method continues to exhibit strong and robust performance, outperforming competing methods in most settings even under randomly positioned change points.
>
>
> **References**
>
> [1] Felix Friedrich, Angela Kempe, Volkmar Liebscher, and Gerhard Winkler. Complexity penalized m-estimation: fast computation. Journal of Computational and Graphical Statistics, 17(1):201–224, 2008.
>
> [2] Rebecca Killick, Paul Fearnhead, and Idris A Eckley. Optimal detection of changepoints with a linear computational cost. Journal of the American Statistical Association, 107(500):1590–1598, 2012.
>
> [3] Solt Kovács, Peter Bühlmann, Housen Li, and Axel Munk. Seeded binary segmentation: a general methodology for fast and optimal changepoint detection. Biometrika, 110(1):249–256, 2023.

---

> ### Author Response · Authors · 2025-11-19
>
> **Table 1:** Means of evaluation metrics for networks simulated from Scenario 1 with randomly sampled change point locations.
>
> | $n$     | Method         | $\lvert \widehat{K} - K \rvert \downarrow$ | $d(\widehat{\mathcal{C}} \mid \mathcal{C}) \downarrow$ | $d(\mathcal{C} \mid \widehat{\mathcal{C}}) \downarrow$ | $C(\mathcal{G}, \mathcal{G'}) \uparrow$ |
> | ------- | -------------- | ------------------------------------------ | ------------------------------------------------------ | ------------------------------------------------------ | --------------------------------------- |
> | **50**  | CPDmrdpg       | 0.05                                       | 2.58                                                   | 0.77                                                   | 98.84%                                  |
> |         | gSeg (nets.)   | 1.40                                       | Inf                                                    | Inf                                                    | 31.82%                                  |
> |         | kerSeg (nets.) | 0.25                                       | Inf                                                    | Inf                                                    | 94.53%                                  |
> |         | gSeg (frob.)   | 0.60                                       | Inf                                                    | Inf                                                    | 83.10%                                  |
> |         | kerSeg (frob.) | 0.35                                       | Inf                                                    | Inf                                                    | 93.93%                                  |
> | **100** | CPDmrdpg       | 0.05                                       | 2.57                                                   | 1.41                                                   | 98.54%                                  |
> |         | gSeg (nets.)   | 1.40                                       | Inf                                                    | Inf                                                    | 29.26%                                  |
> |         | kerSeg (nets.) | 0.20                                       | Inf                                                    | Inf                                                    | 94.25%                                  |
> |         | gSeg (frob.)   | 0.65                                       | Inf                                                    | Inf                                                    | 78.30%                                  |
> |         | kerSeg (frob.) | 0.40                                       | Inf                                                    | Inf                                                    | 96.68%                                  |

---

> ### Author Response · Authors · 2025-11-19
>
> **Table 2**: Means of evaluation metrics for networks simulated from Scenario 2 with randomly sampled change point locations..
>
> | $n$     | Method         | $\lvert \widehat{K} - K \rvert \downarrow$ | $d(\widehat{\mathcal{C}} \mid \mathcal{C}) \downarrow$ | $d(\mathcal{C} \mid \widehat{\mathcal{C}}) \downarrow$ | $C(\mathcal{G}, \mathcal{G'}) \uparrow$ |
> | ------- | -------------- | ------------------------------------------ | ------------------------------------------------------ | ------------------------------------------------------ | --------------------------------------- |
> | **50**  | CPDmrdpg       | 0.59                                       | 7.48                                                   | 0.94                                                   | 95.87%                                  |
> |         | gSeg (nets.)   | 1.90                                       | 35.55                                                  | 5.65                                                   | 82.96%                                  |
> |         | kerSeg (nets.) | 0.25                                       | 3.05                                                   | 1.25                                                   | 98.23%                                  |
> |         | gSeg (frob.)   | 0.50                                       | Inf                                                    | Inf                                                    | 93.15%                                  |
> |         | kerSeg (frob.) | 0.60                                       | 0.45                                                   | 10.20                                                  | 96.43%                                  |
> | **100** | CPDmrdpg       | 0.43                                       | 5.03                                                   | 0.95                                                   | 96.80%                                  |
> |         | gSeg (nets.)   | 1.75                                       | Inf                                                    | Inf                                                    | 80.47%                                  |
> |         | kerSeg (nets.) | 0.40                                       | 2.15                                                   | 1.55                                                   | 98.81%                                  |
> |         | gSeg (frob.)   | 0.35                                       | 0.90                                                   | 2.80                                                   | 98.06%                                  |
> |         | kerSeg (frob.) | 0.30                                       | 0.25                                                   | 4.80                                                   | 97.97%                                  |

---

> > ### Comment · Reviewer_USG5 · 2025-11-20
> > **Response to the authors**
> >
> > I really appreciate the authors' clarification, but I still have concerns about the following two points.
> >
> > First, as far as I can tell, the current estimation pipeline does not enforce any entrywise sparsity in the weight matrices $W_t$ (if I'm not wrong):
> >
> > - Stage I works with a vectorized multilayer representation and CUSUM/SBS.
> > - Stage II refines via a low-rank tensor / PCA-type factorization and then estimates $W_t$ given $\widehat X$ essentially by least squares/tensor regression.
> >
> > In that setup, the estimated $\widehat W_t$ will generically be dense, so the example is out of scope.
> >
> > More fundamentally, because the model is
> > $
> > P_t = X W_t X^\top,
> > $
> > the pair $(X, W_t)$ is only identified up to orthogonal transforms:
> > $
> > X \mapsto XQ,\quad W_t \mapsto Q^\top W_t Q,\qquad Q \in \mathbb{O}(d),
> > $
> > so any entrywise sparsity pattern in $W_t$ is not invariant.
> >
> >
> > In addition, I still do not think the current response adequately addresses $\ell_0$ approaches in the $\textbf{dynamic network}$ setting. The new cited works are all for (essentially) univariate or low-dimensional time series; they are quite far from the multilayer RDPG framework of this paper. By contrast, there now exist $\ell_0$-type methods specifically designed for dynamic relational data with low-rank latent structure --- (e.g., [1] [2]), which combine low-rank latent factors with $\ell_0$-style shrinkage on temporal differences. That method operates in essentially the same modeling regime (latent low-rank + change points) as the present paper.
> >
> > Because of this, a proper comparison or at least discussions with [1] [2] (and related $\ell_0$-based dynamic network methods) is important both conceptually and technically.
> >
> > Overall, I'm not opposed to the paper's contributions, but I think it gives the impression of under-citing or under-discussing closely related work, even when the modeling assumptions and goals are very similar (e.g., dynamic low-rank network models with $\ell_0$-type temporal regulations). A more systematic comparison and clearer literature discussion would help clarify the genuine novelty and practical advantages of the proposed approach.
> >
> > [1] Zhao, P., Bhattacharya, A., Pati, D., & Mallick, B. K. (2022). Factorized fusion shrinkage for dynamic relational data. arXiv preprint arXiv:2210.00091
> >
> > [2] Zhang, Yuzhao, et al. "Change point detection in dynamic networks via regularized tensor decomposition." Journal of Computational and Graphical Statistics 33.2 (2024): 515-524.

---

> ### Author Response · Authors · 2025-11-25
>
> **Time-invariant latent positions.** We sincerely thank the reviewer for the thoughtful comments regarding identifiability and sparsity. We fully agree with these points and acknowledge the oversight in our previous rebuttal. In particular, the current formulation assumes time-invariant latent positions.
>
> While this assumption simplifies exposition, it is natural to consider a broader model in which the latent positions $X(t)$ may vary across change points. For completeness, we outline how such an extension can be incorporated into the D-MRDPG framework, and how our methodology and guarantees extend to this setting.
>
> We redefine the D-MRDPG model so that the sequence of adjacency tensors is generated from a dynamic multilayer random dot product graph model with time-varying latent positions and time-varying layer weight matrices.
>
> Assume that there exist change points
>
> $$
> 0 = \eta_{0} < \eta_{1} < \dots < \eta_{K} < T = \eta_{K+1},
> $$
> such that for $t \in [T-1]$,
>
> $$
> \big( X(t),  W_{(1)}(t),  \dots,   W_{(L)}(t)\big) \neq \big( X(t+1),  W_{(1)}(t+1),  \dots,   W_{(L)}(t+1)\big),
> $$
>
> if and only if $t =  \eta_1,  \ldots, \eta_K$.
> We retain the definition of the $k$-th jump size,
> $$
> \kappa_k = \\|P(\eta_{k+1}) - P(\eta_k)\\|_{\mathrm F},
> $$
> where $P(t)$ is the probability tensor at time $t$.
>
> For any $0 \le s < t < e \le T$, let $\widetilde{\mathbf{P}}^{s,e}(t)$ denote the expected CUSUM-transformed tensor as in Eq.(2) of the main text. To ensure the Tucker low-rank structure required for TH-PCA, Assumption 1 must be modified as follows.
>
> Modified assumptions: (i) Mode-$1$ full-rank condition. Assume that
>
> $$
> \mbox{rank}(\mathcal{M}_1(\widetilde{\mathbf{P}}^{s,e}(t)))=d, \quad
>  \sigma_1( \mathcal{M}_1 (\widetilde{\mathbf{P}}^{s,e}(t) ))      \leq C\sigma_d(\mathcal{M}_1(\widetilde{\mathbf{P}}^{s,e}(t))), \quad
> \sigma_d(\mathcal{M}_1(\widetilde{\mathbf{P}}^{s,e}(t))) \geq C \sqrt{n}.
> $$
>
> When $X(t)$ is time-invariant, this reduces to Assumption 1(i) in the main text.
>
>
> (ii) Mode-3 low-rank condition.
> Let $m^{s,e}_t = \mathrm{rank}( \mathcal{M}_3(\widetilde{\mathbf{P}}^{s,e}(t)) )$. Assume
>
> $$
> \sigma_1(\mathcal{M}_3(\widetilde{\mathbf{P}}^{s,e}(t)) ) $$
>
> $$
> \leq C  \sigma_{\min} (\mathcal{M}_3(\widetilde{\mathbf{P}}^{s,e}(t)) ),
> $$
>
> and
>
> $$
> \sigma_{\min} (\mathcal{M}_3(\widetilde{\mathbf{P}}^{s,e}(t)) ) \geq C.
> $$
> This parallels Assumption 1(ii) in the main text.
>
> These conditions guarantee the necessary low-rankness of the expected CUSUM tensors. We acknowledge that, compared with the original assumptions, the extended conditions, particularly (i), are less direct or transparent in terms of the explicit structural properties of the network model.
>
> Under the modified assumptions above, and provided the signal-to-noise ratio condition
>
> $$
>  \kappa \sqrt{\Delta}  \geq C_{\mathrm{SNR}} \log(T) \sqrt{n L^{1/2} + d^2m_{\max} +  nd+Lm_{\max} },
> $$
>
> where  $m_{\max} = \max_{k \in [K+1]} \mathrm{rank}\big(\mathcal{M}_3(\mathbf{P}(\eta_k))\big)$,  the same two-stage algorithm continues to achieve the same localization rate as in Theorem 1.
> This shows that our methodology naturally extends to settings with time-varying latent positions, without altering either the algorithmic structure or the core statistical guarantees.
>
> We will include this discussion in the final manuscript.

---

> ### Author Response · Authors · 2025-11-25
>
> **Comparison with  $\ell_0$-based approaches.**
> We sincerely thank the reviewer for highlighting the importance of $\ell_0$-type methods specifically designed for dynamic networks and for emphasizing the need for a clearer comparison with closely related work. We address these points below.
>
>
> We begin by clarifying the general $\ell_0$-penalized approach commonly used in change point analysis (e.g. [4, 5]).
> Let $\mathcal{P}$ be a partition of $[T]$ into $K_{\mathcal{P}}$ disjoint intervals:
> $$
> \mathcal{P}
> = \big( (1,\ldots,i_1), (i_1+1,\ldots,i_2),\ldots, (i_{K_{\mathcal{P}}-1}+1,\ldots,i_{K_{\mathcal{P}}}-1) \big),
> $$
> for integers $1 < i_1 < \cdots < i_{K_{\mathcal{P}}} = T+1$ and $K_{\mathcal{P}} \geq 1$. For a tuning parameter $\lambda > 0$, the $\ell_0$-penalized estimator is defined as
> $$
> \widehat{\mathcal{P}} = \arg\min_{\mathcal{P}} \Big( \sum_{I \in \mathcal{P}} \mathcal{G}(I) + \lambda  \vert \mathcal{P}\vert \Big),
> $$
> where $\mathcal{G}$ is a chosen loss function.
>
>
> Reference [7] studied dynamic single-layer networks with time-varying latent factors under temporal dependence. They imposed a group-wise fusion structure (an $\ell_0$-type regularization) on the evolution of the latent factors. This formulation differs from the classical $\ell_0$-penalized approach in change point analysis: their penalty acts directly on the latent factors rather than on partitions of time. Their focus is primarily on estimation rather than localization, but their estimators may be combined with existing localization procedures (e.g. [1]) to improve accuracy. For example, [1] constructed test statistics by estimating each network separately; using the blockwise estimation strategy of [7] could potentially sharpen localization, an interesting direction for future exploration.
>
>
> Reference [6] studied dynamic single-layer networks embedded in a low-dimensional Euclidean space, where change points arise from shifts in the underlying embedding matrix. They adopt an $\ell_1$-penalized approach, a convex relaxation of the $\ell_0$-approach, that is computationally efficient and closely related to the fused Lasso [2]. They established the asymptotic guarantees:
> $$
> \lim_{n \to \infty} \mathbb{P}(\widetilde{K} = K) = 1
> \quad\mbox{and}\quad
> \max_{k \in [K]} \frac{|\widetilde{\eta}_k - \eta_k|}{\Delta} \xrightarrow{p} 0,
> $$
> which demonstrate consistent estimation.
>
>
> In contrast to these works, to the best of our knowledge, we are the first to study offline change point localization in dynamic multilayer networks, an intrinsically higher-order setting than the single-layer models considered in [6, 7]. Our change points are defined through layer-specific connectivity patterns, and as discussed earlier, our framework can be extended to allow both latent position changes and layer weight matrix changes. Building on a novel two-stage algorithm, we establish the non-asymptotic guarantee
> $$
> \mathbb{P} \Big( \widetilde{K} = K \mbox{ and } \vert \widetilde{\eta}_k - \eta_k \vert  \lesssim \frac{\log (T)}{\kappa_k^2},  \forall k \in [K] \Big) \geq 1 - CT^{-c}.
> $$
> This result not only guarantees consistency but also matches (up to a logarithmic factor) the minimax-optimal localization rate established in [3] for single-layer networks, and is sharper than the rates proved in [6].  Moreover, beyond localization, our work derives limiting distributions for refined estimators and constructs fully data-driven confidence intervals, results that, to the best of our knowledge, have not previously appeared in the network change point literature.
>
>
> We will incorporate the comparison with [6, 7]  in the final manuscript.
>
>
> **References**
>
> [1] Oscar Hernan Madrid Padilla, Yi Yu, and Carey E Priebe. Change point localization in dependent dynamic
> nonparametric random dot product graphs. Journal of Machine Learning Research, 23(234):1–59, 2022.
>
> [2] Robert Tibshirani, Michael Saunders, Saharon Rosset, Ji Zhu, and Keith Knight. Sparsity and smoothness via
> the fused lasso. Journal of the Royal Statistical Society Series B: Statistical Methodology, 67(1):91–108, 2005.
>
> [3] Daren Wang, Yi Yu, and Alessandro Rinaldo. Optimal change point detection and localization in sparse dynamic
> networks. The Annals of Statistics, 49(1):203–232, 2021.
>
> [4] Haotian Xu, Daren Wang, Zifeng Zhao, and Yi Yu. Change-point inference in high-dimensional regression
> models under temporal dependence. The Annals of Statistics, 52(3):999–1026, 2024.
>
> [5] Gengyu Xue, Haotian Xu, and Yi Yu. Change point localisation and inference in fragmented functional data.
> arXiv preprint arXiv:2405.05730, 2024.
>
> [6] Yuzhao Zhang, Jingnan Zhang, Yifan Sun, and Junhui Wang. Change point detection in dynamic networks via
> regularized tensor decomposition. Journal of Computational and Graphical Statistics, 33(2):515–524, 2024.
>
> [7] Peng Zhao, Anirban Bhattacharya, Debdeep Pati, and Bani K Mallick. Factorized fusion shrinkage for dynamic
> relational data. arXiv preprint arXiv:2210.00091, 2022.

---

> ### Comment · Reviewer_USG5 · 2025-11-25
> **Response to the authors**
>
> Thanks to the authors for the detailed response. It addresses most of my earlier concerns, and I am happy to increase my score.
>
> I have just one remaining conceptual comment.
>
> The response emphasizes that the $\ell_0$-type penalty in [7] “acts directly on the latent factors” rather than on partitions of time, in contrast to classical $\ell_0$ change-point formulations on $\mathcal{P}$. In the low-rank setting, however, I believe these two viewpoints are in fact very closely related, and in some cases essentially equivalent.
>
> For example, consider the univariate mean problem in [1], where they introduce both
> (i) a penalty on the mean vector (their equation (7)) and
> (ii) a penalty on the partition (their equation (8)),
>
> and then explicitly state:
>
> “We now make the simple observation that the optimization problems (7) and (8) with the same inputs yield the same change point estimators. To see this equivalence we will …”
>
> In other words, penalizing the piecewise-constant signal and penalizing the partition are just two parameterizations of the same underlying $\ell_0$ segmentation problem.
>
> By analogy, in a dynamic low-rank network model, an $\ell_0$-type penalty on temporal differences of latent factors (as in [7]) can often be reinterpreted as an implicit $\ell_0$ penalty on the induced segmentation of time, just expressed in the latent space rather than directly on $\mathcal{P}$. Conceptually, this seems closer to a reparameterization than a fundamentally different paradigm.
>
> This is mainly a clarification point; aside from this, I do not have further substantive concerns.
>
>
> [1] Wang, D., Yu, Y.,  Rinaldo, A. (2020).Univariate mean change point detection: Penalization, CUSUM and optimality. Electronic Journal of Statistics, 14, 1917–1961.

---

> > ### Author Response · Authors · 2025-12-01
> >
> > We sincerely thank the reviewer for the clarification and for raising the score.  We fully agree that an $\ell_0$-type penalty on temporal changes in vectors and an $\ell_0$-type penalty on the time partition are closely related and, in the univariate setting, essentially equivalent. In the dynamic latent-vector models,  the main difference between these two penalties arises from an identifiability issue (as also noted in the reviewer’s earlier comments): because the latent representation is determined only up to an orthogonal rotation, a penalty on changes in the latent vectors depends on the particular parameterization chosen, whereas a penalty on the time partition is invariant to such reparameterization.

---

### Official Review · Reviewer_yKYp · 2025-10-31

**Soundness:** 3
**Presentation:** 2
**Contribution:** 3
**Rating:** 6
**Confidence:** 3

**Summary:**

The paper proposes and analyzes a new moethod for offline change point localization and inference in dynamic multilayer networks (following the D-MRDPGs model). They prove consistency of their estimator, and derive limiting distributions of the refined estimators in general regimes. They provide methods for computing confidence intervals, and demonstrate the favourable performance of their estimator on both synthetic and real data.

**Strengths:**

The paper is original in that they consider a model that has not been studied before. This model is well motivated from the literature and appears in practice. To the best of their (and my) knowledge, this is the first result of its kind in the context of dynamic network data.

The paper is well written, they state the scalings very clearly which is important in the area of change points. Their theory is sound, and they use standard techniques/sub-methods from the change point literature such as CUSUM statistics. They also acknowledge potential improvements in the conclusion section which are sound.

The contribution is significant to some extent, as the experiments indicate an experimental improvement over previous models that seems to come from the fact that they are looking at a more specific problem than competing methods (some competing methods are nonparametric). However, I must mention that the techniques are not in themselves novel (at least at first glance), but it would be good if the authors could comment on this.

**Weaknesses:**

From my understanding, competing methods only work change point detection in single layer dynamic networks, or they are inherently nonparametric methods. Can these be adapted to the multilayer setting? Can you comment on how fair this comparison is? It is not clear to me at this point.

Also, implications from the experimental results could be made more clear, for example, by including figures in the main body of the text. You state that “across all scenarios, our method achieves best performance” but that does not seem to be true, as if I dig into Table 3 in Appendix F.1, it seems that in the n=50 case you do not clearly outperform kerSeg (nets.). Could you comment on this, and at least make it clear in the paper that you do not outperform in all cases, and maybe some intuition as to why? Showing some of these results graphically also would help for clarity.

The comparison with other methods on real world data should be made more clear (in Appendix F.2 for example), and brought into the main paper. This is very important, and can help better clarify the benefit of your method, which is currently not so clear.

**Questions:**

- Middle of page 3: “We allow all model parameters … to diverge with T”. Please clarify this, how do they relate to T exactly?

---

> ### Author Response · Authors · 2025-11-19
>
> We thank the reviewer for the detailed and helpful comments.
>
> **Competing methods.**  To the best of our knowledge, our work is the first to address offline change point detection in dynamic multilayer networks, and thus no existing methods are specifically designed for this setting. For this reason, there are no truly direct competitors.
>
> We included gSeg and kerSeg as baselines because they are general-purpose change point detection methods that have been applied to network data and typically outperform other generic offline procedures. Since both were originally developed for single-layer dynamic networks or vector-valued observations, some adaptation is necessary for multilayer data. To make the comparison as fair as possible, we consider two natural adaptations: (i) applying each method directly to the raw networks after collapsing them into a single enlarged network or a long vector (nets.), and (ii) applying them to the vector of layer-wise Frobenius norms (frob.).
>
> In addition, we compare our method with that of [1], which is specifically designed for online detection in dynamic multilayer networks. We refer to this method as CPDonline and evaluate it over 20 simulations. For reference, we also report the results of CPDmrdpg (our method). As shown in Table 1, applying an online procedure to a multiple-change-point offline problem may lead to missed change points or spurious detections.
>
> In summary, while competing methods were not originally designed for the multilayer offline setting, we adapt them using reasonable strategies. Across most scenarios, our method demonstrates superior performance.  We will include this discussion in the final manuscript.
>
>
> **Implications from the experimental results.**
> We are very grateful for the reviewer’s suggestion to convert our experiment results into a figure to enable a clearer visual comparison. We will prepare such a figure and include it in the final manuscript.
>
> We also thank the reviewer for pointing out that, in Scenario 3 with $n=50$, our method does not outperform the kerSeg (nets.) method. We will revise our wording accordingly to clarify that our method outperforms competitors in most scenarios in our simulation study, rather than claiming superiority in all scenarios.
>
>
> **Real data experiments.** Thank you for the helpful suggestion. We will move the comparison with other methods on the real data from the appendix into the main text to improve clarity in the final manuscript. In addition, we will strengthen the discussion of these results to more clearly highlight the benefits of our proposed method in the real-data setting.
>
>
> **Divergence.**
> The statement “we allow all model parameters to diverge with $T$” means that key structural quantities such as the number of nodes $n$, the number of layers $L$ and the latent dimension $d$ are allowed to grow as functions of the time horizon $T$. We do not impose constraints requiring, for example, $d$ or $L$ to remain of constant order; instead, our theoretical guarantees accommodate their growth as long as the signal-to-noise ratio condition stated in Assumption 2 is satisfied.
>
>
> **References**
>
> [1] Fan Wang, Wanshan Li, Oscar Hernan Madrid Padilla, Yi Yu, and Alessandro Rinaldo. Multilayer random dot product graphs: estimation and online change point detection. Journal of the Royal Statistical Society, Series B, 2025.

---

> > ### Author Response · Authors · 2025-11-19
> >
> > **Table 1:** Means of evaluation metrics for CPDonline and CPDmrdpg methods across all scenarios.
> >
> > | $n$  | Scenario | Method     | $\lvert \widehat{K} - K \rvert \downarrow$ | $d(\widehat{\mathcal{C}} \mid \mathcal{C}) \downarrow$ | $d(\mathcal{C} \mid \widehat{\mathcal{C}}) \downarrow$ | $C(\mathcal{G}, \mathcal{G'}) \uparrow$ |
> > |------|----------|------------|---------------------------------------------|--------------------------------------------------------|--------------------------------------------------------|------------------------------------------|
> > | **50** | **1**      | CPDmrdpg   | 0.0                                       | 0.0                                                    | 0.4                                                    | 99.86%                                   |
> > |      |          | CPDonline  | 0.0                                       | 3.0                                                    | 3.0                                                    | 95.13%                                   |
> > |      | **2**      | CPDmrdpg   | 0.0                                       | 0.0                                                    | 0.0                                                    | 100.0%                                   |
> > |      |          | CPDonline  | 0.7                                       | 14.0                                                   | 16.9                                                   | 67.58%                                   |
> > |      | **3**      | CPDmrdpg   | 0.2                                       | 9.6                                                    | 0.1                                                    | 95.11%                                   |
> > |      |          | CPDonline  | 0.5                                       | 37.1                                                   | 27.4                                                   | 55.53%                                   |
> > |      | **4**      | CPDmrdpg   | 0.0                                       | 0.0                                                    | 0.0                                                    | 99.98%                                   |
> > |      |          | CPDonline  | 2.8                                       | 46.4                                                   | 20.2                                                   | 49.19%                                   |
> > | **100** | **1**      | CPDmrdpg   | 0.0                                       | 0.0                                                    | 0.0                                                    | 100.0%                                   |
> > |      |          | CPDonline  | 0.0                                       | 2.0                                                    | 2.0                                                    | 97.05%                                   |
> > |      | **2**      | CPDmrdpg   | 0.0                                       | 0.0                                                    | 0.0                                                    | 100.0%                                   |
> > |      |          | CPDonline  | 1.6                                       | 24.5                                                   | 16.5                                                   | 66.55%                                   |
> > |      | **3**      | CPDmrdpg   | 0.0                                       | 0.0                                                    | 0.0                                                    | 99.98%                                   |
> > |      |          | CPDonline  | 1.3                                       | 14.1                                                   | 29.6                                                   | 68.92%                                   |
> > |      | **4**      | CPDmrdpg   | 0.0                                       | 0.0                                                    | 0.0                                                    | 100.0%                                   |
> > |      |          | CPDonline  | 1.7                                       | 12.2                                                   | 28.0                                                   | 69.99%                                   |

---

### Official Review · Reviewer_8CQ5 · 2025-11-02

**Soundness:** 3
**Presentation:** 3
**Contribution:** 3
**Rating:** 6
**Confidence:** 2

**Summary:**

This paper proposes a two-stage algorithm for offline change point localization in dynamic multilayer random dot product graphs. Stage I uses seeded binary segmentation with CUSUM statistics; Stage II refines estimates via tensor heteroskedastic PCA. The authors establish consistency for change point detection/localization, derive limiting distributions under vanishing and non-vanishing jump regimes, and provide a data-driven confidence interval procedure. Experiments on synthetic and real networks demonstrate strong performance.

**Strengths:**

1. The theoretical contributions are novel, deriving limiting distributions for change point estimators in network data. The localization rates match minimax optimality for single-layer networks while extending to multilayer settings.

2. The paper provides consistency results, limiting distributions, a data-driven confidence interval procedure, and extensive robustness checks.

**Weaknesses:**

While online change point detection in multilayer networks has been recently studied (Wang et al., 2025), the paper does not clearly articulate what specific technical challenges arise in the offline setting or why such an extension from online methods is non-trivial.

**Questions:**

Please refer to the question in the Weakness section.

---

> ### Author Response · Authors · 2025-11-19
>
> We thank the reviewer for the insightful comments.
>
> **Comparison with [1].**
> We first emphasize that online and offline change point detection are fundamentally different tasks. Online detection aims to identify, as quickly as possible, after a change has occurred, using only past and current data. Offline localization, in contrast, typically involves multiple change points and seeks to estimate both their total number and precise locations by leveraging the entire dataset.
>
>
> While [1] studied online change point detection in multilayer networks, their algorithm differs substantially from ours. In their method (see Section 3 of their paper), at each time $t$, a low-rank tensor-based estimator is used to compute averages over segments, and the Frobenius norm of the difference is compared against a pre-specified threshold. Exceeding this threshold triggers a detection; otherwise, the procedure moves to $t+1$. They derive a localization bound of order
> $$
> \kappa^{-2} \big(d^2m_{\max} + nd + Lm_{\max}\big) \log(\Delta/\alpha),
> $$
> where $\kappa$ is magnitude of the change, $n$ the number of nodes, $L$ the number of layers,  $d$ the latent dimension, $m_{\max}$ a rank parameter, $\alpha$ the Type-I error level and $\Delta$ the location of the true change point.
>
>
> In contrast, the offline problem requires globally optimal localization and correct estimation of the total number of change points, tasks that online sequential methods are not designed to address. Our paper develops a novel two-stage procedure tailored specifically to the offline setting.
> Stage I performs coarse localization via seeded binary segmentation with CUSUM statistics and Stage II refines these candidates using localized scan statistics built upon tensor-based low-rank estimation. This yields a substantially sharper localization rate of order
> $$
> \kappa_k^{-2}\log(T),
> $$
> and we further guarantee that, with high probability, the estimated number of change points satisfies
> $$
> \widetilde{K} = K.
> $$
> Beyond localization, our work also derives limiting distributions for the refined estimators and constructs fully data-driven confidence intervals, results that, to the best of our knowledge, have not previously appeared in the network literature.
>
> We will include this clarification in the final manuscript.
>
> **References**
>
> [1] Fan Wang, Wanshan Li, Oscar Hernan Madrid Padilla, Yi Yu, and Alessandro Rinaldo. Multilayer random dot product graphs: estimation and online change point detection. Journal of the Royal Statistical Society, Series B, 2025.

---

### Author Response · Authors · 2025-12-02

We study offline change point detection in dynamic multilayer networks and, to the best of our knowledge, ours is the first work to address this problem in the literature. We introduce a novel two-stage algorithm, combining seeded binary segmentation with a tensor-based refinement step, that achieves a sharp localization error rate. In addition, we derive the limiting distributions of the refined estimators and provide fully data-driven confidence intervals. To our knowledge, these inferential results are established for the first time in the network literature.

In response to the reviewers' comments, we provided substantial clarifications, extended analyses and additional experiments addressing the following points:

1.  **Competitors.**  We compared our method in detail with Wang et al. (2025), both algorithmically, where we propose a new two-stage procedure, and theoretically, where our localization guarantees are significantly sharper. We also conducted additional simulations comparing our method with the online multilayer procedure of  Wang et al. (2025), and our method consistently outperforms theirs. Reviewer USG5 also suggested several related papers; we have incorporated a detailed comparison with these papers.

2. **Minimal spacing condition.** We discussed how this assumption can be relaxed and conducted additional simulations in which change points are drawn randomly rather than evenly spaced. These experiments show that our method remains robust and outperforms competitors in most scenarios.

3. **Extension to allow changes in latent positions.** We added a detailed discussion of an extension that permits changes in latent positions. By modifying the key low-rank and spectral-gap assumptions, we showed that our method naturally extends to this setting without altering the core statistical guarantees.

We emphasize that we fully resolved the concerns of reviewer USG5, who explicitly noted satisfaction with our response and **raised their score** accordingly. Overall, we believe these clarifications, additional experiments and theoretical extensions significantly strengthen the paper's contributions and confirm its novelty.

**References**

Fan Wang, Wanshan Li, Oscar Hernan Madrid Padilla, Yi Yu, and Alessandro Rinaldo. Multilayer random dot product graphs: estimation and online change point detection. Journal of the Royal Statistical Society, Series B, 2025.

---

### Meta-Review · Area_Chair_yUNq · 2025-12-30

**Summary:**

The reviewers have raised the following major concerns:

(1) Conceptual comparison with a recent online detection method and other competing approaches.

(2) Interpretation of the numerical results.

(3) Scope, novelty, and narrow regime.

(4) Technical issues related to spacing constraints and the resulting restricted simulation setting.

**Reviewer Concerns:**

Based on the rebuttal and the follow-up discussions, I believe that most of the reviewers’ concerns have been adequately addressed. As such, the work constitutes a meaningful and worthwhile contribution to ICLR.

After briefly reviewing the revised manuscript, I have two additional suggestions:

(1) To improve the paper’s readability, particularly for a general audience, the authors may consider providing more intuition for the technical components. For instance, the complicated expressions in Assumptions 1 and 2 and Theorems 1 and 2 are difficult for non-experts to appreciate. Clarifying their physical or practical interpretations, or providing simple illustrative examples, would help readers better understand both the conditions and the conclusions.

(2) In my view, the paper may be better suited for a theoretical statistics or mathematics journal. As a primarily theoretical contribution, the lengthy proofs in Appendices D and E require rigorous verification by all reviewers to ensure correctness. However, such thorough validation may be difficult to achieve within the tight time constraints of a conference review process.

**Reviewer Scores:**

Reviewer USG5, who initially provided the only negative assessment, has indicated a willingness to revise the score positively.

---

### Decision · Program_Chairs · 2026-01-26

Accept (Poster)